# Inferring histology-associated gene expression gradients in spatial transcriptomic studies

Jan Kueckelhaus[1,2,12] ✉, Simon Frerich [3,4,12], Jasim Kada-Benotmane [1,5], Christina Koupourtidou [6,7], Jovica Ninkovic[6,7], Martin Dichgans[3,7,8], Juergen Beck [5], Oliver Schnell[2] & Dieter Henrik Heiland [1,2,9,10,11] ✉

Spatially resolved transcriptomics has revolutionized RNA studies by aligning RNA abundance with tissue structure, enabling direct comparisons between histology and gene expression. Traditional approaches to identifying signature genes often involve preliminary data grouping, which can overlook subtle expression patterns in complex tissues. We present Spatial Gradient Screening, an algorithm which facilitates the supervised detection of histology-associated gene expression patterns without prior data grouping. Utilizing spatial transcriptomic data along with single-cell deconvolution from injured mouse cortex, and TCR-seq data from brain tumors, we compare our methodology to standard differential gene expression analysis. Our findings illustrate both the advantages and limitations of cluster-free detection of gene expression, offering more profound insights into the spatial architecture of transcriptomes. The algorithm is embedded in SPATA2, an open-source framework written in R, which provides a comprehensive set of tools for investigating gene expression within tissue.

In recent years, significant advancements have been made in the field of spatial biology, providing essential tools for profiling gene, protein, and metabolic expression in biological tissues[1]. These developments have been crucial in various research domains, such as developmental biology[2], neuroscience[3], and cancer microenvironment[4] studies. The discoveries emerging from these studies have greatly enhanced our understanding of spatial organization in different tissues. While healthy tissue typically exhibits a highly ordered structure, diseases can disrupt this order, leading to a complex range of dynamic alterations.

The human neocortex, for instance, is generally understood through a well-established model of six cortical layers. This organized structure contrasts sharply with the chaotic and heterogeneous architecture of malignant CNS tumors, a phenomenon encapsulated in the concept of intertumoral heterogeneity[5,6]. The heterogeneous complexity of pathologies in general, benign and malignant alike, poses significant challenges in medical care, given that effective treatments rely on recurring biological patterns or functions that can be targeted. In this context, ensuing efforts of the past decades have resulted in the

---

[1]Microenvironment and Immunology Research Laboratory, Medical Center, Faculty of Medicine, Freiburg University, Freiburg, Germany. [2]Department of Neurosurgery, Medical Center, Faculty of Medicine, Erlangen University, Erlangen, Germany. [3]Institute for Stroke and Dementia Research (ISD), University Hospital, LMU Munich, Munich, Germany. [4]Graduate School of Systemic Neurosciences, LMU Munich, Munich, Germany. [5]Department of Neurosurgery, Medical Center, Faculty of Medicine, Freiburg University, Freiburg, Germany. [6]Department of Cell Biology and Anatomy, Biomedical Center (BMC), LMU Munich, Munich, Germany. [7]Munich Cluster for Systems Neurology (SyNergy), Munich, Germany. [8]German Center for Neurodegenerative Diseases (DZNE), Munich, Germany. [9]Comprehensive Cancer Center Freiburg (CCCF), Medical Center, University of Freiburg, Freiburg, Germany. [10]German Cancer Consortium (DKTK) partner site Freiburg, Freiburg, Germany. [11]Department of Neurological Surgery, Lou and Jean Malnati Brain Tumor Institute, Robert H. Lurie Comprehensive Cancer Center, Feinberg School of Medicine, Northwestern University, Chicago, IL, USA. [12]These authors contributed equally: Jan Kueckelhaus, Simon Frerich. ✉e-mail: jan.kueckelhaus@uk-erlangen.de; dieter.henrik.heiland@uniklinik-freiburg.de

identification of key histomorphological niches, that have become crucial factors in contemporary diagnostics and research. In glioblastoma, for instance, necrosis and the border between tumor and healthy tissue are notable examples. Recent advances in spatial transcriptomics have also revealed recurrent patterns of gene expression reflecting responses to inflammatory or metabolic stimuli and different stages of development[4]. The recurring nature of these spatial niches, whether of histomorphological or molecular nature, highlights their significance in understanding these medical conditions. However, to fully comprehend their roles and dynamics within the microenvironment, sophisticated analysis tools for supervised screening approaches are essential. Conventional approaches, such as clustering followed by differential expression analysis (DEA), encounter substantial limitations when applied to spatial multi-omic studies. The binary nature of clustering and its imposition of artificial boundaries can obscure nuanced expression patterns and fail to capture critical features within intricate tissues. Furthermore, the outcomes are reliant on the selected number of clusters, which is influenced by sample characteristics and algorithmic parameters. This reliance presents challenges in data interpretation, particularly given the continuous nature of gene expression in spatial samples. Consequently, clustering with DEA is suboptimal for addressing questions concerning spatial gene expression patterns, especially in complex and disordered tissues like malignancies. To overcome the limitations of DEA, unbiased computational methods like SpatialDE[7] and SPARKX[8] have been developed. While they are effective in identifying genes based on spatial variability, these algorithms primarily offer a holistic view and do not allow to incorporate additional information important to the sample and the specific query. Consequently, they may identify genes with statistically significant spatial expression patterns that are, however, not related to specific areas of the tissue architecture the researcher wants to focus on, Supplementary Fig. 13a–h. In certain scenarios, a more refined approach is necessary, one that can provide insights specifically tailored to the specificities of the tissue sample and the research questions at hand.

In this work we present a flexible, supervised screening approach attuned to detecting spatial subtleties. Furthermore, we aim to capture spatial expression dynamics through gradients rather than group-based log-fold changes, recognizing the inherent continuous nature of expression data in a spatial context[8]. Our efforts have led to the development of two methods falling under the umbrella term spatial gradient screening (SGS). These methods empower users to define spatial locations of interest and use them as reference points while screening for genes and other continuous features with relevant biological meaning. We demonstrate that this dual focus on location-specific screening and spatial gradients seamlessly complements and extends established gene identification approaches in spatial biology, catering to both exploratory and hypothesis-driven inquiries.

## Results

### DEA is unreliable in predicting gene expression with spatial dependencies

To demonstrate the challenges inherent in analyzing gene expression in relation to defined tissue architecture, we analyzed a glioblastoma Visium dataset with three histologically distinct regions: a tumor core, a transition area, and the infiltrative cortex area, Fig. 1a–c. We hypothesized that genes exhibiting a gradual change of gene expression from the core to the infiltrative regions of glioblastoma are inadequately represented by traditional differential gene expression analysis. We compared classical differential gene expression analysis with our Spatial Trajectory Screening to characterize the histological regions based on their gradient and group-based gene expression profiles. To incorporate the histological classification into our data, we utilized SPATA2's interactive annotation tool and labeled each barcoded spot depending on the histological region it was located in. This

resulted in the division of spots into three groups: (1) Tumor, (2) Transition, and (3) Infiltrated Cortex, as depicted in Fig. 1a–c and Supplementary Fig. 1a, b. To validate our histological classification, we inferred copy number alterations (CNA) using SPATA2's implementation of the infercnv R package. This showed an inferred gain of chromosome 7 and loss of chromosome 10 in the malignant region on the left, displayed in Fig. 1f, g, consistent with previous studies[5]. Additionally, the integration of histologically defined areas enabled us to quantify CNA across histological groups. We observed the highest abundance of Chr 7 and 10 alterations in the tumor area, partial alterations in the transition zone, and almost no alterations in the infiltrated cortex (ANOVA, Chr 7 $p < 2.2 \times 10^{-22}$, Chr 10 $p < 2.2 \times 10^{-22}$), Fig. 1h, i, confirming our histological annotation. To further investigate the role of the transition zone we aimed to determine significantly upregulated genes of the annotated regions. The copy number aberration (CNA) profile suggests that the transitional region exhibits a significantly reduced tumor proportion, thereby representing the boundary zone towards the normal cortex. A border-like function of this area was also supported by spatial clustering results using the BayesSpace algorithm[9] which incorporates transcriptional and spatial distance, suggesting two spatially segregated clusters that largely overlapped with our manually annotated transition area, Supplementary Fig. 1c, d. To identify uniquely expressed genes, we performed differential expression analysis (DEA) based on our manual histological annotation. This resulted in the identification of 3489 significant differentially expressed genes (DEGs) using the default thresholds (avg_log2FC > 0.25; p_adj < 0.05). We found 1698 DEGs in the tumor area, 533 DEGs in the transition group, and 1258 DEGs in the infiltrated cortex group. DEA results further supported our categorization of the left area as a tumor region, evident from the presence of marker genes such as EGFR, and the right area as cortex, characterized by neural marker genes like SNAP25. The transition area exhibited elevated expression levels of genes associated with glial cells, such as MBP and MOG. Intriguingly, this region shared 78 differentially expressed genes (DEGs) with the tumor area and 21 genes with the infiltrated cortex, as highlighted in the volcano plot presented in Fig. 1k. Notably, there were no shared DEGs between the tumor area and the infiltrated cortex. Attributing DEA the capacity to predict gene expression in space, we hypothesized that after removing shared DEGs, the remaining ones should be exclusively expressed within the boundaries of the histological area occupied by the group they were assigned to as marker genes. We refer to DEGs that were not shared between two groups as area-specific DEGs. The top 18 of those in terms of significance (lowest adjusted p-values) are displayed in Fig. 1e.

To benchmark whether the remaining area-specific marker genes featured a corresponding spatial expression, we deployed a supervised spatial trajectory to predefine the axis along which gene expression changes are to be examined, Fig. 2a. We used this spatial trajectory to infer the gene expression of the top 18 group specific DEGs gradient along it. For a detailed description of the process involved in obtaining an expression gradient using a spatial trajectory please refer to the method section as well as to Supplementary Figs. 7e–i and 8b. While the expression of some genes along the trajectory clearly reflected the corresponding region, as exemplified in Fig. 2b, c, we found that DEA results were unreliable in determining the precise spatial extent of gene expression, even after filtering for area-specific marker genes, showcased in Fig. 2e–h. Inferring the expression gradient showed that the expression of even the most significant unique DEGs from the tumor and the transition group did not always decline abruptly when crossing the boundary. Unique marker genes from the transition zone featured gradual decreasing patterns transgressing both borders to the cortex- and the tumorous area alike, e.g. genes EEF1A1 and MBP. Furthermore, unique marker genes from the tumorous area featured rather a gradual transgression through the borders to the transition area, while marker genes from the cortex-like area declined rather

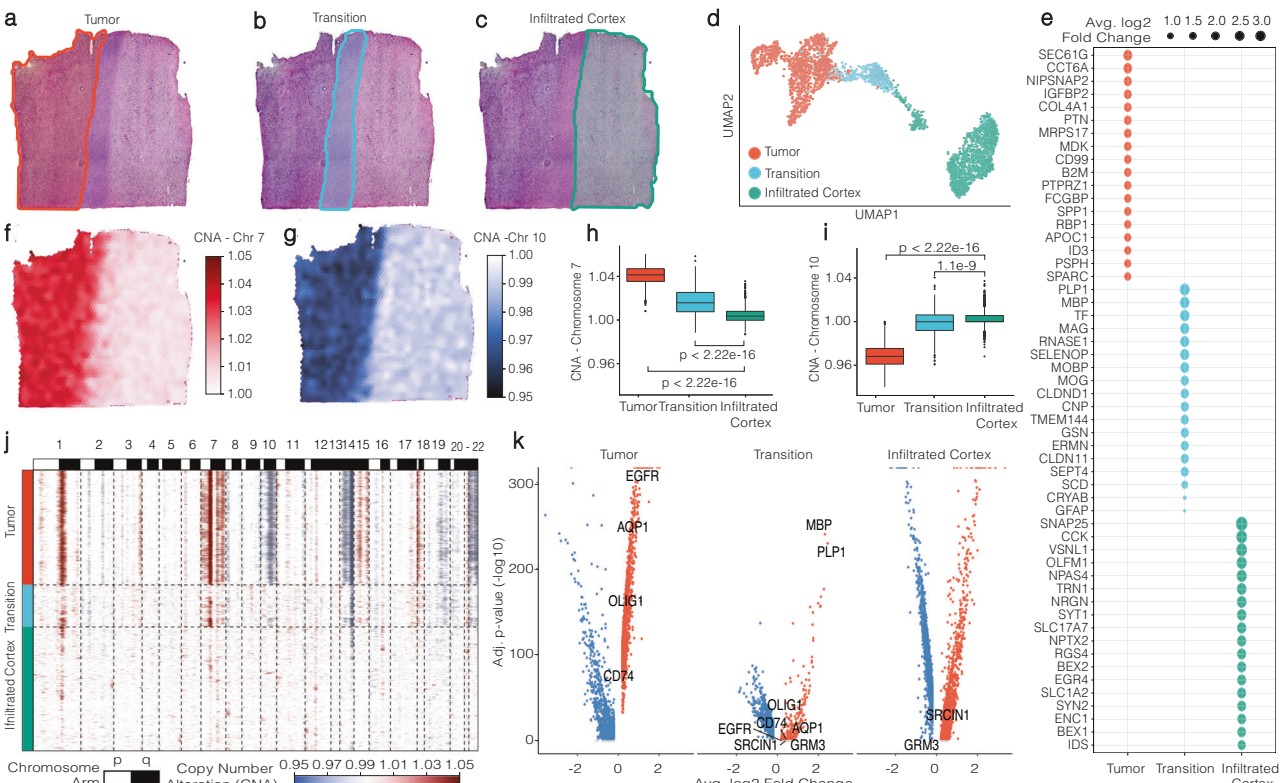

**Fig. 1 | Histologically defined group-based analysis. a–c** SPATA's manual annotation tool is employed to delineate borders, facilitating the grouping of spots based on histology. **d** Gene expression-driven UMAP projection of spots. **e** A dot plot showcases the 18 most statistically significant unique marker genes, ranked by their average log2-fold change, in accordance with histological areas. **f** and **g** Surface plots are color-coded to highlight inferred copy number alterations that are characteristic of glioblastoma. **h** and **i** Statistical analyses examine copy number alterations across histological areas (two-sided test, no adjustments for multiple comparisons, Tumor ($n = 1307$), Transition ($n = 478$), Infiltrated Cortex ($n = 1428$). The minima represent the smallest and the maxima represent the largest value

within 1.5 times the interquartile range (IQR) below or above the first or third quartile (Q1, Q3), respectively. The median is shown as a line inside the box. The box bounds are the first quartile (Q1, 25th percentile) and the third quartile (Q3, 75th percentile). The whiskers extend from Q1 and Q3 to the minima and maxima. **j** A heatmap provides a comprehensive view of alterations across all chromosomes in relation to histological areas, corroborating the tumor area's classification with multiple alterations, the nearly unaltered cortex infiltrated by the tumor, and the transition zone exhibiting intermediate levels of alterations. **k** A volcano plot from the DEA analysis across histological areas highlights marker genes for the Tumor and Transition areas, characterized by an adjusted $p$-value of 0 (infinite –log10).

abruptly right before the passage of the transition zone. Giving DEA the benefit of the doubt, we hypothesized that our potentially biased manual annotation could have affected the results, hindering DEA from reliably identifying genes with gene expression confined to the borders of the marked areas. To this end, we employed DEA based on the grouping suggested by BayesSpace clustering, Supplementary Fig. 2a. Example genes whose gradients do correspond to the area of their clusters are displayed in Supplementary Fig. 2b, c. Still, even among the most significant cluster-specific marker genes (Supplementary Fig. 1e, f), multiple genes did not feature gene expression patterns confined to the area covered by their corresponding cluster, Supplementary Fig. 2d–h. Our spatial trajectory methodology underscores the difficulty in establishing precise boundaries when analyzing spatial transcriptomic data, emphasizing the importance of acknowledging the gradual changes of gene expression in tissue. Furthermore, it reveals notable shortcomings of differential expression analysis in examining spatial expression patterns, thereby highlighting the necessity for analysis approaches that are independent of predefined groups.

### Exploring the local environment of histological microstructures independently of grouping

We showed that capturing gene expression in the form of gradients can be valuable for gene pattern identification related to specific spatial structures, but the applicability of linear trajectories as a

spatial reference is limited to rectangular areas. While they suit a-to-b architectural scenarios like the border between the tumor and healthy tissue, they do not capture tissue patterns related to spatial niches of a circular nature. To exemplify this limitation, we employed a Visium dataset of two mouse brain sections with stab wounds representing traumatic brain injury[10]. Our objective was to incorporate the location and spatial dimensions of these wounds into our spatial gradient screening process. To achieve this, we utilized SPATA2's interactive spatial annotation tools, which facilitate the manual definition of spatial reference areas by directly interacting with the image, Supplementary Fig. 6a. Please refer to the method section for an elaboration on the differences between manual annotation of data points, as conducted for the three histological groups in Fig. 1a, and spatial annotations. Figure 3a illustrates the resulting stab wound outlines. Given the initial processing, which unveiled a noticeable cluster around the stab wounds perturbing the otherwise healthy CNS architecture of the mouse cortex, Fig. 3b, we assumed that these injuries likely exerted a substantial impact on their immediate surroundings. To explore gradual changes of gene expression, likely inflicted by the stabwounds on their surroundings, we conducted spatial gradient screening, inferring gene expression gradients as a function of distance to each stab wound, Fig. 3c, d, and screened for multiple patterns, Supplementary Fig. 14d. This approach allowed us to identify genes with descending expression patterns, reflecting their fine-grained biological

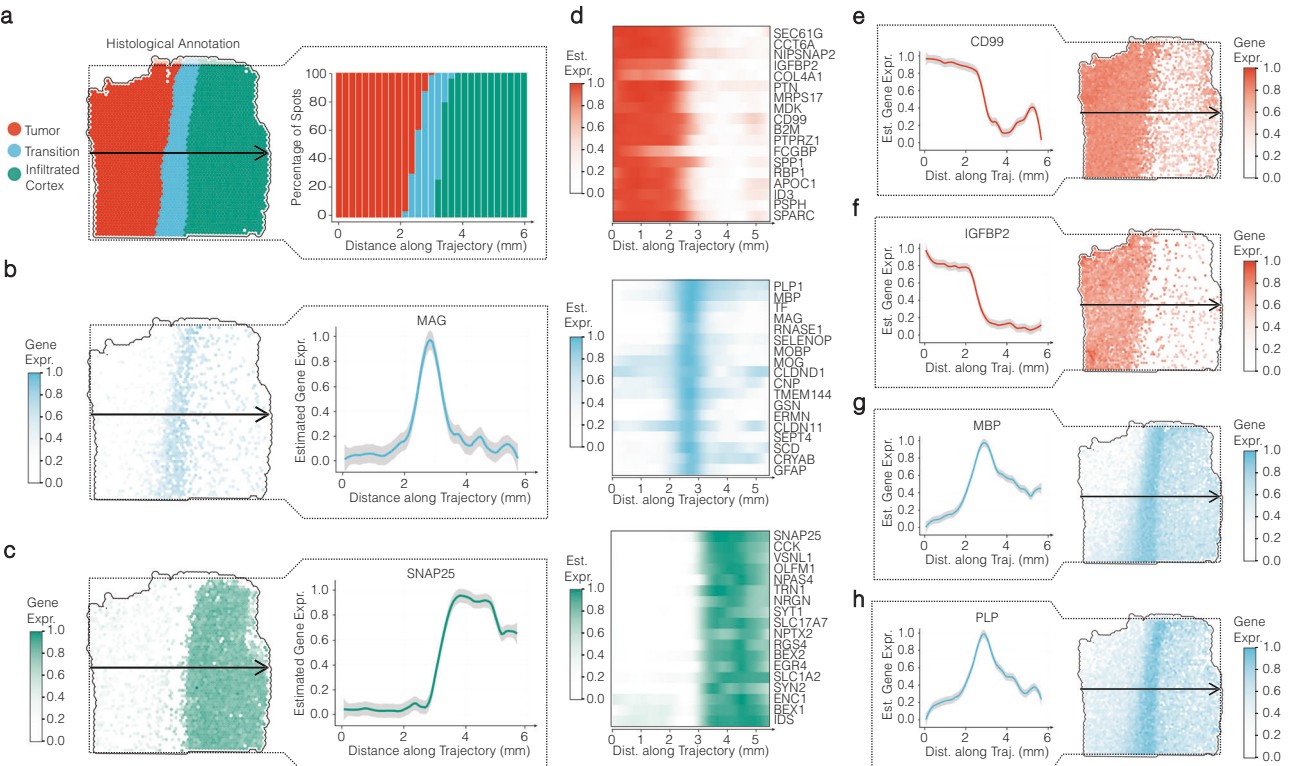

**Fig. 2 | Spatial expression analysis of genes identified by differential expression analysis (DEA) along a linear trajectory. a** A surface plot provides a visual representation of the trajectory's course (indicated by an arrow), complemented by a barplot illustrating the proportion of histological groups along this trajectory. **b** and **c** Surface plots offer insights into the gene expression patterns of genes predominantly localized within their respective assigned areas. Error bands of line plots indicate the confidence interval (level: 0.95). **d–g** Heatmaps present the expression profiles of the top 18 differentially expressed genes (DEGs) per histological group (as referenced in Fig. 1e). Notably, while these genes have been identified as unique DEGs, those from the tumor and transition regions do not consistently exhibit confined expression patterns to their designated areas, in contrast to DEGs from the (infiltrated) cortex. **e–h** Line plots and surface plots further elucidate the expression patterns of selected genes, demonstrating how they may traverse area boundaries in various ways. Error bands of line plots indicate the confidence interval (level: 0.95).

association with the injury. Figure 3c–f visualizes inferred gradients of genes up to a distance of 1.5 mm from the injury border. Notably, several genes associated with immune activity (C1qa, C1qb, B2m), immune cell types (Cd68, Aif1), migration (Vim, Tyrobp), proliferation (Cd64, Ccnd1), and wound healing (S100a16, Lamp2) exhibited increased expression closer to the injury, Fig. 3g–j and Supplementary Fig. 3b–e, that faded with increasing distance. Subsequent screening within a shorter distance of 0.75 mm reaffirmed these findings and revealed additional genes that suggest increased immune activity, migration, and proliferation near the injury zone (Jun, Manf, Camk1, Ier5l) displayed in Supplementary Fig. 3f, g. It is worth noting that the genes depicted in Supplementary Fig. 3f, g were not identified as marker genes through DEA we conducted on the clustering (Supplementary Fig. 3a). This observation underscores the capability of spatial gradient screening to detect even the most subtle and intricate expression patterns. Lastly, the estimation of cell density, obtained through single-cell deconvolution via Tangram[11], highlighted increased microglia and macrophage density near the injury site decreasing with increasing distance, as suggested by the genes identified by spatial annotation screening, Fig. 3k–m. In summary, our study demonstrates that spatial annotation screening can adapt to complex spatial reference features and infer gene expression gradients related to them. Our findings with spatial annotation screening align with existing knowledge of the central nervous system's response to injury, highlighting its capabilities to identify spatial gene expression patterns related to spatial areas and paving the way for new insights.

## Identification of confounders of gene expression in spatial transcriptomic studies

Glioblastoma presents a unique challenge in spatial transcriptomics due to its heterogeneous and chaotic nature, making it difficult for clustering algorithms to discern optimal cluster numbers and identify marker genes and spatial niches. This complexity sharply contrasts with the well-structured architecture of its healthy counterpart, the human neocortex, which demonstrated clearly defined spatially segregated layers that can be reliably derived from clustering, as illustrated in Supplementary Figs. 4, 5, 14a–c. One of our examined samples, UKF313T, exemplified this challenge. Initially, we observed a prominent central necrotic region, manually outlined as the "necrotic center" using SPATA2's image annotation tool. Clustering the sample with BayesSpace, Supplementary Fig. 4d, revealed the absence of a clear elbow point in the clustering assessment, Supplementary Fig. 4b, highlighting the inherent challenges in defining clusters and borders in such samples. Annotating the central necrotic area alongside the clustering results, we noted that clusters proximal to the necrotic outline either coincided spatially with necrosis (B1) or organized themselves in a circular fashion around the necrotic center (B2, B3), akin to the injury cluster observed around the stab wound in Fig. 3b. However, when we computed top marker genes of each BayesSpace cluster in space, Supplementary Fig. 4d, marker gene expression did not align with the boundaries of their initially assigned clusters, emphasizing the unsuitability of this sample for traditional cluster analysis. This chaotic gene expression pattern was further evidenced by many

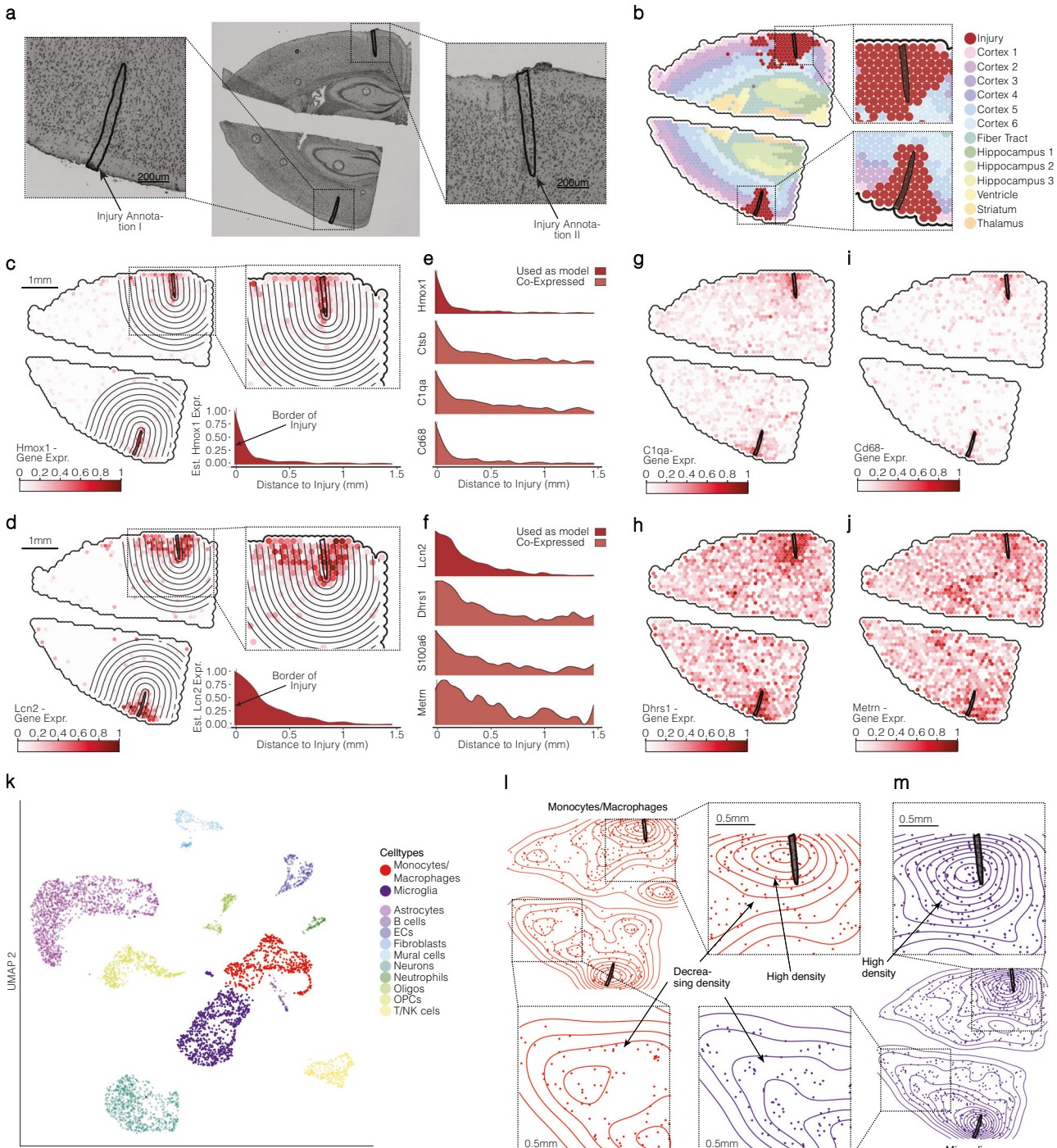

**Fig. 3 | Integration of visium mouse brain dataset for gene co-expression analysis and eq data examination. a** An H&E image provides an overview of the analyzed sample, with two enlarged windows highlighting the stab wounds inflicted on the brain (*n* = 1 mouse with bilateral injury for Visium, *n* = 3 mice for scRNA-seq experiments). **b** Surface plots reveal clustering results obtained through the Scanpy pipeline, emphasizing significant barcode spot clusters within the injury area. **c** and **d** Surface plots and gradient ridge plots illustrate the gene expression patterns of two marker genes for the injury area, Hmox1 and Lcn2. **e** and **f** Ridgeplots visualize the expression gradients of genes sharing similar patterns with either of the two example genes, as identified through spatial annotation screening. **g**–**j** Surface plots showcase the expression profiles of co-expressed genes identified via spatial annotation screening. **k** UMAP of the scRNA-seq dataset, from the same mouse model as the Visium sample. **l** and **m** Visualization of Tangram results, featuring 2D Density plots that highlight an elevated density of monocytes, macrophages, and microglia in the vicinity of the injury zone, as compared to a control area.

marker genes sharing nearly identical average log2 fold changes across multiple clusters, Supplementary Fig. 4c. Assuming a descending relationship between marker genes of clusters B2 and B3 and the necrotic area, similar to the relationship between the stab wound and its environment, we computed the gene expression of marker genes relative to the distance from the necrotic center. The inferred gradients of CD44, NDRG1, THBS2, and IGFBP3 (predicted by DEA to be strongly expressed in the circular area represented by cluster B4) led to the assumption that a spatial dependency between the expression of these genes and the presence of necrosis exists. We

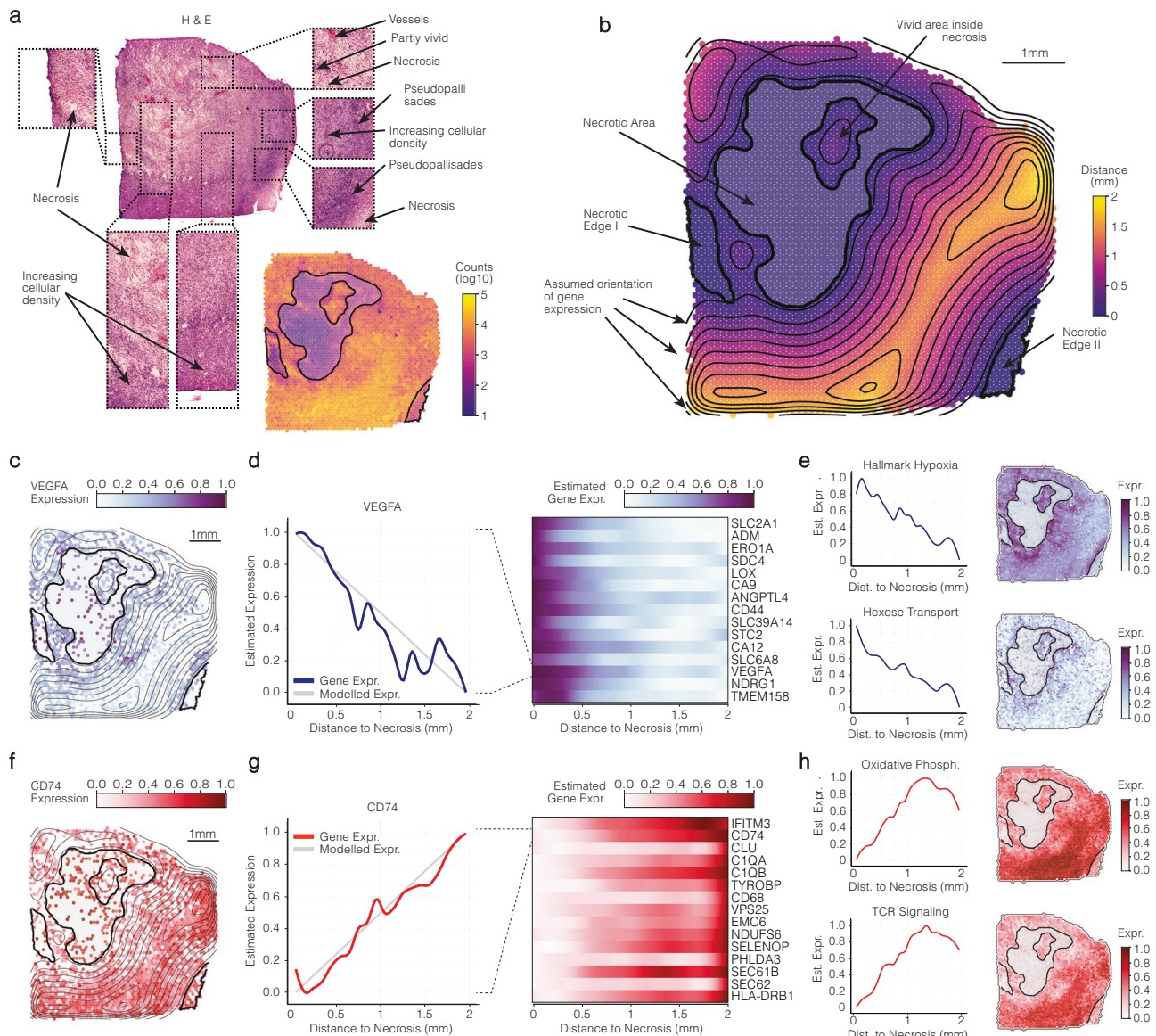

**Fig. 4 | Comprehensive annotation of histology and gene expression relative to necrosis in glioblastoma. a** Presents the H&E image of the glioblastoma sample UKF313T, emphasizing key areas within the sample that are detailed in the following figure (**b**). Additionally, a surface plot visualizes the count distribution, supporting our necrosis (dead tissue) annotation. **b** Provides a visualization of distance values from the necrotic areas, with lines illustrating the assumed gradient direction and

orientation based on proximity to necrotic regions. **c–e** Showcase representative genes exhibiting a pattern reminiscent of association with necrosis, displaying decreasing expression levels with increasing distance from necrotic regions. **f–h** Feature representative genes showing a pattern resembling the recovery of expression levels with increasing distance from necrotic areas.

also observed an inverse relationship in genes from clusters distant to necrosis, suggesting their repulsion by the presence of necrosis. This observation was supported by the decline of their expression at ~2–3 mm, Supplementary Fig. 4d, coinciding with the influence zone of another necrotic area at the bottom right of the sample, which was initially not accounted for in our single necrotic center annotation. To accurately account for necrotic areas in this sample, we introduced two additional annotations, referred to as "necrotic edge I" and "necrotic edge II", Fig. 4a, b. Given the insights acquired from the gradients displayed in Supplementary Fig. 4d and the significant role necrosis plays as a histomorphological correlate of malignancies, we hypothesized that necrosis significantly confounded gene expression in the sample. Subsequently, we conducted comprehensive spatial gradient screening, considering the spatial extent of all three necrotic annotations, Fig. 4b. We employed SPARKX to preselect genes predicted to exhibit spatial significance, resulting in a set of

genes ($n = 11,478$, adj. $p$-value < 0.05) that we subsequently subjected to the screening. We focused on subsets of descending and ascending models, Supplementary Fig. 14d, to specifically identify genes associated with necrosis and those repelled by it. Prominent examples of necrosis-associated genes included those involved in hypoxia response (VEGFA), glycolytic metabolism (SLC2A1), and cell cycle arrest (TMEM158), illustrated in Fig. 4c, d. Conversely, genes repelled by necrosis were linked to oxygen-dependent metabolism (NDUFS) and TCR receptor signaling (CD74, HLA-DRB1), displayed in Fig. 4e, f. In summary, our analysis involving spatial gradient screening using spatial annotations illuminated intricate spatial patterns within glioblastoma, shedding light on the association and repulsion of specific genes in response to necrosis. In particular, it shed light on a potential spatial dependency between necrosis, hypoxia and TCR immune response, which we aimed to investigate closely in the following section.

## Elucidating the spatial dynamics of T cell abundance and hypoxic metabolism in glioblastoma

Building upon prior findings that T cell receptor (TCR) signaling diminishes in the immediate vicinity of necrotic regions but steadily intensifies within one to two millimeters from these necrotic areas in glioblastomas, we hypothesized that the hypoxic metabolic environment found surrounding necrotic regions significantly impacts the immune landscape within these tumors. We examined the distribution of immune cells from hypoxic to non-hypoxic areas by combining spatial annotations from six glioblastomas with identified hypoxic niches. Cell abundance was determined using the cell2location algorithm, and CytoSpace was employed to enhance resolution from spot to single-cell level. Further, we integrated spatially resolved T cell sequencing[12] (SPTCR-seq) to support our hypothesis and understand the distribution of clonal and non-clonal T cells dependent on the presence of hypoxia, Fig. 5a. We began by characterizing hypoxia niches in six glioblastoma samples based on gene expression, Fig. 5b, c and Supplementary Fig. 6c. Horizontal integration of each samples results were facilitated by SPATA2s incorporation of SI units. We then assessed the distribution of cell types from hypoxic to non-hypoxic areas, aggregating data by average cell abundance. Our analysis showed that gene expression related to hypoxia normalized beyond ~1000 μm from hypoxic regions (elbow at 954.3 μm), Fig. 5d. By focusing on lymphoid cells, we found that T cells peak in abundance around 1000 μm from hypoxic cores and are present up to 1500 μm (spanning from 500 to 2000 μm). Beyond 2000 μm, T cell presence diminishes, Fig. 5e. Similarly, bone-derived myeloid cells, abundant in hypoxic areas, decrease in number beyond 2000 μm, mirroring the T cell distribution, Fig. 5f. This immune response pattern relative to distance from hypoxia was also evident in the varying abundance of mesenchymal-like (higher towards hypoxia) and NPC-like (high towards infiltration regions) malignant cell populations Figures 5f, g. Integration of T cell receptor sequencing (SPTCR-seq) data confirmed the estimated T cell abundance from cell type deconvolution, Fig. 5h, i. We then focused on the distribution of exhausted and cytotoxic CD8 T cells based on their proximity to hypoxic areas. By analyzing gene expression markers for cytotoxic (e.g., GZMA, GZMB) and exhausted (e.g., PDCD1, LAG3) CD8 T cells, we found a higher concentration of exhausted CD8 T cells near hypoxic regions, whereas cytotoxic gene expression was more prevalent in T cells further from these areas, Fig. 5j. In summary, we demonstrated the utility of annotation screening in elucidating the immune response architecture in relation to hypoxic metabolism in glioblastoma. Our findings suggest a robust spatial dependency between hypoxia and T-cell abundance shared across six glioblastoma samples contributing to the quest of understanding the intricate biological architecture of glioblastoma.

## Statistical challenges and benchmarking

Our endeavors to identify gene expression patterns related to spatial reference areas or trajectories introduce a statistical challenge: accurately identifying patterns amid the noise and variability inherent to biological data. To tackle this challenge, we employ LOESS smoothing to model gene expression data along the distance, Supplementary Fig. 8a, b, and quantify the degree of randomness in the emerging patterns using the total variation, Supplementary Fig. 8c. We validated our approach through extensive simulations wherein controlled noise levels were introduced into predefined spatial patterns. This allowed us to evaluate the efficacy of our method in distinguishing between these patterns amidst various noise types and noise intensities, Supplementary Figs. 9–11. In addition, we compared our method to two existing approaches designed for scRNA-seq pseudotime trajectories, namely tradeSeq[13] and PseudotimeDE[14,15]. SPATA2's SAS and the said algorithms share a common goal of detecting differential expression patterns along a one-dimensional axis. In our simulated scenario, SAS exhibited a higher correlation with ground truth noise, achieving a

mean $R^2$ of 0.75 across different pre-defined patterns, in contrast to the $R^2$ of 0.59 for tradeSeq's Wald Statistic and an $R^2$ of 0.64 for PseudotimeDE's test statistic, Fig. 6a–c. Moreover, our benchmark illustrates that SPATA2's SAS can rapidly screen 10,000 genes within a few minutes on a personal laptop (32GB RAM, 10 cores), Supplementary Fig. 14e. This is at least an order of magnitude faster than tradeSeq and PseudotimeDE model fitting alone (run on 256GB RAM, 48 cores; see the "Methods" section). This emphasizes the substantial computational efficiency and applicability of our method for large-scale datasets.

Lastly, while the spatial reference features—spatial annotations or spatial trajectories—can be placed in an automated manner, Supplementary Fig. 6b, c, we explored the resilience of our method against human error in case of manual placement. We evaluated the implications for spatial annotation screening (SAS) and spatial trajectory screening (STS) test performance after systematically modifying the positions or orientations of selected spatial annotations and trajectories in our dataset, Supplementary Figs. 6d–g and 12. Our simulations indicate that both SAS and STS can be susceptible to type II errors with increasing deviation from the original annotation or trajectory placement. However, this susceptibility remained well-controlled within the expected bounds of human error, and notably, type I error rates stayed below 5% across all deviations, ensuring that random patterns are rarely incorrectly identified as non-random, even with large deviations in annotation. In conclusion, our thorough simulations and benchmarking highlight the robustness, adaptability, and computational efficiency of our approach for identifying spatial gene expression patterns along a one-dimensional gradient. This reaffirms our method as an advantageous application for hypothesis-driven spatial biology research.

## Discussion

Despite the transformative potential of spatial transcriptomics, the field has encountered limitations with existing analytical methods, which often do not fully capture the continuous and intricate patterns of gene expression across diverse tissues. To address these challenges, we introduce spatial gradient screening (SGS), an algorithm designed to capture gene expression patterns along a spatial continuum. It pursues the hypothesis that specific genes—or other numeric features for that matter—display non-random expression patterns in relation to spatial reference features. When screening for spatially variable genes, we utilize these reference features to incorporate the integration of potential biological forces, such as the direction of tumorous infiltration using spatial trajectories (spatial trajectory screening, STS), Fig. 2a, or the presence of stab wounds and necrotic areas using spatial annotations (spatial annotation screening, SAS) (Figs. 3a, b and 4a, b). STS captures changes along a-to-b trajectories, while SAS focuses on radial changes from core to periphery. Together they facilitate a comprehensive framework for both exploratory and hypothesis-driven analyses. A particularly notable finding from our work is the identification of a spatial interplay between necrotic and hypoxic zones and T-cell distribution within glioblastomas, hinting at a potentially stratified immune landscape within these tumors.

Thorough benchmarking of the evaluation metrics showed that spatial gradient screening reliably differentiates between existing patterns and randomness. However, despite their utility, both spatial trajectory screening and spatial annotation screening have limitations. The flexibility they offer necessitates a close examination of histological architecture and other potential confounding factors. Interactive placement using SPATA2's tools can introduce human error, impacting the course of a spatial trajectory and the outline of a spatial annotation and, thus the inferred expression gradients. We conducted a thorough investigation into the sensitivity of spatial gradient screening to such variations and our findings demonstrate that both spatial gradient screening approaches yield robust results against an expected degree

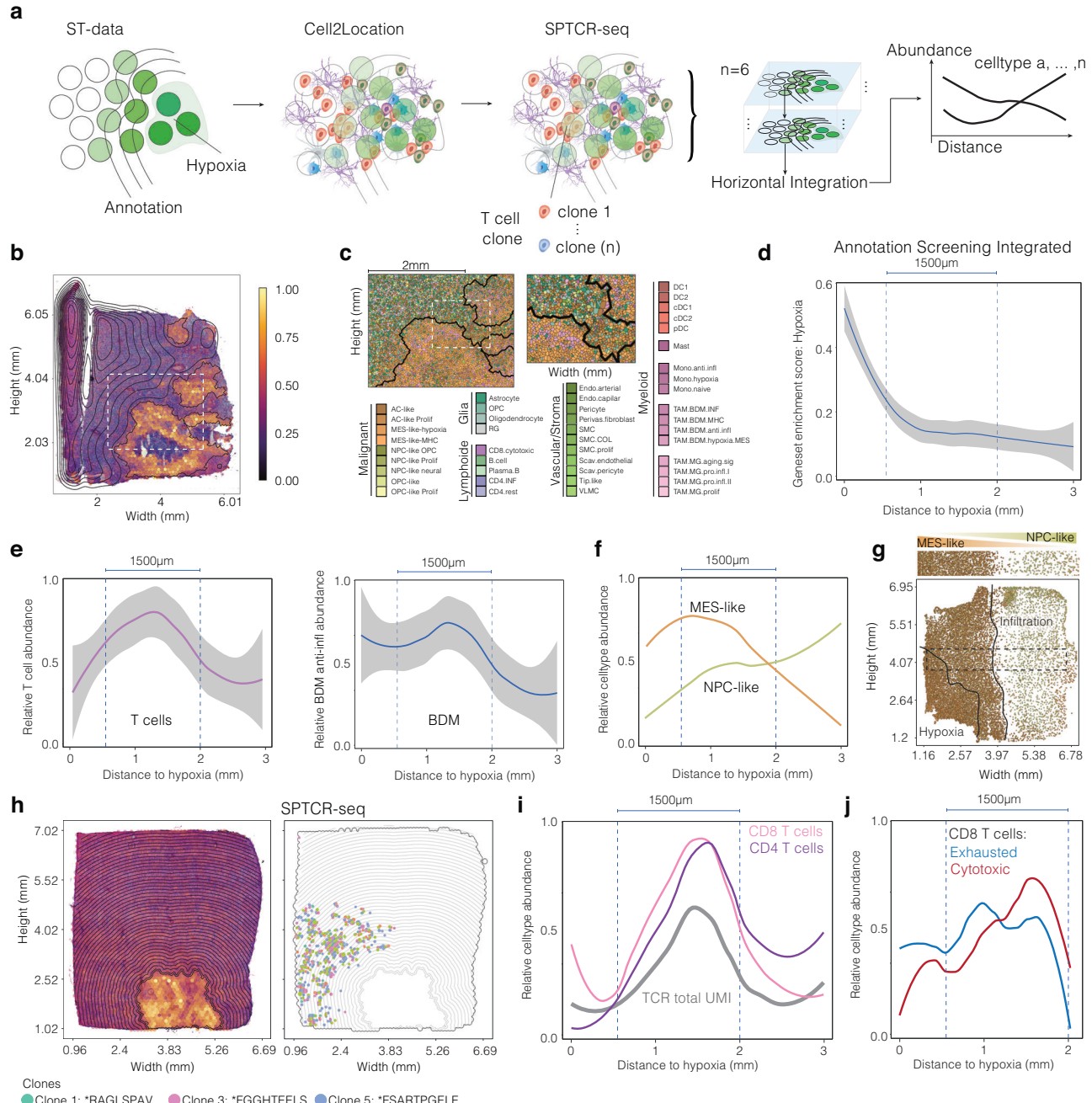

**Fig. 5 | Comprehensive and integrated analysis of spatial relationships between hypoxia and T-cell abundance in glioblastoma. a** Overview of the workflow, encompassing spatial transcriptomic (ST) data, single-cell deconvolution, and spatial T-cell receptor sequencing (SPTCR-seq), showcasing the horizontal integration of six glioblastoma samples featuring prominent hypoxic spatial niches. **b** A representative Visium glioblastoma sample (UKF260T) with multiple hypoxic areas, annotated using SPATA2's automatic annotation tool. Lines indicate the screening direction guided by the hypoxic gradient. **c** Presentation of single-cell deconvolution results for sample UKF260T, highlighting proximity to annotated hypoxic regions. **d** The inferred gradient of hypoxic gene signatures merged from data obtained from six samples. Error bands of line plots indicate the confidence interval

(level: 0.95) **e** and **f** Evaluation of T-cell and anti-inflammatory bone-derived macrophage abundance as a function of distance from the hypoxic areas. Error bands of line plots indicate the confidence interval (level: 0.95). **f** Abundance of cell types described by Neftel et al. (2019), revealing distinct abundance patterns. **g** Visualization of MES-like and NPC-like cell abundance in sample UKF269T. **h** An illustration of a glioblastoma example integrated into the comprehensive screening, featuring a solitary hypoxic area and a separate area of T-cell abundance approximately 1 mm distant from the hypoxic region. **i** Gradient representation of T-cell abundance as a function of distance from hypoxia. **j** Gradient plot depicting various T-cell subtypes as a function of distance from hypoxia, revealing an inverse correlation between T-cell cytotoxicity and distance to hypoxia, peaking at ~1 mm.

of human-induced variation. Still, for precise outcomes, careful placement of spatial reference features is essential. The inclusion of confounding elements or the omission of crucial ones can significantly affect results, potentially leading to the misinterpretation of gene expression.

Our findings highlight a critical limitation of traditional DEA in capturing genes that exhibit gradual expression shifts in response to microstructural changes or metabolic gradients within complex tissues. It is important to note that, while we have identified these limitations, our intention is not to diminish the significant contributions

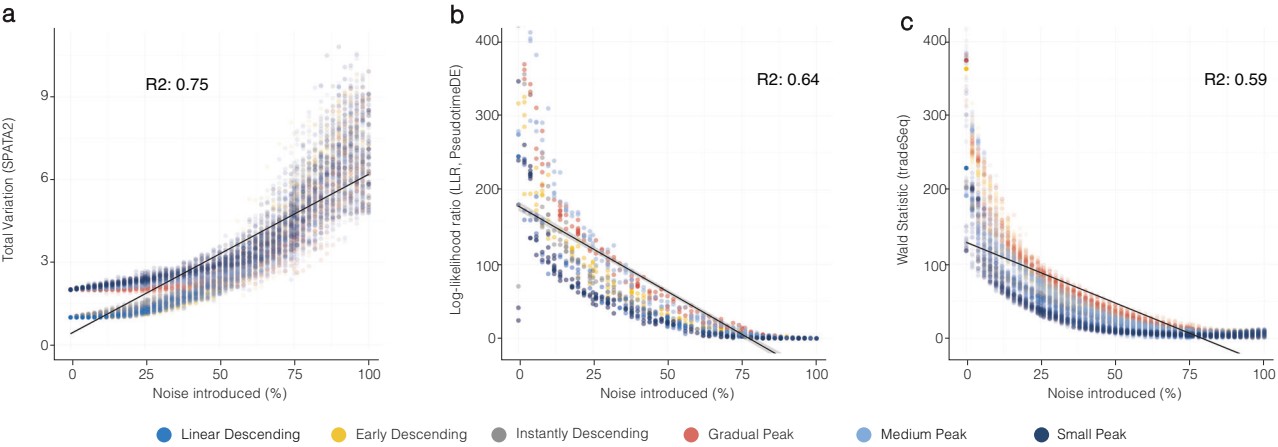

**Fig. 6 | Statistical comparison of the test statistic from SPATA2's spatial gradient screening with those of analogous methods PseudotimeDE and tradeSeq. a–c** Scatterplots display the relationship between different test statistics used by different algorithms to determine the noise ratio of simulations. Colors indicate the type of simulated pattern hidden by noise (also see Supplementary Figs. 9b, 10b).

of these traditional analysis methods. Both clustering-based and manual annotation-based approaches demonstrated their value in numerous spatial transcriptomics studies and will continue to do so. However, for a nuanced analysis of spatial expression patterns and gradients, it is evident that integrating supplementary tools is essential to achieve comprehensive interpretation. Spatial gradient screening represents our contribution to this evolving field, augmenting classical methodologies with enhanced capabilities to interpret complex spatial gene expression data. Although here we focused on brain-derived disease models, spatial gradient screening is a helpful tool in a broad variety of tissues in health and disease conditions in which spatial dependencies and heterogeneity are important players. To streamline the integration of spatial gradient screening into downstream analysis, we have incorporated it into SPATA2, an R-based framework that provides a user-friendly implementation of well-established analytical techniques. This integration encompasses the identification of spatially variable genes, facilitated by SPARKX, interactive manual annotation of barcode spots, and a suite of algorithms for tasks such as inferring copy number alterations, clustering, conducting differential expression analysis, and performing gene set enrichment analysis.

Furthermore, with regard to spatial analysis, SPATA2 fully incorporates the utilization of distance measures in SI units. This feature greatly enhances usability, allowing for the seamless integration of multiple samples. Additionally, it ensures that results remain scalable to different image resolutions, offering an intuitive framework for analysis, interpretation, and visualization. Moreover, while SPATA2's default pre-defined models, as employed in this study to guide further interpretation of identified genes, cover a substantial portion of biologically relevant patterns, Supplementary Fig. 14d, specific research scenarios may necessitate tailored models. SPATA2 offers additional functions to expand the range of models available for screening. Lastly, it's worth noting that while SPATA2 was initially developed with the 10X Visium platform in mind, it has been extended to support virtually all spatial platforms, regardless of the modality and observational unit (e.g., barcoded spots, single cells, beads). SPATA2 offers various functions to ensure compatibility with platforms established in recent years, such as Seurat[16,17], Giotto[18], Scanpy[19], and Squidpy[20]. To assist users in adopting SPATA2, we offer user-friendly tutorials on our website. All in all, we believe that SPATA2 and the spatial gradient screening approach will be a valuable tool in the analysis of an exciting and rapidly developing field in spatial biology, that is spatial transcriptomics.

## Methods

### Ethical statement
The study design, data evaluation, and imaging procedures were given clearance by the ethics committee at the University of Freiburg, as delineated in protocols 100020/09 and 472/15_160880. All methodologies were executed in compliance with the guidelines approved by the committee. Informed consent, in written form, was received from all participating subjects. The Department of Neurosurgery of the Medical Center at the University of Freiburg, Germany, was responsible for securing preoperative informed consent from all patients participating in the study. Mice were housed and handled under the German and European guidelines for the use of animals for research purposes. Experiments that included mice were approved by the institutional animal care committee and the government of Upper Bavaria (ROB-55.2-2532.Vet_02-20-158).

### The R-package SPATA2
SPATA2 is an R package that offers an object-oriented programming framework centered around an S4 object, named spata2. This object serves as a container for raw expression data, processed data, and the results of various downstream analyses, such as inferring copy number alterations, clustering using multiple algorithms, differential expression analysis (DEA), gene set enrichment analysis (GSEA), and identification of spatially variable genes and histological microstructures in image analysis. The package's name is an acronym derived from Spatial Transcriptomic Analysis, highlighting its focus on spatial transcriptomics, specifically the 10X Visium platform. However, the standard analysis pipelines for clustering, DEA, GSEA, etc., can be applied to any type of expression data, including single-cell sequencing. Note that extensive tutorials can be found at our SPATA2 website: https://themilolab.github.io/SPATA2/.

### Architecture of the S4 SPATA2 object
At the core of the SPATA2 package is the S4 SPATA2 object, which serves as a container for both data and analysis progress. the package to be compatible with a wide range of spatial biology platforms provided that the data structure adheres to the following criteria: First, the numeric variables under analysis should correspond to molecule counts, such as RNA read counts, metabolite counts, or protein counts. Second, a clearly defined observational unit must exist to which the numeric variables can be mapped. For example, the Visium platform's observational unit consists of barcoded spots, while the SlideSeq platform utilizes barcoded beads as its observational unit. In the case

of the Xenium- or the MERFISH platform, the observational unit is the individual cell. Given the different observational units possible we refer to them with the umbrella term data points throughout this manuscript. Third, the observations must be equipped with *x*- and *y*-coordinates for analysis in two-dimensional space and should be equally distributed over the analyzed tissue. We provide helping functions to initiate analysis via SPATA2 for the standardized data output of the following platforms:

- MERFISH: `initiateSpataObjectMERFISH()`
- SlideSeq: `initiateSpataObjectSlideSeq()`
- Visium: `initiateSpataObjectVisium()`
- Xenium: `initiateSpataObjectXenium()`

Lastly, the flexible function `initiateSpataObject()`, allows the user to initiate analysis from the output data of any other platform, provided that it adheres to the aforementioned requirements. Information around the platform used is stored inside the created SPATA2 object and might decide on which features of the package can be used. (E.g. BayesSpace clustering is only compatible with the ST or Visium technique.) The Spatial Gradient Screening algorithms are compatible with *all* platforms.

## Naming convention and families of functions

Most functions of the SPATA2 package start with a verb which indicates the family to which the function belongs. The most important families are: add*(): Add additional content ranging from grouping variables from external clustering algorithms (e.g. `addFeatures()`) or manually set up trajectories (e.g. `addSpatialTrajectory()`). create*(): Add additional content by creating, which either implies the necessity for interaction or additional computation is done b. (e.g. `createSpatialSegmentation()` to create a grouping variable based on manually encircling regions and labeling the barcode spots that fall into the circle, `createImageAnnotations()` to annotate the regions and microstructures on the histological image.) get*(): Extract results in the form of data.frames, lists, or vectors. (e.g. `getCoordsDf()`, `getDeaResultsDf()`, `getSpatAnnBorderDf()`) ggpLayer*(): Create additional layers of miscellaneous aspects that can be added to corresponding plots of the ggplot2 framework via the + operator. (e.g. `ggpLayerSpatAnnOutline()` to add the border of previously annotated microstructures and regions to a surface plot, `ggpLayerScaleBar()` to add a scale bar for visual indication of the physical distance in SI units to a surface plot.) plot*(): Plots results. Usually by outputting objects of class gg from the ggplot2 package. (e.g. `plotSurface()` to visualize barcode spots that fall on the underlying tissue, usually colored by a variable.) run*(): Runs algorithms that are implemented from external packages (e.g. `runDEA()` to run differential expression analysis using `Seurat::FindAllMarkers()`, `runCNV()` to run the pipeline of the infercnv-package). The results are stored inside the SPATA2 object and can be extracted with corresponding get*() functions. set*(): Set miscellaneous content. Recommended if programming with SPATA2 to prevent bugs in case of changing architecture of the S4 object.

## Implemented packages and algorithms

SPATA2 implements a variety of external algorithms and presents them in user-friendly wrapper functions that. The architecture of the SPATA2 object allows to conveniently store the results in the SPATA2 object to extract them via the corresponding `get*()` functions (e.g. `getDeaResultsDf()`) and to plot them via corresponding plot*() functions (e.g. `plotDeaVolcano()`).

## Dimensionality reduction

Dimensional reduction is implemented in two ways. If a SPATA2 object is created using the Seurat pipeline, the embedding of the dimensional reduction (PCA, TSNE, and UMAP) is inherited from the Seurat object.

If not, the dimensional reduction can be conducted from within SPATA2 using `runPCA()` which implements `irlba::prcomp_irlba()`, `runTSNE()` which implements `tsne::tsne()` and `runUMAP()` which implements `umap::umap()`. The embedding of each dimensional reduction currently stored in the SPATA2 object can be extracted via `getPcaDf()`, `getTsneDf()` or `getUmapDf()` and plotted via `plotPCA()`, `plotTSNE()` and `plotUMAP()`.

## Spatial clustering

BayesSpace clustering is implemented as a wrapper of the pipeline suggested by the R-package BayesSpace. The corresponding function in SPATA2 is called `runBayesSpaceClustering()`. This function is a wrapper around all functions needed to obtain cluster results based on the BayesSpace algorithm, including `BayesSpace::readVisium()` or alternatively `asSingleCellExperiment()`, `BayesSpace::spatialPreprocess()`, `BayesSpace::qTune()`, `BayesSpace::spatialCluster()`. The resulting grouping variable is stored together with all other variables that do not refer to the expression of single genes in the feature data.frame of the SPATA2 object. The resulting grouping can be obtained via `getFeatureDf()` and can be used for downstream analysis by referring to the name of the grouping variable (chosen by the user while calling `runBayesSpaceClustering()`) in the recurring arguments across and grouping_variable as in `'my_spata_obj <- runDEA(object = my_spata_obj, across = 'bspace_7')'`.

## Differential expression analysis (DEA)

Differential expression analysis (DEA) is implemented via the function `runDEA()` relies on `Seurat::FindAllMarkers()`. A temporary Seurat object is created using the counts matrix of the SPATA2 object. The Seurat object is processed according to the specifications of the user. Then, the grouping variable based on which the testing is supposed to be conducted is transferred to the meta.data of the Seurat object and to the slots @active.idents. Then the function `Seurat::FindAllMarkers()` is called, which outputs a data.frame, that is stored in the SPATA2 object. The SPATA2 object contains a slot @dea where DEA results are stored according to the grouping variable they base on as well as the method with which the testing is run (defaults to the default of Seurat which is Wilcoxon Sum Rank testing). Using arguments across to specify the grouping variable and methode_de to specify the method, results can be extracted via, e.g. `getDeaDf()` or `getDeaGenes()` or they can be plotted via. e.g. `plotDeaVolcano()`, `plotDeaHeatmap()` or `plotDeaDotplot()`.

## Gene set enrichment analysis (GSEA)

Gene signature enrichment analysis (GSEA) is implemented using the hypeR package[21] which conducts GSEA using hypergeometric testing. GSEA is conducted using the stored DEA results. Therefore, `runDEA()` with a specific combination of grouping variables and DEA method has to be called beforehand. GSEA results are stored in slot @dea next to the corresponding dea results. Results can be extracted via `getGseaDf()` or can be plotted via `plotGseaDotPlot()`. Extensive tutorials about GSEA in SPATA2 can be found here: https://themilolab.github.io/SPATA2 Gene sets can be used for gene set enrichment analysis and visualization and are stored in an extra R-object, a data.frame called gsdf. The SPATA2 object carries it in slot @used_genesets. The gene set collection can be expanded by user-defined gene signatures using the function `addGeneSets()`. The gene set data.frame contains a collection of more than eleven thousand gene sets downloaded from https://www.gsea-msigdb.org/gsea/index.jsp. This includes gene ontology genesets for biological processes (prefixed with BP.GO), cellular components (prefixed with CC.GO), molecular functions (prefixed with MF.GO), as well as hallmark gene sets (prefixed with HM), biocarta gene sets (prefixed with BC) and reactome gene sets (prefixed with RCTM).

## Inferring copy number alterations (CNA)

Inferring copy number alterations (CNA) is implemented as a wrapper of the pipeline suggested by the R-package infercnv. The corresponding function in SPATA2 is called `runCNV()`, which is a wrapper around all functions needed to infer copy number variations. This includes: `infercnv::CreateInfercnvObject()`, `infercnv::require_above_min_mean_expr_cutoff()`, `infercnv::require_above_min_cells_ref()`, `infercnv::normalize_counts_by_seq_depth()`, `infercnv::anscombe_transform()`, `infercnv::log2xplus1()`, `infercnv::apply_max_threshold_bounds()`, `infercnv::smooth_by_chromosome()`, `infercnv::center_cell_expr_across_chromosome()`, `infercnv::subtract_ref_expr_from_obs()`, `infercnv::invert_log2()`, `infercnv::clear_noise_via_ref_mean_sd()`, `infercnv::remove_outliers_norm()`, `infercnv::define_signif_tumor_subclusters()`, and infercnv::plot_cnv(). The output of infercnv::plot_cnv() is stored under a specified directory from which the results are read back into the R session and stored together with metadata regarding the analysis in slot @cnv of the SPATA2 object. Results can be extracted via `getCnvResults()` and visualized via `plotCnvLineplot()` or `plotCnvHeatmap()` as well as with common plotting functions like, e.g. `plotSurface(…, color_by = 'Chr7')` or `plotBoxplot(…, variables = 'Chr7', across = 'histology')`.

## Spatially variable genes

Identification of spatially variable genes using SPARKX is implemented as a wrapper around the function SPARK::`sparkx()` from the SPARK package. The corresponding function in SPATA2 is called `runSPARKX()`. The count matrix as extracted via `getCountMtr()` is provided as input for argument count_in. The coordinates, as extracted via `getCoordsMtr()` is provided as input for argument locus_in. The results are stored in slot @spatial of the SPATA2 object which is a list with a slot named $sparkx. The original output can be obtained via `getSparkxResults()`. A shortcut to obtain genes with significant spatial variability is offered by the function `getSparkxGenes()` which defaults to a $p$-value threshold of <0.05.

## Manual annotation of data points (spatial segmentation)

Manual annotation of data points is done with `createSpatialSegmentation()`. In the interface the function provides the user can interactively annotate data points on the histology image based on their spatial location. While drawing on the image the cursor's position is captured every five milliseconds and creates a detailed polygon of the annotated region. Each data point within the polygon is then labeled according to the user's designation and stored in a grouping variable in the SPATA2 object's feature data.frame. This allows for statistical testing and data subsetting based on the annotated regions. Results can be obtained via `getFeatureDf()` and can be used for downstream analysis by referring to the name of the created grouping variable (chosen by the user within the interface of `createSpatialSegmentation()`) in the recurring arguments across and grouping_variable as in `'my_spata_obj <- runDEA(object = my_spata_obj, across = 'histology')'`. We use the term spatial segmentation to contrast the labeling of barcoded spots from our spatial annotation approach outlined below.

## Distance and area measurements in SPATA2

Distance handling and transformation in SPATA2 is based on a package-built unit system that encompasses both SI units (nanometer (nm), micrometer (um), millimeter (mm), centimeter (cm)), and pixels (px). Every histology image processed by the method is accompanied by a set fiducial frame, with the example of the 10X Visium platform having a frame of 8 mm × 8 mm. However, the width and height of the loaded image in R may vary based on the resolution. This inbuilt system relies on the barcode-spots having the same distance of 100 μm to each other. Using that as the ground truth, the spatial coordinates of the spots provided in pixel units for alignment with the image section can be converted into SI units, or alternatively, the parameter adjustments that relate to distance measures can be converted to pixel units behind the scenes of each SPATA2 function. To simplify the transformation of pixel-based distances to SI units, SPATA2 offers several transformation functions. These functions allow to provide distance measures in SI units for precise adjustments of spatial parameters in algorithms such as Spatial Trajectory Screening and Image Annotation Screening as, for example, the `distance` and `binwidth` parameters. Additionally, the extracted image sections can be adjusted using these parameters. This concept is applied seamlessly to measuring areas in the case of image annotations, too.

## Spatial annotations

In spatial experiments, spatial annotations are pivotal for marking regions of interest. Unlike manual annotation of data points (e.g. barcoded spots, beats, or cells) through spatial segmentation, where each annotation is translated into a single label per data point, spatial annotations are distinct. Each annotation consists of at least one detailed polygon, with vertices outlining the area of interest, thereby defining its spatial boundaries. Additional polygons might be needed to outline holes within the annotation (Fig. 5a, b, vivid area). Each annotation is uniquely identified by an ID and can be enriched with tags and metadata. This information, encompassing spatial position, extent, tags, and metadata, is encapsulated in an S4 object of class `SpatialAnnotation`. We categorize spatial annotations into three types, each characterized by different methods of generating the outlining polygon: image annotations, numeric annotations, and group annotations. Supplementary Fig. 6a–c showcases each concept. Image Annotations focus on histomorphological features discerned through visual inspection of histological tissue. Users can manually outline structures of interest using an interactive interface, adding tags and labels as needed. This concept is depicted in Supplementary Fig. 6a and is facilitated by the SPATA2::`createImageAnnotations()` function in SPATA2. Group Annotations are generated automatically. Data points are grouped based on criteria like prior clustering. Then DBSCAN is applied to identify and remove spatial outliers that could disproportionately distort the outline. Lastly, the concaveman algorithm is employed to outline the filtered spots. This method is illustrated in Supplementary Fig. 6b and is implemented via the SPATA2::`createGroupAnnotations()` function in SPATA2. Numeric Annotations are automatically created by binning data points according to expression values, either through $k$-means clustering or a manually set threshold. DBSCAN is then applied to identify and remove spatial outliers that could disproportionately distort the outline. The concaveman algorithm subsequently outlines the remaining spots, as shown in Supplementary Fig. 6c. This process is executed through the SPATA2::`createNumericAnnotations()` function in SPATA2. Beyond their use in the spatial annotation screening algorithm, the spatial properties of these annotations, such as area, center, centroid, and outline, can be leveraged for further computational analysis and the development of customized analytical approaches. The user is free to add individually obtained spatial annotations using the function SPATA2::`addSpatialAnnotation()` which takes a data.frame of $x$- and $y$-coordinates that determine the vertices of the polygon outlining the area of interest.

## Spatial trajectories

Spatial Trajectories abstract a linear direction along with expression gradients inferred. They can be interactively created using the function SPATA2::`createSpatialTrajectories()`, which allows the user to interact with the spatial sample by determining the start and end point of the trajectory via double clicks. Alternatively, the trajectory's start and end points can be programmatically determined using

`SPATA2::addSpatialTrajectory()`. The results are stored in specific S4 objects that can be enriched with metadata. They also carry the ID with which each trajectory is identified.

### Simulation of expression patterns related to spatial annotations and trajectories

In our study, we simulated expression patterns related to spatial annotations and trajectories. This process was integral to validating our Spatial Gradient Screening methodology and to benchmark different evaluation metrics in their capability to quantify randomness as well as to benchmark their sensitivity to human deviations and variations caused by human bias. Furthermore, it is repeated with every call to the spatial gradient screening if argument `estimate_r2` is set to TRUE to estimate the reliability of the results. The simulation is structured into three steps. Supplementary Figs. 9, 10 visualize the steps for both, spatial annotations and spatial trajectories, respectively. The first step involves the computation of distance values for each data point (the same way as conducted during inferring an expression gradient). Following the distance computation, the data points are categorized into bins based on these distance values (Supplementary Figs. 9a, 10a). The bins are then systematically ordered in ascending fashion according to the mean distance values of each bin and provided with an index. In the second step, expression values are assigned to each member of a distance bin. These values are extracted from a numeric vector, the length of which corresponds to the number of bins. The arrangement of these values, when plotted against their indices in the vector, represented the specific pattern intended for simulation, as displayed in Supplementary Figs. 9c, d, 10c, d. Initially, this approach resulted in identical expression values for all members within a distance bin. Note that for visualization purposes a higher binwidth was chosen for Supplementary Figs. 9b, 10b. The simulation process we conducted for the actual benchmarking utilized a binwidth equal to the center-to-center distance of the visium spots, which is 100 μm. Results of this are displayed in Supplementary Figs. 9d–f,10d–f. The third and final step introduces noise into the simulated expression data in a controlled and systematic manner. This is achieved by creating a numeric vector from a randomized uniform distribution, equal in length to the number of data points. The range of values in this vector is aligned with the range of expression values assigned during the second step. The integration of noise with the pattern-like expression from the second step was executed in four distinct manners, each representing a different type of noise. Results of this integration are displayed in Supplementary Figs. 9d–f, 10d–f. These types included equally distributed (ED, *d*), where each data point's simulated and noisy expression values were scaled based on the noise ratio and then combined; equally punctuated (EP, *e*), where a percentage of randomly selected spots received random expression values; focally punctuated (FP, *f*), which differed from EP in that the spots receiving random values were not chosen randomly but were instead centered around initially selected data points, creating spatial niches of randomness; and Combined which amalgamated all three previous noise types). To ensure a comprehensive estimation, the simulations used for the benchmarking of our evaluation metrics spanned every possible combination of six pattern variations and four noise types. These were conducted at incremental noise levels, ranging from 0% to 100%, with a step size of 2%, resulting in a total of 61,200 simulations. Each simulation was uniquely named following a specific syntax: SE. <pattern>.<noise type>.<noise percentage>.<iteration>. This naming convention facilitated detailed tracking and analysis of each simulated iteration. We placed particular emphasis on Equally Distributed and Combined noise types, as they closely resemble patterns likely to be encountered in real-world data. While the Equally Punctuated and Focally Punctuated noise patterns are not commonly encountered in real-life scenarios, their inclusion was crucial for a thorough evaluation of our algorithm.

### Inference of gene expression gradients and screening

SPATA2 introduces two group-independent algorithms that allow the user to identify and visualize genes whose expression stands in meaningful relation to regions or microstructures identified by image analysis. Gene expression gradients can be inferred along spatial trajectories with Spatial Trajectory Screening or in spatial relation to image annotations Image Annotation Screening. First, this section explains how expression gradients are inferred along spatial trajectories. Second, it explains how expression gradients are inferred as a function of distance to image annotations. Third, it explains how the screening for specific gradients is conducted by fitting inferred expression gradients to predefined models and how their fit is evaluated.

### Inferring an expression gradient

By inferring an expression gradient, we mean capturing how the expression levels of a particular gene change in relation to a spatial feature, such as along a trajectory or depending on the distance to a spatial annotation's outline. This encompasses three substeps which are displayed in Supplementary Fig. 8a, b when using spatial annotations (a) and spatial trajectories (b). (Please refer to Supplementary Fig. 7 for a visual glossary of the terms used throughout the method section with regards to spatial annotations, trajectories, and the screening in general.). First, we calculate the distance of each data point to the relevant spatial feature. In the case of spatial trajectories all data points within the trajectory frame are projected onto the trajectory's $T$ that connects the origin of trajectory $P$ to the barcode spot of interest. Then, $C$ is projected onto $T$ such that projection $P$ corresponds to:

$$P = \frac{C \cdot T}{|T|} \tag{1}$$

The magnitude of the vector $P$ corresponds to the projection length (PL) and the projection length in turn corresponds to the distance along the trajectory:

$$Dist = PL = |P| \tag{2}$$

In the case of spatial annotations, each data point is projected to its closest vertex of the polygon, forming the outline of the spatial annotation and the distance is computed[22]. After obtaining the distance values, the gene's expression levels of each data point are related to the corresponding distance.

Then, locally weighted scatterplot smoothing (LOWESS or LOESS) is used to fit a curve that approximates the changes in gene expression along this distance. The $\alpha$ parameter for this loess fit, which determines the degree of smoothing, is standardized, and calculated as follows:

$$\alpha_{loess} = \frac{Resolution}{Distance \cdot CF} \tag{3}$$

where

- Resolution defaults to the average minimal center-to-center distance (CCD) of the data points. In the case of regularly fixed data points, as with Visium's barcoded spots, the value is given (100 μm). For irregularly scattered data points, as is the case for single cell-based platforms, this value is computed.
- Distance indicates the total distance covered in the screening.
- CF is the correction factor computed as the proportion of data points that exist from the total data points required to call the data set complete (see Supplementary Fig. 7c, d).

The resolution should not exceed the CCD. Generally speaking, the higher the resolution, the more reliable the results, however, the

resolution can not be increased infinitely since this will run into errors with the LOESS fitting. Generally speaking we found that a resolution between the CCD and half of the CCD provides good results. The reasoning behind using a correction factor is as follows: Due to platform limitations, tissue morphology, and data quality, the necessary data for a complete screening is often only partially available. For instance, Supplementary Fig. 7a–d illustrates a screening setup that references the largest of the three necrotic areas and includes the environment up to a distance of 3 mm. Supplementary Fig. 7d demonstrates how both the tissue's edge and the capture area of the Visium platform can impose constraints on data completeness. Consequently, the dataset represents only a subset of the data points needed to fully address the hypothesis. In this example, only about 42% of the necessary data points are available to consider the dataset complete in the context of our hypothesis. (If two or more annotations are used in the screening as displayed in Fig. 5b, the correction factor is computed according to the data requirements of all annotations.) A similar problem arises with spatial trajectories and the width of the screening area. If the alpha parameter of the LOESS fit only depends on the distance screened and the platform resolution, this can run into errors if the number of data points is too small due to incompleteness. Dividing the alpha parameter by the proportion of available data points ensures results in a higher alpha proportional to the incompleteness of the data set. If all required points are available, the proportion is 1 and the alpha parameter stays as is.

In summary, as the resolution of the platform increases, or the distance screened increases, the fit allows more details. The more incomplete the data set, the less details it allows. The more details it allows the better the screening can differentiate between random and non-random gradients as indicated by the R2 between total variation and noise percentage that is estimated beforehand to every screening set up if the argument `estimate_R2` is set to `TRUE`.

With inferred gradient we refer to a numeric vector of expression estimates capturing the resulting pattern of the fitted curve. To obtain it, we utilize the loess model for gene expression estimation along the distance via `stats::predict()`. The number and position of the expression estimates are computed by averaging the distance values of all data points within predefined distance bins, using a binwidth that corresponds to the spatial screening resolution. E.g. if the distance screened is 3 mm and the resolution is 0.1 mm, 30 expression estimates form the gradient approximately starting at a distance of 0.05 mm and ending at 2.95 mm. Finally, the vector of expression estimates is standardized to a defined range (we use 0–1, equivalent to 'low' to 'high'). When plotted against their corresponding distance values, the expression estimates form a polygonal curve representing the pattern of the inferred gradient.

## Identification of non-random gradients

The second step in spatial gradient screening is focused on the identification of genes whose expression gradients exhibit a pattern that is unlikely due to randomness. Depending on the algorithm used, spatial annotation screening (SAS) or spatial trajectory screening (STS), for every inferred gradient we posit the null hypothesis:

Null Hypothesis (H0): The expression pattern of the tested gene does not show spatial significance in relation to specific spatial references, such as delineated areas or spatial trajectories, and is attributable to random chance rather than being influenced by proximity to defined areas within the tissue.

Correspondingly, for every inferred gradient we formulate the alternative hypothesis:

Alternative Hypothesis (H1): The expression pattern of the tested gene exhibits spatial significance when analyzed in relation to specific spatial references, such as delineated areas or spatial trajectories. It forms a recognizable pattern that statistically distinguishes it from genes whose expression is not influenced by proximity to these areas,

suggesting a biologically meaningful connection between the gene and the reference feature.

Representative examples of either hypothesis as well as for either screening approach, are displayed in Supplementary Fig. 8a, b. To be able to adopt either of these hypotheses, we posit that, if the inferred gradient in question stems from a gene whose inferred expression gradient is dependent on the spatial trajectory or the spatial annotation, it should not merely consist of randomly scattered expression values. Instead, it should display a discernible degree of gradual expression change forming a recognizable pattern. Thus, the smoother the inferred gradient, the less random it is, and vice versa. Assuming this relationship, we quantify the degree of randomness of a gradient using its total variation (TV). This metric is calculated by considering the absolute differences between adjacent expression estimates, as displayed in Supplementary Fig. 8c. The total variation is calculated according to the formula:

$$TV = \sum_{i=1}^{n-1} |y_{i+1} - y_i| \qquad (4)$$

where

- TV is the total variation.
- $n$ is the number of expression estimates in the gradient.
- yi represents the gene expression value at expression estimate $i$.
- $|y_{i+1} - y_i|$ calculates the absolute difference between the gene expression values of adjacent expression estimates.

To assess the effectiveness of the total variation in capturing the percentage of noise or randomness introduced into a gradient, we leveraged the simulated dataset discussed above, where a certain degree of noise was introduced per simulation, represented by randomly generated expression values. Each resulting simulation was a combination of pattern-specific expression (Supplementary Figs. 9c, 10c) and randomly generated expression, based on different noise types (Supplementary Figs. 9d–f, 10d–f). We examined the relationship between the total variation and the degree of randomness across different underlying patterns and various types of noise. Our findings consistently demonstrated a strong linear relationship between the total variation and noise ratio across all simulation modalities using a resolution of 100 μm. The high $C$ values of 0.77–0.92 and 0.78–0.91 in the noise types equally distributed and combined (which we consider the most realistic manifestation of noise in real-life data) indicate that this metric effectively quantifies the degree of randomness.

Using `estimate_r2 = TRUE`, the spatial gradient screening algorithm conducts the simulation with the sample and the setup chosen by the user which estimates the reliability of the evaluation metrics in terms of their capability to account for randomness and noise. We recommend doing this, since we noted that the resulting $R^2$ varies depending on the distance screened, the resolution as well as the number of data points available. Generally speaking, we found that $R^2$ increased with increasing resolution. However, the resolution can not be increased infinitely since the data points available become too sparse for the LOESS to fit a curve to the expression vs. distance plot. We recommend choosing a resolution equal to or lower than the center-to-center distance of the data set, which can be obtained via `SPATA2::getCCD()`.

## Calculation of *p*-values in spatial gradient screening

In our study, we used total variation (TV) as a crucial metric to determine the randomness in an inferred gradient. We define the *p*-value for the hypotheses presented in the previous section as the likelihood of observing a TV as low as or lower than that of gradients observed under random conditions. Thus, to calculate a *p*-value for a given inferred gradient, we first simulate random expression gradients by assigning random expression values to all data points within the

screened area and continue to infer the resulting expression gradient as described above. This simulation is repeated 10,000 times to build a robust distribution of TV scores under random conditions. To further ensure a robust distribution of TV scores that represent the distribution of total variation under randomness potential outliers are removed by calculating the interquartile range (IQR) and identifying TV scores that lie significantly outside the IQR above the third or below the first quartile. Using the resulting distribution of randomly generated total variation values, we calculate the *p*-value according to the following formula:

$$p - \text{value} = \frac{\sum_{i=1}^{n} I(\text{rTV}_i \leq \text{oTV})}{n} \quad (5)$$

where

- *p*-value is the *p*-value.
- $\text{rTV}_i$ represents the total variation for the *i*th simulated gradient under complete randomness.
- oTV is the total variation of the observed inferred gradient from the gene of interest.
- *n* is the number of random simulations (with a default of 10,000), subtracted by the number of outlier TV removed by IQR.
- $I(\cdot)$ is the indicator function that equals 1 if the condition inside the parentheses is true and 0 otherwise.

Supplementary Fig. 11c, f provides a comprehensive visualization of the relationship between noise levels and *p*-values across all simulation modalities. In the case of equally distributed noise, we observed a consistent and stable relationship between *p*-values and noise ratios regardless of the underlying pattern, highlighting that the mentioned differences in TV baseline across simulated patterns did not have an effect. The resulting corrected *p*-values are adjusted according to the Benjamini–Hochberg approach and returned in a separate column called *fdr* (false discovery rate). We recommend using the adjusted *p*-values with a threshold of lower than 0.05.

### Identification of biologically relevant pattern

The final step of our methodology is centered on identifying non-random gradients that are indicative of biologically relevant patterns. In this process, each gradient inferred from the data is systematically compared against a series of predefined models. These models are numeric vectors that match the length and range of the inferred gradients and are designed to represent simplified versions of biologically relevant dynamics. Expression patterns can be complex and vary depending on the specific research questions. To address this variability, our approach not only supports the integration of user-defined models but also includes a basic set of models, as shown in Supplementary Fig. 14d. These standard models are developed to simplify complex gene expression patterns into three primary types, providing a practical and comprehensive framework for analyzing diverse biological data. This strategy ensures that our methodology is both versatile and grounded, capable of accommodating different research requirements while offering a solid base for the interpretation of gene expression patterns in relation to spatial features. The three patterns we provide standardized models for are:

- Association pattern: Higher gene expression near the annotation, decreasing with distance, indicative of an association. This is exemplified by the hypoxic gene signatures near the necrotic area (descending models, Fig. 4d).
- Recovery pattern: Lower expression near the annotation, increasing with distance, suggesting recovery. An example is the increase in oxygen-based metabolism away from necrotic areas (ascending models, Fig. 4g).
- Layered pattern: Transient increase in expression at a certain distance, forming a layer-like organization (peaking models, Fig. 2b).

Predefined models provide precision in addressing specific research questions, offering an intuitive framework for embedding findings in biological contexts (Supplementary Fig. 8d). The goodness of the fit between each gradient (*G*) and model (*M*) can be evaluated using two metrics: mean absolute error (MAE) and root mean squared error (RMSE). MAE evaluates average absolute deviation, robust to outliers and offering straightforward interpretation, and is computed according to the formula:

$$\text{MAE} = \frac{1}{n} \sum_{i=1}^{n} |G_i - M_i| \quad (6)$$

RMSE emphasizes larger errors and is sensitive to outliers. It is computed according to the formula:

$$\text{RMSE} = \sqrt{\frac{1}{n} \sum_{i=1}^{n} (G_i - M_i)^2} \quad (7)$$

### Identification of zero-inflated variables

Given the limitations of the spatial gradient screening algorithm in handling variables with a high proportion of zero values, we advise preemptively removing such variables. To facilitate this, we have incorporated an option within the algorithm, which can be activated by setting the parameter rm_zero_infl to TRUE. When this option is enabled, each variable considered for screening undergoes an outlier detection process. This process involves calculating the Interquartile Range (IQR) and excluding spots that fall significantly outside 1.5 times the IQR, either above the third quartile or below the first quartile. Should the process result in the retention of only zero values, indicating that all non-zero spots are outliers, the variable is deemed zero-inflated and subsequently removed from consideration. This approach helps mitigate the algorithm's sensitivity to zero-inflated variables, ensuring more robust and reliable screening outcomes.

### Sensitivity to human bias in spatial gradient screening

Both spatial reference features, spatial trajectories, and spatial annotations can be generated either computationally or manually through user interaction. While the interactive creation of both methods enables direct tissue interaction, it introduces the potential for human error, leading to variations in outlining spatial areas in the case of image annotations or drawing spatial trajectories. To assess susceptibility to human bias in both spatial annotation and spatial trajectory screening, we conducted a comprehensive investigation. In both approaches, we identified potential sources of human error and simulated deviations from originally created annotations or trajectories with increasing degrees of variation. Subsequently, we generated a ground truth dataset of expression variables using our simulated expression dataset, created using either the original trajectory or the original spatial annotation. The positive (non-random) ground truth consisted of a subset of 2400 simulated expression variables with an adjusted *p*-value (FDR) of 0. Conversely, the negative (random) ground truth was defined as a subset of expression variables with a noise percentage of 100%. Supplementary Figs. 6g, 12a presents the resulting distribution of total variation values for each population. These simulations were developed using the original spatial features. We then conducted spatial annotation and spatial trajectory screening with annotations and trajectories that deviated from the original ones in various ways. Subsequently, we compared the genes identified as random and non-random in these runs with the original population and quantified the ratio of false positives and false negatives. False positives were defined as randomly simulated expression variables incorrectly identified as non-random due to the introduced deviations in spatial features. Conversely, false negatives were defined as non-

randomly simulated expression variables falsely identified as random due to the introduced deviations in spatial features.

## Sensitivity to human error using spatial annotations

To introduce increasing variation into spatial annotations, we systematically added noise to the spatial annotation outlines and progressively rotated them by increasing degrees. The degree of introduced variation was quantified by measuring the deviation from the original outline and assessing their overlap. Supplementary Fig. 6d displays representative examples along with their quantified degree of deviation from the original outline, which is displayed in black, and their respective IDs. The IDs are used to map their screening results in Supplementary Figs. 6e, f. Notably, the percentage of false positives remains consistently low, staying close to 0%. However, the percentage of non-random expression variables missed due to the introduced variation increases linearly with the degree of deviation, with some outliers. Observing outliers along this linear increase suggests that not only the degree of deviation but also the nature and shape of the overlap contribute to decreased test performance measures. Most importantly, however, the increase in false negatives only became apparent when the outline deviation exceeded 20%, highlighting the robustness of spatial annotation screening via image annotations against human-introduced bias. Supplementary Fig. 6e illustrates a representative example with a 13% deviation from the original outline.

## Sensitivity to variations using spatial trajectories

The creation of spatial trajectories can introduce variations in terms of their start and endpoint, as well as variations in the angle at which they deviate from the original or optimal placement. To simulate variations in the start and endpoint, we generated trajectories and randomly displaced the start and endpoint of each trajectory randomly. Afterward, we measured the introduced deviation by calculating the resulting length of the new trajectory and subtracting it from the length of the original trajectory. Representative examples are presented in Supplementary Fig. 12b. Supplementary Fig. 12c displays the screening results compared to the introduced deviation in length, demonstrating that the percentage of false positives remains consistently low, approaching 0%, regardless of the introduced variation. It also indicates that variations in the start and endpoint do lead to non-random expression variables being missed (false negatives), with the proportion increasing linearly. However, the increase in false negatives only becomes noticeable when the deviation exceeds 0.75 mm (the original trajectory is approximately 5.75 mm in length), surpassing the 5% threshold. This suggests that there is a sufficient margin for variations in the start and endpoint. To simulate variations in degree, we generated deviating trajectories by displacing the endpoint along a vector perpendicular to the course of the original trajectory, resulting in trajectories with deviations of up to 25°. Supplementary Fig. 6d provides representative examples of these deviations. The test performance, influenced by the integrated deviations is displayed in Supplementary Fig. 6e. This figure demonstrates that the number of false positives remains unaffected, too, by variations introduced in this manner. Additionally, it reveals that the number of false negatives remains consistently low, only beginning to increase when the deviation degree reaches 15°. Based on the examples shown in Supplementary Fig. 12d, we conclude that the range of realistic variations introduced by human error does not exceed this threshold and spatial trajectory screening is robust with regards to this. Finally, we assessed the impact of the screening area size using spatial trajectories, defined as a rectangle formed by multiplying the trajectory's length with a parameter referred to as trajectory width. By default, the trajectory width equals the trajectory length, resulting in squares encompassing as many data points as possible. In cases where specific areas within this square might confound the screening process, one can either remove data points from these areas individually or reduce the width (Supplementary Fig. 7e–i). Our investigation revealed that reducing the width had no discernible effect on the number of false positives, which consistently remained low. Furthermore, the impact on the number of false negatives was negligible (Supplementary Fig. 12f, g). In conclusion, we found that spatial trajectory screening remains robust against variations introduced by human error.

## Comparison with test statistics derived from scRNA-seq pseudotime methods

Given that pseudotime-dependent gene expression identification methods and spatial gradient screening both aim to detect non-random patterns along a one-dimensional axis (distance or pseudotime), we conducted a comparative analysis of our total variation (TV) test statistic, as illustrated in Supplementary Fig. 8c, against well-known metrics from pseudotime-centric algorithms—specifically, the waldStatistic from tradeSeq and the log-likelihood from PseudotimeDE. To evaluate the correlation between each method's test statistic and the degree of noise obscuring the ground truth pattern in our simulated expression dataset, we selected a subset of 10,000 simulated genes from a 'combined' noise type. We analyzed their gradient in relation to the distance from a 'necrotic_center' spatial annotation, considering barcode spots as cells and utilizing distances up to 3 mm—the scenario for which the dataset was simulated.

For this analysis, distances to the annotation, retrieved via SPATA2::getCoordsDfSA(), substituted the pseudotime variable. Spots beyond the scrutinized region, the environment, were excluded. Following guidelines from the tradeSeq (https://statomics.github.io/tradeSeq/articles/tradeSeq.html) and PseudotimeDE (https://htmlpreview.github.io/?https://rpubs.com/dongyuansong/842884) tutorials, we fitted a generalized additive model (GAM) for each gene. For tradeSeq, we chose nknots=7 for tradeSeq::fitGAM() based on the Akaike Information Criterion (AIC) visual inspection. Due to computational limitations, a random sample of 1000 genes was analyzed for PseudotimeDE, using $n = 100$ subsamples for PseudotimeDE::runPseudotimeDE() as analyzing 10,000 genes would surpass 24 h on a system with 256GB RAM and 48 cores. We then calculated the linear correlation between the introduced noise in these simulated genes and the derived metrics from both tradeSeq (waldStat) and PseudotimeDE (test.statistics), alongside our total variation measure from SPATA2::spatialAnnotationScreening() (column tot_var). This correlation was assessed using the square of the base R cor() function, as depicted in Fig. 6b, c, to gauge the linear relationship between the noise level and the test statistics.

## Benchmarking computational efficiency

To measure runtime, we used bench::mark() on a subset of sample #UKF275_T_P, annotated for its hypoxic core, Supplementary Fig. 6c. Subsets included all combinations of randomly selected spots (35, 350, 3500) and genes (10, 100, 1000, 10,000), generated using the base R sample function. Each subset underwent 15 iterations. Benchmarks were run on a MacBook Pro with 32 GB RAM and 10 cores. Results are displayed in Fig. 6d.

## Cross-platform compatibility

Converting functions are provided by SPATA2 that seamlessly convert S4 objects of class spata2 to S4 objects from platforms such as Giotto or Seurat and vice versa as well as to AnnData-format for compatibility with platforms relying on Python. These functions come with the prefix as*(). E.g. `asGiotto()`, `asSeurat()`, `asSingleCellExperiment()`, `asAnnData()`.

## Data acquisition and processing of glioblastoma samples

The raw data for both samples, #UKF269 and #UKF313, were obtained from the online database of our 10XVisium platform using the SPATAData R package. SPATAData provides an interface with the

SPATAData::launchSpataData() function that allows access to all 10XVisium samples used in previous publications by the Micro-environment and Immunology Research Group Freiburg. Currently, this collection comprises 32 samples, including 25 malignancies of the central nervous system (CNS), such as #UKF269T and #UKF313T (for detailed information on how these datasets were generated, see Ravi et al. [4]). The raw data sets were downloaded using the SPATAData::downloadRawData() function. Data processing, clustering and DEA have been conducted the same way for both samples, as described below. From the downloaded data sets we used the initiateSpataObject_10X() function to create both SPATA2 objects. For data normalization, we used the pipeline with Seurat::SCTransform() using the default parameter setup of the function as illustrated in the tutorials on how to use Seurat::SCTransform() here: https://satijalab.org/seurat/articles/sctransform_vignette.html. As described previously, the count matrix, as well as the scaled matrix and dimensional reductions, are inherited by the respective SPATA2 object as outputted by initiateSpataObject_10X() which is a direct implementation of the default pipeline suggested by Stuart and Butler et al. 2019.

### Data acquisition and processing of mouse brain sample #MCI_LMU

**Mouse experiments.** Operations were performed on 9–11 weeks old C57Bl6/J male mice, housed and handled under the German and European guidelines for the use of animals for research purposes. Experiments were approved by the institutional animal care committee and the government of Upper Bavaria (ROB-55.2-2532.Vet_02-20-158). The anesthetized animals received bilateral stab wound lesions in the cerebral cortex by inserting a thin knife into the cortical parenchyma using the following coordinates from Bregma: RC: −1.2; ML: 1–1.2 and from Dura: DV: −0.6 mm. To produce stab lesions, the knife was moved over 1 mm back and forth along the anteroposterior axis from −1.2 to −2.2 mm. Animals were euthanized 3 days post-injury (dpi) by cervical dislocation[10].

**Visium experiments.** A mouse brain was embedded and snap-frozen in an isopentane and liquid nitrogen bath as recommended by 10x Genomics (Protocol: CG000240). During cryosectioning (Thermo Scientific CryoStar NX50), the brain was resected to generate a smaller sample, and two 10 µm-thick coronal sections of the dorsal brain area were collected in one capture area. The tissue was stained using H&E staining and imaged with the Carl Zeiss Axio Imager.M2m Microscope using ×10 objective (Protocol: CG0001600). The sequencing library was prepared with the Visium Spatial Gene Expression Reagent Kit (CG000239) using 18 min permeabilization time. An Illumina, a paired-end flow cell, was used for sequencing on a HiSeq1500 following manufacturer protocol, to a sequencing depth of 75,398 mean reads per spot. Sequencing was performed in the Laboratory for Functional Genome Analysis of the LMU in Munich.

**scRNA-seq experiments.** The lesioned grey matter of the somatosensory cortex from three C57BL/6J mice at 3dpi was isolated using a biopsy punch (∅ 0.25 cm) and the cortical cells were dissociated using the Papain Dissociation System (Worthington, # LK003153) followed by the Dead Cell Removal kit (Miltenyi Biotec # 130-090-101), according to manufacturer's instructions. Incubation with dissociating enzyme was performed for 60 min. Single-cell suspensions were resuspended in 1xPBS with 0.04% BSA and processed using the Single-Cell 3′ Reagent Kit v2 from 10x Genomics according to the manufacturer's instructions. In brief, this included the generation of single-cell gel beads in emulsion (GEMs), post-GEM-RT cleanup, cDNA amplification, and library construction. Illumina sequencing libraries were sequenced on a HiSeq 4000 following manufacturer protocol, to a mean depth of 30,000 reads per cell. Sequencing was

performed in the genome analysis center of the Helmholtz Center Munich.

**scRNA-seq data analysis.** Read processing was performed using 10X Genomics Cell Ranger (v3.0.2). After barcode assignment and UMI quantification, reads were aligned to the mouse reference genome mm10 (GENCODE vM23/Ensembl 98; 2020A from 10xGenomics). Further processing was performed using Scanpy[1] (v1.9.1). Cells were excluded if they had ≤300 or ≥6000 unique genes, or ≥20% mitochondrial gene counts. The count matrix was normalized (sc.pp.normalize_total) and log($x+1$)-transformed (sc.pp.log1p), before proceeding with dimensionality reduction and clustering (sc.tl.pca, sc.pp.neighbors with n_pcs=20, sc.tl.umap, sc.tl.leiden with resolution=0.6). Cell types were manually annotated using known marker genes ('ECs': ['Cldn5', 'Pecam1'], 'Mural cells': ['Vtn', 'Pdgfrb', 'Acta2', 'Myocd'], 'Fibroblasts': ['Dcn', 'Col6a1', 'Col3a1'], 'Oligodendrocytes': ['Mbp', 'Enpp2'], 'OPCs': ['Cspg4', 'Pdgfra'], 'Neurons': ['Rbfox3', 'Tubb3'], 'Astrocytes': ['Aqp4', 'Aldoc'], 'Microglia': ['Aif1', 'Tmem119'], 'Monocytes/Macrophages': ['Cd14', 'Itgb2', 'Cd86', 'Adgre1'], 'B cells': ['Cd19'], 'T/NK cells': ['Cd3e', 'Il2rb', 'Lat'], 'Neutrophils': ['S100a9']).

**Visium data analysis.** Read processing was performed using 10x Genomics Space Ranger (v1.2.2). After barcode assignment and UMI quantification, reads were aligned to the mouse reference genome mm10 (GENCODE vM23/Ensembl 98; 2020A from 10xGenomics). Scanpy (v1.9.1) was used for further processing of the Visium dataset. Barcode spots with <400 counts were excluded (sc.pp.®lter_cells). The count matrix was normalized (sc.pp.normalize_total) and log($x+1$)-transformed (sc.pp.log1p), before proceeding with dimensionality reduction and clustering of barcode spots (sc.tl.pca with n_comps = 40, sc.pp.neighbors, sc.tl.leiden). Clusters were annotated based on histology and known locations of the injury sites. The full-sized Space Ranger input image (7671×7671 px) was used to segment nuclei using Squidpy and Cellpose[23] via sq.im.segment with method = cellpose_he and flow_threshold = 0.8; as suggested in https://squidpy.readthedocs.io/en/stable/external_tutorials/tutorial_cellpose_segmentation.html). Next, Tangram was used to integrate scRNA-seq and Visium datasets by providing a cell type probability score per barcode spot, based on spatial correlation of genes shared by the datasets[2]. This probability score was used to deconvolve the Visium dataset by assigning each segmented nuclei a most likely cell type (using tg.pp_adatas with 1238 overlapping training genes from the top 125 marker genes of each single-cell cluster, tg.map_cells_to_space, tg.project_cell_annotations, tg.create_segment_cell_df, tg.count_cell_annotations, tg.deconvolve_cell_annotations; total assigned nuclei 4,356; as suggested in https://squidpy.readthedocs.io/en/stable/external_tutorials/tutorial_tangram.html). Processed h5ad files were imported to SPATA2 by first loading them to R via anndata::read_h5ad() and converting them using asSPATA2().

### Initiation and processing of the SPATA2 objects

Data was read into R using SPATA2::initiateSpataObjectVisium() from the SpaceRange outs folder. The count matrix was processed using the R function Seurat::NormalizeFeatures(). For further analysis, the normalized matrix of slot @layers from the corresponding Seurat object was used. The tissue outline (tissue edge) was identified using SPATA's inbuilt image processing pipeline SPATA2::identidySpatialOutliers(). Spatial outlier spots were identified and removed using SPATA2::identifySpatialOutliers() and SPATA2::removeSpatialOutliers(). Afterward, we clustered the barcoded spots using the BayesSpace algorithm as implemented in SPATA2::runBayesSpaceClustering() (number of clusters ranging from $n=3$ to $n=15$). Next, we identified spatially

variable genes using the SPARKX implementation `SPATA2::runSparkx()`.

## Downstream analysis of sample #UKF269

First, we inferred copy number alterations using SPATA2 implementation of the infercnv R-package via `SPATA2::runCNV()`. Second, for sample #UKF269 we created a grouping variable based on the histological architecture of the sample using the function `SPATA2::createSpatialSegmentation()` called histology. The results can be obtained via the R command `spatial_segmentations$T269`. Differential expression analysis based on the grouping variable histology was conducted using the default of `SPATA2::runDEA()` which in turn calls the default of `Seurat::FindAllMarkers()`. The default parameters were used. DEA was conducted based on the grouping of the Bayes space clustering and the histological segmentation using the default of `runDEA()`. Identification of spatially variable genes was conducted using `SPATA2::runSparkx()`. The spatial trajectory was named *horizontal_mid* and added via `SPATA2::addSpatialTrajectory()`. Start point was set to the min of all *x*-coordinates and the mean of all *y*-coordinates. End point was set to the max of all *x*-coordinates and the mean of all *y*-coordinates. Spatial trajectory screening was conducted with the function `SPATA2::spatialTrajectoryScreening()`. The parameter `variables` were set to the vector of genes that were identified as spatially variable by SPARKX as obtained by `getSparkxGenes(…, threshold_pval = 0.05)`. The output of STS is an S4 object of class `SpatialTrajectoryScreening` containing the results in slot `@results`. This is a data.frame in which each row corresponds to ta gene-model fit as indicated by the columns variables (theoretically, all numeric variables can be included in the screening process) and models.

## Downstream analysis of sample #MCI_LMU

Differential gene expression analysis was conducted using the function `runDEA()` based on the clustering suggested by the Scanpy pipeline (see above). The function `SPATA2::createImageAnnotations()` was used to manually draw image annotations guided by prior information on injury location and histology. The two image annotations were labelled inj1 (upper section) and inj2 (lower section). Spatial annotation screening was conducted two times. The first run (Main Fig. 3) included parameter adjustments `distance = "1.5 mm"` and `resolution = "50um"`. Models corresponding to the gene expression gradient of Hmox1 and Lcn2 used for the screening were obtained via `SPATA2::getSasDf()` with equal parameters to the screening, converted to a list via `base::as.list()` and added with the argument `add_models`. The second run (Supplementary Fig. 3) included parameter adjustments `distance = "0.75 mm"` and `resolution "50 um"`. In both cases, as well as for all visualizations, parameter `ids` were set to `c("inj1", "inj2")` including both injury annotations.

## Downstream analysis of sample #UKF313

After visual inspection of the histology image and identification of the necrotic area as well as the pseudopallisades we created a spatial annotation that captured the spatial extent of the central necrotic area using the function `SPATA2::createImageAnnotations()` and named it necrotic_center (Supplementary Fig. 7a). The vivid part within the annotated area of necrotic_center was also interactively annotated and labeled vivid. Additionally, the spatial annotations necrotic_edge and necrotic_edge2 were also created within the interactive interface of `SPATA2::createImageAnnotations()`. The spatial annotation necrotic_center was equipped with the spatial annotation vivid as a hole using `SPATA2::addInnerHoles()` and the resulting spatial annotation was called necrotic_area. The spatial annotation screening, as visualized in Fig. 5b, was conducted using the function `SPATA2::spatialAnnotationScreening()` with parameter ids set to

`c("necrotic_area", "necrotic_edge", "necrotic_edge2")`, distance set to "dte". Genes screened were subsetted according to the list of spatially variable genes as provided by `SPATA2::getSparkxGenes(…, threshold_pval = 0.05)`. The output of the image annotation screening algorithm is an S4 object of the class `SpatialAnnotationScreening`. Resulting model fits were filtered for genes with an adjusted *p*-value (FDR) of <0.05 and with an RMSE evaluation of <0.25 for either of the descending models (Fig. 5c) or the ascending models (Fig. 5d). The remaining genes were grouped by the model class (descending or ascending) and supplied to `hypeR::hypeR()`. Gene sets were provided by `SPATA2::getGeneSetList`. The results were filtered for gene sets in either group with an FDR <0.05. Gene sets used for Fig. 5e, h were picked as examples for either group. Their original name, as listed in the data.frames of SPATA2 are HM_HYPOXIA and RCTM_CELLULAR_HEXOSE_TRANSPORT (e) as well as HM_OXIDATIVE_PHOSPHORYLATION and RCTM_TCR_SIGNALING (h).

## Cell2Location

We have integrated the cell2location model into our study, using it to bridge the Visium spatial transcriptomics data with the GBMap single-cell dataset of glioblastoma. The single-cell dataset was downsampled to 100,000 cells to accommodate computational demands. Signature estimation from the single-cell dataset was conducted via the cell2location Negative Binomial regression model, producing the inf_aver_sc.csv file, which served as the foundation for the spatial deconvolution process. Shared genes between the signature genes and the spatial dataset were identified, leading to the initiation of the cell2location model. The model was trained adhering to recommended hyperparameters and utilizing early stopping criteria based on ELBO loss. Post-training, the posterior distribution of cell abundance was quantified and extracted for subsequent analytical pursuits. The expected expression for each cell type was computed, and cell-specific expressions were documented.

## CytoSpace

For the decomposition of cell types, we utilized the GBMap atlas, which encompasses a dataset of over one million cells. We have developed a pipeline for single-cell deconvolution employing CytoSpace in conjunction with SPATA objects, details of which are accessible at our dedicated GitHub repository (githun.com/heilandd). The R script named "CytoSpace_from_SPATA.R" provides a detailed workflow for preparing files compatible with the CytoSpace suite, supplemented by a bash script to facilitate the batch processing of SPATA2 objects. The CytoSpace analysis itself runs within a bash environment, and upon its completion, a script is made available for importing the results back into the SPATA2 framework using the CytoSpace2SPATA function.

## SPTCR-seq

We utilized the spatial T cell receptor sequencing samples published in Benotmane et al. 2023. Data were downloaded at GEO accession code GSE238071 and processed by the analysis script https://github.com/heilandd/SPTCR_seq_code.

## Horizontal integration of spatial annotation screening

Horizontal Integration of the Spatial Annotation Screening was performed with the output of the `SPATA2::getSasDf()` function which provides inferred expression estimates at distance intervals as explained in section Inferring an expression gradient. The `distance` parameter was set to 3 mm and the `resolution` parameter was set to 100 μm. The data.frames of all six samples containing their respective expression estimates of all variables displayed in Fig. 5 were merged using `base::rbind()`.

## Statistics and reproducibility

Statistical analysis was conducted with R (version 4.1.2). No statistical method was used to predetermine the sample size. No data was excluded from the analysis. The experiments were not randomized. The investigators were not blinded to allocation during experiments and outcome assessment. SPATA2 is a package that undergoes continuous improvements and adjustments. A stable version that can be used to reproduce the analysis and figures of this manuscript using the source data file can be installed via 'devtools::install_github(repo = "kueckelj/SPATA2", ref = "0e3eb85")'. The latest version of the package can be installed via 'devtools::install_github(repo = "theMILOlab/SPATA2")'. The following packages are required as dependencies for the SPATA2 version with which this study has been conducted: BiocGenerics >= v0.40.0; DT >= v0.23; DelayedArray >= v0.20.0; DelayedMatrixStats >= v1.16.0; EBImage >= v4.36.0; FNN >= v1.1.3.2; Matrix.utils >= v0.9.8; S4Vectors >= v0.32.4; Seurat >= v5.0.2; SingleCellExperiment >= v1.16.0; SummarizedExperiment >= v1.24.0; aplot >= v0.1.6; batchelor >= v1.10.0; broom >= v0.8.0; colorspace >= v2.1-0; concaveman >= v1.1.0; confuns >= v1.0.3; dbscan >= v1.1-10; dplyr >= v1.1.2; ggalt >= v0.4.0; ggforce >= v0.3.3; ggplot2 >= v3.4.3; ggridges >= v0.5.3; ggsci >= v2.9; glue >= v1.7.0; grid >= v4.1.2; keys >= v0.1.1; limma >= v3.50.3; lubridate >= v1.8.0; magick >= v2.7.3; magrittr >= v2.0.3; paletteer >= v1.4.0; pheatmap >= v1.0.12; pracma >= v2.3.8; progress >= v1.2.2; psych >= v2.2.5; purrr >= v1.0.1; readr >= v2.1.2; reticulate >= v1.34.0; rlang >= v1.1.1; scattermore >= v1.2; shiny >= v1.7.1; shinyWidgets >= v0.7.0; shinybusy >= v0.3.1; shinydashboard >= v0.7.2; shinyhelper >= v0.3.2; sp >= v1.5-0; stringi >= v1.7.6; stringr >= v1.5.0; tibble >= v3.2.1; tidyr >= v1.2.0; tidytext >= v0.3.3; umap >= v0.2.8.0; units >= v0.8-0; viridis >= v0.6.2.

## Reporting summary

Further information on research design is available in the Nature Portfolio Reporting Summary linked to this article.

# Data availability

The raw stRNA-seq data from human glioblastoma used in this study have been deposited at https://datadryad.org/stash/dataset/doi:10.5061/dryad.h70rxwdmj. The raw scRNA-seq and stRNA-seq data from the injured mouse brain is deposited at https://www.ncbi.nlm.nih.gov/geo/query/acc.cgi?acc=GSE226211. Source data and code to reproduce the panels presented in all main and Supplementary Figs. are available in the source data file. Cluster results, interactively created segmentations as well as spatial annotations are additionally available as lists in the SPATA2 package. The data can be obtained using the commands `SPATA2::clustering`, `SPATA2::spatial_segmentations`, `SPATA2::spatial_trajectories` and `SPATA2::spatial_annotations`. Furthermore, processed SPATA2 objects used in this study can be downloaded using `SPATA2::downloadFromPublication()`. Source data are provided with this paper.

# Code availability

The SPATA2 package is available https://github.com/theMILOlab/SPATA2. SPATA2 version 3.0.0 which contains the features presented in this manuscript will be made available within two weeks from publication of this manuscript. Further information and requests for resources, raw data and reagents should be directed and will be fulfilled by the contact: D. H. Heiland, henrik.heiland@uk-erlangen.de.

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

# Acknowledgements

This project was funded by the German Cancer Consortium (DKTK), Else Kröner-Fresenius Foundation (DHH). The work is part of the MEPHISTO project (DHH), funded by BMBF (iGerman Ministry of Education and Research) (project number: 031L0260B). Funding was received from the German Research Foundation (DFG) as part of the Munich Cluster for Systems Neurology (EXC 2145 SyNergy – ID 390857198, to MD), the CRC 1123 (B3; to MD), DI 722/16-1 (ID: 428668490/40535880, to MD), DI 722/13-1, and DI 722/21-1 (to MD); a grant from the Leducq Foundation (to MD);

the European Union's Horizon Europe (European Innovation Council) programme under grant agreement No 101115381 (to MD); ERA-NET Neuron (MatriSVDs, to MD), and the Vascular Dementia Research Foundation. SF was supported by the Joachim Herz Foundation.

## Author contributions

The study was designed and coordinated by D.H.H. SPATA2 was developed by J.K., D.H.H. SPATA2 is maintained by J.K. and S.F. Spatial gradient screening was conceived by J.K., D.H.H., S.F. and J.K.B. The statistical and informatic implementation was conducted by J.K. Simulations and benchmarking were conducted by J.K. and S.F. Mouse experiments and generation of mouse cortex Visium and scRNA-seq data were carried out by C.K. and J.N. Visium samples were selected and processed by S.F. Analysis of glioblastoma and human neocortex data sets was conducted by J.K., S.F., and D.H.H. Analysis of mouse cortex data sets was conducted by J.K. and S.F. Main figures were created by J.K. and D.H.H. Supplementary Figs. were created by J.K. and S.F. Main part of the manuscript was written by J.K., D.H.H., and S.F. Method part of the manuscript was written by J.K. The manuscript was edited by O.S., J.B., M.D., and J.K.B.

## Funding

## Competing interests

The authors declare no competing interests.
