## [Peer Review File · Nature Communications]

Reviewers' Comments:

Reviewer #1:

Remarks to the Author:

In this manuscript by Kueckelhaus et al., the authors describe two new algorithms added to SPATA2, an open-source R-based bioinformatics framework for spatial transcriptomics data analysis, they developed earlier.

The addition of the two new algorithms (Spatial Trajectory Screening (STS) and Image Annotation Screening (IAS)) allows for exploration of gene expression patterns within tissue microdomains and at cellular level within the native histological architecture.

The authors demonstrate the use and functionality of the upgraded and improved SPATA2 on 10x Visium samples from two brain disease models: malignant brain tumor (glioblastoma) and traumatic brain injury.

Using these new features of SPATA2, the authors convincingly show its capability to identify gradually varying gene expression patterns between histologically distinct regions, including normal brain areas, adjacent transition areas and tumor core. This is certainly an important feature of spatial transcriptome profiling, as it allows for a comprehensive interpretation of gene expression changes (abrupt or gradual) between these areas. Additionally, the analyzes might provide valuable insights regarding disease progression and prognosis. Spatially segregated marker gene expression analysis within the transition zone, tumor core and normal brain tissue, confirmed the boundaries set by histological analyzes of the tissue.

The authors identify two limitations to the STS algorithm: can screen in only one direction at a time and it has low flexibility due to linear and rectangular shape of the area of interest. To overcome this limitation, the authors introduced IAS as an additional algorithm, which allows screening the local surrounding of an image annotation. The authors show the utility of this tool, when analyzing the necrotic area surrounding the glioblastoma. Because of IAS' flexibility in defining the area that marks the microstructure of interest and the surrounding area to be screened, this tool could be useful in other research areas of neurobiology of disease. For instance, these tools could be used in studies of gene expression changes following ischemic stroke or amyloid plaque deposition and progression of neuroinflammation. Analyzing the gradient of gene expression depending on the distance from damaged area or plaque deposition, could provide insights into biological processes that predict disease progression and/or outcome of treatment strategies. Therefore, these tools have the potential to be used in translational/clinical research. A TBI model was used to demonstrate the utility of SPATA2 in the investigation of fine-grained transcriptome changes at the site of injury and in the surrounding brain areas. Using a single-cell RNA sequencing data set from the same TBI mouse model with multiple cell types from the brain, in combination with cell segmentation of the Visium H&E images, the authors were able to identify single cell transcriptome profiles, that showed cell type density gradients depending on the distance from injury. As a proof of principle, the cells that showed gradient density were identified based on marker genes as macrophages and monocytes.

This manuscript has the strength of providing the research community with an open-source tool for in-depth analysis of spatial gene expression data. Although the two disease models that were used for demonstration purposes are brain disorders, I think the tool can be applied to other tissues as well. The authors should expand in the discussion the applicability. If there is a way to test on other tissue, it would make the paper stronger.

The authors did not mention in the manuscript whether SPATA2 is compatible with Visium only. It would be good to test if it is compatible with other techniques besides Visium, such as GeoMax.

I recommend that in discussion the authors present potential clinical applications, including how this tool might aid drug discovery.

Please also check for editing, small grammatical mistakes and incomplete sentences, such: "SPATA2 implements a variety of external algorithms and presents them in user-friendly wrapper

functions that.”

Reviewer #2:

Remarks to the Author:

Kueckelhaus and colleagues propose that two classes of gene expression patterns are of interest in relation to histological features in tissues.

One class is a set of unidimensional functions along a rectangle defined in an image.

The second class is a set of bivariate functions gradually increasing (or decreasing) with the proximity to the outside of a polygon enclosing a region of interest.

The authors argue that identifying genes whose expression closely follow these patterns are of greater interest than genes whose means change between histologically defined regions.

It is motivated why the authors find the linear rectangular gradients of the first class interesting. It is not extremely clear why specifically genes increasing in value on the outside of the polygon in the second class is biologically interesting.

For the sake of identifying genes following these patterns, the authors developed a piece of software that allows a user to interactively define rectangles for the first class, and polygons for the second class, in H&E stain images paired with 10x Visium gene expression measurements.

Unlike typical statistical inference, here each gene's observed expression value is compared to a small finite set of manually pre-defined functions (there is no parameter inference to cover a continuous space of functions).

Fit between the function and the observed values is evaluated using an average of Pearson correlation and something the authors refer to as 'Residuals Area Over the Curve' (and 'Residuals Area Under the Curve'). These RAOC/RAUC terms appear to be a new definition by the authors, since there is no reference, it is not standard terminology in the field, and Google searching these terms only returns the website for the tool presented in this manuscript.

The definition of RAUC in equation (5) is inscrutable. What is the residual of? What is the function f ? Why is this fitness measure of interest rather than something more standard such as root mean squared error or mean of absolute errors, or some statistical likelihood?

Throughout the manuscript, the authors report P-values for the combined RAOC-Pearson fitness measures. It is not described how these P-values are obtained, and it is unclear how that would even be possible with the odd RAOC component.

In all, the authors wanted to find genes following certain patterns, and wrote an approach to do this, which is presented with some examples in this manuscript.

There are no discoveries, no conclusions, no insights, nor confirmations, claimed by the authors. Only a large number of reported observations without connection to any particular hypothesis or question.

There are no methods to compare the authors' approach to, since they are likely the first to propose looking at these particular expression patterns. However, there is also not much in terms of investigation that would be typical for a new regression method. No investigation into sensitivity

to noise. No investigation into false positive rates using simulated negative data. Etc.

Minor issues:

How did the authors define the 'tumor', 'transition', and 'infiltrated cortex' classes in the glioblastoma data?

On line 401-401 "Figure 6e-j", it is unclear if these example genes refer to genes that correlate with Hmox1 / Lcn2 levels, or if they are genes that follow distance based gradients.

Figure numbering for supplementary figures seems wrong?

There are several cases where figures in the manuscript appear to be covering the main text, making it impossible to read. For example, line 247 ends with starting a new sentence, but it is not clear where the continuation of this sentence is.

Reviewer #3:

Remarks to the Author:

Review report on "Inferring histology associated gene

expression gradients in spatial transcriptomic studies"-

NCOMMS-23-23627-T

In this manuscript, the authors present two algorithms, namely Spatial Trajectory Screening (STS) and Image Annotation Screening (IAS). These algorithms aim to capture local gene expression signatures without the prerequisite of prior clusters. They have implemented these algorithms in a high-quality open-source R package. While the introduction of these algorithms is intriguing, there are certain concerns that need to be addressed.

1. The tissue examined in this manuscript demonstrates notable differentiation among the various histological regions. However, it remains uncertain whether this method can be effectively applied to H&E images lacking evident demarcations. For instance, consider the case of cancerous tissue (HCC.tif) depicted in the following link: <https://doi.org/10.6084/m9.figshare.21061990.v1>, in which the tumor regions are distributed in multiple regions.
2. The reliability and validity of the obtained results are subject to the accuracy and quality of the researcher's manual segmentation of the H&E image for spatial domain assignment. In the event of inaccurate or flawed segmentation, there may be concerns regarding the reliability of the findings. Therefore, conducting sensitivity analysis becomes essential to assess the robustness of the results and account for potential variations stemming from segmentation errors.
3. The specification of the spatial trajectory in this study is user-dependent, which introduces three concerns. Firstly, if users do not possess a priori knowledge or specific interests in the trajectory, is there a method available to generate latent spatial trajectories using algorithms? For instance, in scRNA-seq data analysis, can the cell lineage trajectory inference methods be employed? Secondly, is the screening of spatial trajectories sensitive to slight variations, such as changes in the start point, end point, or width? Additionally, the authors established the spatial trajectory of sample #UKF269 with a width of 3mm and length of 5mm. Are there any guidelines or rationale behind these specification? Thirdly, an important aspect to address is whether the trajectory can be a curve rather than strictly limited to a straight line. Exploring the possibility of curved trajectories is crucial for capturing more complex spatial relationships and patterns that may not be adequately represented by linear trajectories.

4. On line 1030, page 31, the statement mentions that "as the trajectory had a length of 5mm, this resulted in 50 bins." Why "that the trajectory had a length of 5mm

resulted in 50 bins"? And what is the difference of more or less bins?

5. In the analysis of sample #UKF313, for a researcher not a pathologist, it is not an easy thing to identify the necrotic area. But the image annotation constitutes the important first step, it may hinder the applications of SPATA2 for non-pathologist. Have the authors considered employing computational methods for image annotation?

6. During the downstream analysis of sample #MCI_LMU, it is stated that "Only genes with a p-value lower than 0.05 were included in the subsequent analysis and visualization steps." However, to effectively control the familywise error in multiple testing scenarios, it is recommended to use adjusted p-values instead of raw p-values. Therefore, the authors should consider incorporating genes with adjusted p-values below 0.05 in their analysis to ensure more rigorous statistical control and accurate interpretation of the results.

7. In Supplementary Figure 4 (f)-(h), left pane, the meaning of the x-axis is not explicitly labeled. This lack of labeling makes it unclear whether it represents the trajectory direction or some other variable. In contrast, Supplementary Figure 3 (iii) clearly labels the x-axis as "Trajectory direction". To ensure consistency and clarity across the figures, it is advisable to label the x-axis consistently across all figures representing the same concept.

8. It appears that SPATA2 is specifically designed for use with the 10X Visium platform. However, it raises the question of whether it can be applied to other platforms commonly used in spatial transcriptomics, such as seqFISH, Slide-seq, Slide-seqV2, or Stereo-seq.

Minors:

1. On line 635, page 21, there appears to be a typo in the sentence. The phrase "Additionally, barcode-spots. the local surrounding of image annotations..." seems to be missing proper punctuation and may contain an extra period. Please carefully check the full text.

2. Line 1034, page 34, "in which each row corresponds to ta gene-model fit ...", where "ta" may be "a"?

3. To ensure clarity and proper referencing, it is recommended to include the analyzed dataset information in the general title of each main figure. This practice allows readers to immediately identify the specific dataset under investigation.

Point by Point

Section 1, General Comments:

We would like to express our sincere gratitude to the reviewers for dedicating their time and expertise to review our manuscript. Their feedback has been instrumental in shaping our work, leading to significant enhancements in our approach. In this document, we systematically address each of the reviewers' comments, providing comprehensive responses and corresponding revisions in a structured chronological order. While doing so, we refer to figures in this letter with *Figure n*, main figures in the revised manuscript with *Main Figure n* and supplementary figures in the revised manuscript with *Supplementary Figure n*. To enhance the clarity and comprehensibility of the revisions made, we have incorporated three preliminary sections before delving into our responses to the reviewers' concerns. These sections aim to elucidate essential aspects of our methodology and changes we've made. While the method section contains a more detailed description, we believe that additional presentations of these fundamental steps will aid in better understanding our approach.

In Section 2 “**Fundamental Changes**”, we offer an initial overview of the modifications made compared to the initial manuscript. This section focuses on conceptual changes and nomenclature updates.

In Section 3, “**Simulation of Expression Data**”, we provide a step-by-step illustration of our process for simulating the dataset necessary for conducting the requested benchmarking and exploring statistical robustness.

In Section 4, “**Spatial Gradient Screening Revised**”, we present our refined approach for inferring expression gradients along Spatial Trajectories and as a function of distance to spatial areas. This section outlines our revised methodology and key improvements.

After our general novel changes, we discuss each reviewer's comments in detail (Point by Point). Throughout our responses, we refer to the information presented in Sections 3 and 4 as well as to visualizations in main and supplementary figures of the revised manuscript. We are committed to addressing the reviewers' feedback thoroughly and hope that these additional sections and revisions contribute to a more comprehensive and improved manuscript.

Section 2, “Fundamental Changes”:

The insightful comments and suggestions provided by the reviewers have led us to reevaluate and modify some fundamental aspects of the SPATA2 package, as well as the algorithms and analysis methodologies outlined in this paper. While we are committed to addressing each comment in detail, we believe it is beneficial to introduce you to the significant changes upfront. We are doing so to establish a clear reference point for these alterations since we will be referring to them by name throughout our responses.

- Building upon the comment provided by reviewer 3, we have expanded the original concept of *Image annotations* into a broader framework we named *Spatial annotations*. Unlike the previous approach that solely permitted **manual** marking of areas of interest, we have introduced two additional methods for annotating spatial regions of interest: one based on continuous expression (*Numeric Annotations*) and another based on the spatial aggregation of groups, such as clusters (*Group Annotation*). All three methods for annotating space now fall under the umbrella of spatial annotations. Consequently, the algorithm *Image Annotation Screening IAS* has been renamed as *Spatial Annotation Screening* to accurately reflect this enhanced scope. For a more detailed explanation of how the two new types of Spatial Annotations are implemented, please refer to our response to concern 5.3.4. For an elaboration of our motivation to enhance the scope and how this is embedded in a broader picture please refer to the revised introduction of our manuscript as well as to our response to comment 5.2.1.
- The term *Spatial Gradient Screening* is now used to refer to overall concept of both approaches *Spatial Trajectory Screening* and *Spatial Annotation Screening* where first the gradient of a numeric variable is inferred, second the inferred gradient is tested for non-randomness and third the inferred gradient is fitted to predefined models for guidance in biological interpretation.
- We have conducted a comprehensive revision of the Spatial Gradient Screening (SGS) methodology. While the core concept remains intact, significant changes have been made, particularly in how we obtain a spatial gradient. Instead of binning barcoded spots and averaging gene expression by distance bins, we now employ polynomial regression fitting (`stats::loess()`) to capture the overall expression gradient pattern. Eventually, the gradient pattern is obtained by predicting gene expression along the screened distance using the loess model and `stats::predict()`. We found that this revised approach adheres more to the continuous nature of gene expression is more robust against outliers.

- In this manuscript, our primary focus is on utilizing the Visium platform, which quantifies gene expression as mRNA counts mapped to barcoded spots uniformly distributed across the sample's spatial layout. It's important to note that the methodology we present here is versatile and can be applied to various types of numeric, continuous data. This includes spatial protein expression, metabolite counts, and even gradients of cellular density. For the sake of clarity and simplicity, we collectively refer to all these data types as "gene expression." Similarly, we use the term "barcoded spots" or simply "spots" to refer to the observational unit, encompassing spots, beads (SlideSeq), or cells. This uniform terminology enhances readability and understanding throughout the manuscript.

Section 3, "Simulation of Expression Data":

Due to the reviewers' recurring concerns centering around benchmarking, sensitivity to noise, and the need for further investigation into statistical performance, we took the initiative to construct a simulated expression datasets for both samples and thus for both methods, Spatial Annotation Screening and Spatial Trajectory Screening. Our goal was to conduct a comprehensive examination of the statistical methods employed in our research. We found it necessary to generate this simulation ourselves because, despite conducting an extensive review of the existing literature, we could not find an established framework capable of simulating expression patterns in relation to delineated spatial areas - which is what we aim to identify. Moreover, such a framework did not exist for simulating increasing degrees of noise and randomness. To be able to quantify the degree of noise or randomness in a simulation, however, was essential for benchmarking various evaluation metrics to identify randomness as such. For a more detailed description of how we generated the simulated expression data, please refer to the methods section of the revised manuscript and Supplementary Figure 9 and 10. In this section, we use the large necrotic center of sample UKF313T as our example.

In both cases we simulated a data set of 63,600 expression variables across the following simulation modalities:

Pattern: We integrated six different kinds of patterns to account for possible variations among them. Three followed a descending expression with increasing distance with differing degree of steepness and three followed a peaking pattern with differing sharpness of the peak. We computed the distance of each barcoded spot to the outline of the necrotic center and binned the spots with a bin width that corresponded to the highest possible resolution, namely 100um. Each bin was assigned a number corresponding to its position when aligning the bins based on their distance to the annotation. This index was used to assign expression values to the

spots of each group such that different pattern emerged as displayed in Figures 3.1 and 3.2. As we only screened the environment of the necrotic center the values within its core did not have any effect on the screening. Spots inside the core of the annotation obtained randomized expression in a range of 0 - 0.25.

Noise Type (NT): In spatial biology it is not only the degree of randomness that affects the quality of a spatial pattern but also the spatial pattern of the noise itself. We hypothesized that differently emerging noise can affect the visibility of different pattern differently. We thought of the following four types from which noise could emerge:

- Equally Distributed (ED): Noise is applied to each individual barcoded spot with increasing percentage, gradually replacing the simulated expression values, but without any spatial aspect.
- Equally Punctuated (EP): Noise is selectively applied to a growing number of spots.
- Focally Punctuated (FP): Noise is selectively applied to spots that lie closely together, resembling other spatial niches that are not affected by the overall pattern.
- Combined (CB): All of the three previously mentioned noise types were applied in combination.

Noise Percentage (NP): In each simulation, a certain degree of noise was introduced, represented by randomly generated expression values. The resulting simulated variable was a combination of pattern-specific expression and randomly generated expression, based on one of the four noise types. The sum of these influences always added up to 100% of the simulated expression. Therefore, in case of 100% noise the simulation corresponds to complete randomness. The impact of increasing noise on the simulated pattern, depending on the noise type, is illustrated in Figures 3.3 and 3.4.

To perform simulations, we considered every possible combination of patterns (6 variations) and noise types (4 types). These simulations were conducted at incremental noise levels, ranging from 0% to 100%, with a step size of 2% (51 levels in total). For each combination and noise level, we performed 50 iterations using different random seeds. This extensive process resulted in a total of $4 \times 6 \times 51 \times 50 = 61,200$ simulations. To facilitate tracking simulation details, each simulation was named following the syntax: *SE.pattern.noise type.noise percentage.iteration*.

- SE: A prefix indicating simulated expression.
- pattern: An abbreviation denoting the specific pattern used (in capital letters).
- noise type (NT): The type of noise applied.
- noise percentage: (NP) The percentage of random values contributing to the expression.

- iteration: An index corresponding to each iteration, generated using `set.seed()` with `seed = 123 * i`, where `i` ranges from 1 to 50.

It's worth emphasizing that the suffix, such as `*"l.1,"` corresponds to the index of a seed used to generate the noise. Therefore, in the case of 100% noise, all pattern simulations under the same random seed end up in the same noisy pattern regardless of the hidden pattern. This is because at a noise ratio of 100% the expression from the pattern did not have any influence on the final simulated expression.

Additionally, noise patterns such as *Equally Punctuated (EP)* and *Focally Punctuated (FP)* are seldom encountered in real-life scenarios. It's basically impossible to come across gene expressions that exhibit a flawless pattern of smoothness throughout the entire sample, only to be sporadically disrupted by individual outlier points or clusters of outlier points. Nevertheless, we deemed it valuable to simulate these noise patterns since *Combined* noise pattern amalgamates all of them. In our perspective, *Equally Distributed (ED)* and *Combined (CB)* noise are the types most likely to be encountered in real-world data. Therefore, during our benchmarking process, we placed particular emphasis on evaluating this type of noise pattern. We acknowledge the theoretical implications of the other two patterns to pinpoint and underscore potential weaknesses of our proposed algorithm.

Figure 3.1: Visualization of the six expression pattern in space simulated at 0% noise percentage.

Figure 3.2: Visualization of the six expression pattern at 0% noise percentage as inferred gradients.

Figure 3.3: Simulated gene expression pattern with increasing amount of noise as displayed on the surface. The black polygon corresponds to the outline of the necrotic center.

Figure 3.4: Simulated gene expression pattern with increasing amount of noise when inferred as gradients. Points correspond to the predicted expression. The green line is used as a smoothed visualization of the pattern. The black vertical line corresponds to the outline of the necrotic center (at 0mm distance).

Section 4, “Spatial Gradient Screening Revised”:

Motivated by the valuable comments of reviewer 2, we have undergone a comprehensive revision of our statistical approach in Spatial Gradient Screening. While overall the concept remains intact, we have refined our approach by anchoring it in a hypothesis-driven framework and delineating the process into three distinct steps.

Notably, among the three steps encapsulated in Spatial Gradient Screening, only the initial one—termed “Inferring an Expression Gradient”—varies between Spatial Annotation Screening and Spatial Trajectory Screening. Subsequent stages maintain identical mathematical, statistical, and informatic procedures. Therefore, steps two and three are illustrated using only an inferred expression gradient from Spatial Annotation Screening.

(Supplementary Figure 7 represents a visual glossary of terms that are frequently used when discussing spatial annotations and spatial trajectories.)

4.1 Step 1: Inferring the Expression Gradient

With *inferring a gene expression gradient* we mean estimating how the expression levels of a particular gene change in relation to a spatial feature, such as along a trajectory or depending on the distance to a spatial annotation’s outline. This encompasses several steps.

4.1.1 Calculating the Distance

First, we calculate the distance of each barcoded spot to the relevant spatial feature. In case of spatial trajectories, all barcode spots within the trajectory rectangle are projected onto the trajectory's vector by calculating vector \mathbf{C} that connects the origin of trajectory \mathbf{T} to the barcode spot of interest. Then, \mathbf{C} is projected onto \mathbf{T} such that projection \mathbf{P} corresponds to:

$$\mathbf{P} = \frac{\mathbf{C} \cdot \mathbf{T}}{|\mathbf{T}|}$$

The magnitude of the vector \mathbf{P} corresponds to the projection length

$$Dist = PL = |\mathbf{P}|$$

and the projection length in turn corresponds to the distance along the trajectory. In case of spatial annotations, each barcoded spot is projected to its closest vertex of the polygon forming the outline of the spatial annotation and the distance is computed. The results of this step are displayed in Figure 4.1.

Figure 4.1: Illustration of computed distance values (a) along a spatial trajectory or (b) to the outline of a spatial annotation.

4.1.2 Distance-Based Gene Expression Smoothing

After obtaining the distance values, the gene's expression levels of each barcoded spot are related to the corresponding distance. Then, locally weighted scatterplot smoothing (LOWESS or LOESS) is used to fit a curve that approximates the changes in gene expression along this distance. This process is visually represented in Figures 4.2 and 4.3. The α parameter for this loess fit, which determines the degree of smoothing, is standardized and calculated as follows:

$$\alpha_{\text{loess}} = \frac{\text{Resolution}}{\text{Distance} * \text{CF}}$$

Where:

- *Resolution* stands for resolution and defaults to the average minimal center-to-center distance (CCD) of the data points. In the case of regularly fixed data points, as with Visium's barcoded spots the value is given. For irregularly scattered data points, as is the case for single cell based platforms, this value is computed.
- *Distance* indicates the total distance covered in the screening.
- *CF* stands for correction factor and is computed as the ratio of data points available from the amount of data points required to call the data set complete. See Supplementary Figure 7.

Therefore, the alpha parameter is inversely proportional to both the resolution of the platform and the distance screened. In other words, as the resolution increases or the distance screened becomes longer, the fit allows more details.

Figure 4.2: Visualization of two example genes for Spatial Trajectory Screening (a) and for Spatial Annotation Screening (b)

Figure 4.3: Gene expression patterns along a trajectory (a) and in relation to the distance from a necrotic center (b). Black points represent gene expression for each spot, while curves from the loess fitting reveal spatial trends as a function of distance.

4.1.3 Generating the Inferred Gradient

With *inferred gradient* we refer to a numeric vector of expression estimates capturing the resulting pattern of the fitted curve. To obtain it, we utilize the loess model for gene expression estimation along the distance with uniform intervals. The level of detail in the inferred gradient depends on the number of expression estimates (NEE) it comprises. The number and position of the expression estimates is calculated by averaging the distance values within predefined distance bins, using a binwidth that corresponds to the spatial screening resolution. For instance, if the distance equals 3 mm and the resolution 0.1 mm the number of expression estimates equals 30. Figure 4.4 visualize the expression. Finally, the expression estimate vector is standardized to a defined range (we use 0-1, equivalent to 'low' to 'high'). When plotted against their corresponding distance values, the expression estimates form a polygonal curve representing the pattern of the inferred gradient, as shown in Figures 4.4 and 4.5.

estimates via vertical lines intersecting the curve from the loess fit at equal intervals.

Figure 4.4: Illustration of the expression estimates displayed by the vertical lines intersecting the curve at equal intervals obtained by the loess fit for both examples, spatial trajectory screening (a) and spatial annotation screening (b).

Figure 4.5: Illustration of inferred expression gradients and their patterns along a trajectory (a) and concerning the distance to a necrotic center (b). Note that compared to the previous plot the values are rescaled to 0-1.

4.2 Step 2: Identification of Non-Random Gradients

The second step in Spatial Gradient Screening is focused on the identification of genes whose expression gradients exhibit a pattern that is unlikely due to randomness. Depending on the algorithm used, Spatial Annotation Screening (SAS) or Spatial Trajectory Screening (STS), for every inferred gradient we posit the null hypothesis:

“There is no discernible relationship between gene expression and the spatial reference feature used to infer the expression gradient. Gene expression levels are distributed randomly with respect to their spatial direction or relative distance.”

Correspondingly, for every inferred gradient we formulate the alternative hypothesis:

“A non-random relationship exists between gene expression and the spatial reference feature used to infer the expression gradient. Gene expression levels are not randomly distributed with respect to their spatial direction or relative distance but form a recognizable pattern that might be of biological interest.”

To be able to adopt either of these hypotheses we posit that, if the inferred gradient in question stems from a gene whose expression pattern depends on the distance to the annotation, it should **not** merely consist of randomly scattered expression values. Instead, it should display a discernible degree of gradual expression change forming a recognizable pattern. Thus, the smoother the inferred gradient, the less random it is, and vice versa. Assuming this relationship, we aimed to quantify the degree of randomness of a gradient. For that, we used the total variation (Supplementary Figure 8c).

4.2.1 Quantifying Randomness of a Gradient

To quantify the degree of gradual expression or, conversely, the degree of randomness in an inferred expression gradient, we calculate its total variation by considering the absolute differences between adjacent expression estimates according to:

$$v = \sum_{i=1}^{n-1} |y_{i+1} - y_i|$$

Where:

- v is the total variation.
- n is the number of expression estimates in the gradient.
- y_i represents the gene expression value at expression estimate i .
- $|y_{i+1} - y_i|$ calculates the absolute difference between the gene expression values of adjacent expression estimates.

To assess the effectiveness of the total variation (v) in capturing the level of noise or randomness introduced into a gradient representing a perfectly smooth pattern, we leveraged the simulated dataset discussed in Section 3. We examined the relationship between the total variation and the level of randomness on average (Figure 4.6), as well as across different underlying patterns and various types of expected noise.

Our analysis revealed two key observations when dealing with peaking patterns as opposed to gradual increases or declines. Firstly, we noted that patterns deviating from a linear ascending gradient exhibited a slightly higher baseline total variation at 0% noise. However, these differences in baseline total variation diminished when the noise percentage exceeded 25%, as illustrated in Figure 4.6. Additionally, peaking patterns demonstrated a relatively higher susceptibility to focally punctuated noise. Despite these nuances, our findings consistently demonstrated a strong linear relationship between the total variation and noise ratio across all simulation modalities. The high R^2 values of 0.78-0.91 (Supplementary Figure 11 b & e) in the noise types *Equally Distributed* and *Combined* (which we consider the most realistic manifestation of noise in real life data) indicated that this metric effectively quantifies the degree of randomness.

We observed that the correlation coefficient R^2 between integrated noise and total variation differed across various screening setups. Notably, an increase in resolution typically led to a decrease in R^2 . To accommodate this variability, we advise users to set `estimate_R2` to `TRUE` when utilizing the algorithm. This setting activates a simulation tailored to the user's dataset, estimating the R^2 for that specific run. This estimated R^2 serves as an indicator of the reliability of the conducted screening.

Figure 4.6: Relationship between the noise percentage and the total variation score across noise types. Each point represents a simulation. Colors indicate the baseline pattern that is affected by noise. The adjusted R2 displayed is the mean of all adj. R2 across the pattern under a specific noise type.

4.2.2 Obtaining the p-value

Given that the total variation has a strong overall relationship with the degree of randomness in an inferred gradient, we define the p-value corresponding to the hypothesis presented above as the probability of observing an equal or lower TV than the gradient in question under entirely

random conditions. To achieve this, we simulate completely random expression gradients by assigning randomized expression values to all barcoded spots in the screened area and proceed to infer the expression gradient. We repeat this simulation process n times and calculate the p-value according to the following formula:

$$\text{p-value} = \frac{\sum_{i=1}^n I(v_{r_i} \leq v_o)}{n}$$

Where:

- p-value is the p-value.
- v_{r_i} represents the total variation for the i -th simulated gradient under complete randomness.
- v_o is the total variation of the observed inferred gradient from the gene of interest.
- n is the number of random simulations, with a default of 1000.
- $I(\cdot)$ is the indicator function that equals 1 if the condition inside the parentheses is true and 0 otherwise.

The default value for n is 1000, but users are free to define n as long as it is equal to or higher than 100. (Increasing n beyond 1000, however, did not result in significant changes during our simulations, except for an increase in runtime.)

Supplementary Figure 11 provides a comprehensive visualization of the relationship between noise levels and p-values across all simulation modalities. In the case of equally distributed noise, we observed a consistent and stable relationship between p-values and noise ratios regardless of the underlying pattern, highlighting that the mentioned differences in TV baseline across simulated pattern did not have an effect. Figure 4.7 provides a visual representation of the averaged relationship between the resulting p-values and the percentage of noise. Although there are slight differences depending on the underlying pattern the relationship appeared robust.

We noticed that discernible effects of equally distributed noise tend to manifest in the pattern once a noise level of 30-40% is reached. Consequently, the p-value assumes a notably stringent character, considering that the significance threshold of 0.05 begins to be consistently exceeded when the noise ratio approaches approximately 60-65%. This kind of p-value is comparatively stringent. Yet, it aligns with our overarching objective, which is to offer researchers a robust and statistically sound framework for interpretation. Our intention is to facilitate intuitive analysis and interpretation, sparing researchers the need to repeatedly filter extensive lists of genes.

Figure 4.7: Relationship between computed p-values and percentage of noise quantifying the degree of randomness. Note the differing scales on the y-axes. Horizontal lines display the significance threshold of 0.05. In panel (a) each point corresponds to a simulation colored by the baseline pattern. Its position is determined by the noise ratio of the simulation and the resulting p-value. In panel (b) points are put in the rear and lines are used to approximate the relationship between the noise percentage and the resulting p-values across the six different patterns, showing no significant differences.

4.3 Step 3: Identifying Biologically Interesting Pattern

The third and last step aims to identify non-random gradients that correspond to specific biologically interesting patterns. Each inferred gradient is compared to predefined models that reflect abstractions of biological dynamics to aid in interpretation. Of course, gene expression patterns can be complex. But in case of distance dependent pattern, we think that the most important ones can be simplified into the following ones:

Association Pattern: Gene expression is higher close to the annotation and decreases with distance, indicating an association. For instance, an association between the necrotic zone and hypoxic gene signatures is described via a *descending* pattern, for which Figures 3.1 (top) and 3.2 (top) show examples.

Recovery Pattern: Gene expression is decreased close to the annotation but recovers as distance increases. For instance, the fact that oxygen-based metabolism is decreased around the necrotic area and increases with distance to it is revealed with the predefined *ascending* models.

Layered Pattern: Gene expression transiently increases at a certain distance after which it decreases, contributing to a layer-like organization. We refer to genes following these patterns as *peaking*. The simulated pattern in Figures 3.1 (bottom) and 3.2 (bottom) exemplify this.

Predefined models allow precision in addressing specific research questions and provide an intuitively interpretable framework where findings can be embedded in biological questions revolving around the environment of spatial niches. If an inferred gradient is positively evaluated as non-random, indicated a significant threshold set by the user (default FDR < 0.0), it is submitted to comparison against models, pre-defined by the user, that abstract biological dependencies of interest. In SGS, we use the term *models* to refer to numeric vectors of the same length and of the same range as the inferred gradient vector of expression estimates that encapsulate the pattern of a gene of interest. The goodness of each gradient-model fit is evaluated by three evaluation metrics: Pearson Correlation (PC), Mean Absolute Error (MAE), and Root Mean Squared Error (RMSE).

Figure 4.8: Illustration of the models used for the simulations regarding the necrotic center of sample UKF313T.

4.3.1 Metric Presentation

This section displays the three metrics we included in the benchmarking, along with the characteristics they are known for.

4.3.1.1 Mean Absolute Error (MAE)

The metric is illustrated in Figure 5.2 and expressed in the formula¹:

$$MAE = \frac{1}{n} \sum_{i=1}^n |G_i - M_i|$$

¹ (In all formulas, G corresponds to the numeric vector that contains the inferred gradient and M corresponds to the numeric vector that contains the modeled gradient.)

It features the following characteristics:

- Evaluates the average absolute deviation between observed and model values.
- Robust to outliers, making it suitable for data with extreme values.
- Offers a straightforward, unit-matched interpretation.

4.3.1.2 Root Mean Squared Error (RMSE)

The metric is expressed in the formula:

$$RMSE = \sqrt{\frac{1}{n} \sum_{i=1}^n (G_i - M_i)^2}$$

It features the following characteristics:

- Similar to MAE but emphasizes larger errors due to the square term.
- Considers the magnitude of errors, valuable for identifying significant deviations.
- More sensitive to outliers, with extreme errors having a greater impact.

4.3.1.3 Pearson Correlation (PC)

The metric is expressed in the formula:

$$PC = \frac{\sum_{i=1}^n (G_i - \bar{G})(M_i - \bar{M})}{\sqrt{\sum_{i=1}^n (G_i - \bar{G})^2 \sum_{i=1}^n (M_i - \bar{M})^2}}$$

It features the following characteristics:

- Measures linear relationships, indicating both direction and strength.
- Provides values ranging from -1 to 1 for intuitive interpretation.
- Valuable for identifying overarching trends and patterns. Unlike MAE and RMSE, PC also differentiates between *no relationship* and *inverse relationship* between the inferred gradient and the modeled one.

4.3.2 Metric Benchmarking

Motivated by reviewer 2 we conducted a sensitivity to noise analysis and a general benchmarking of all three evaluation metrics. In essence, we sought to identify the metric that provided the most accurate quantification of a simulated gradient's deviation **from its original pattern**, as influenced by varying degrees of noise. How well each metric performed predicting

the degree of deviation from the original pattern due to increasing noise was again evaluated using R2.

In general, we observed trends shared by all metrics. First, we noted that all metrics had their best score when evaluating a simulated expression with zero percent noise and all metrics showed a continuously worsened evaluation with growing noise levels. Second, we noted that there were only slight differences between the evaluation noise ratio of the descending and peaking models. RMSE (Root Mean Squared Error) performed better for descending models and Mean Absolute Error (MAE) performed better in patterns resembling a peak.

However, with an R2 of 0.81 averaged across all simulation modalities (4.10 a), it became evident that generally speaking both metrics perform equally well. In contrast, Pearson correlation did not achieve comparable R2 values, primarily because it struggled to detect the presence of increasing noise levels up to 50%. Furthermore, it showed high variance when confronted with punctuated noise highlighting its sensitivity to outliers. All in all, we decided to use RMSE as the primary metric to evaluate a gradient-model fit when looking for descending or ascending pattern and MAE when looking for peaks.

Figure 4.9: Visualization of the Relationship Between Noise and Evaluation for all three metrics. Each point in the scatterplots represents the evaluation of a simulation when compared to its baseline model. The x-axis displays the noise percentage of each simulation, while the y-axis shows the evaluation score of the inferred gradient when compared to the model representing its original pattern.

Figure 4.10: Panel (a) displays the average metric benchmarking results and merges all subplots. Panel (b) displays the subplot for the expressions using the linear descending model and noise type equally distributed as the baseline. The position of three example simulations is displayed by three separate additional shapes.

Figure 4.11: The inferred gradients of the three example simulations highlighted in the scatterplots above. Black points correspond to the expression estimates of their inferred gradient. The green line represents the pattern of the inferred gradient, while the blue line corresponds to the optimal linear descending model, which served as the baseline model for these simulations. Additionally, the p -value associated with the randomness hypothesis is displayed, along with the evaluation scores when compared to the linear descending model. Note their noise levels of 20%, 50%, and 80%, respectively. As the noise level increases, the evaluation scores deteriorate. The evaluation scores correspond to the position of the simulation in the previous scatterplots.

Reviewer #1:

In this manuscript by Kueckelhaus et al., the authors describe two new algorithms added to SPATA2, an open-source R-based bioinformatics framework for spatial transcriptomics data analysis, they developed earlier. The addition of the two new algorithms (Spatial Trajectory Screening (STS) and Image Annotation Screening (IAS)) allows for exploration of gene expression patterns within tissue microdomains and at cellular level within the native histological architecture. The authors demonstrate the use and functionality of the upgraded

and improved SPATA2 on 10x Visium samples from two brain disease models: malignant brain tumor (glioblastoma) and traumatic brain injury. Using these new features of SPATA2, the authors convincingly show its capability to identify gradually varying gene expression patterns between histologically distinct regions, including normal brain areas, adjacent transition areas and tumor core. This is certainly an important feature of spatial transcriptome profiling, as it allows for a comprehensive interpretation of gene expression changes (abrupt or gradual) between these areas. Additionally, the analyzes might provide valuable insights regarding disease progression and prognosis. Spatially segregated marker gene expression analysis within the transition zone, tumor core and normal brain tissue, confirmed the boundaries set by histological analyzes of the tissue. The authors identify two limitations to the STS algorithm: can screen in only one direction at a time and it has low flexibility due to linear and rectangular shape of the area of interest. To overcome this limitation, the authors introduced IAS as an additional algorithm, which allows screening the local surrounding of an image annotation. The authors show the utility of this tool, when analyzing the necrotic area surrounding the glioblastoma. Because of IAS' flexibility in defining the area that marks the microstructure of interest and the surrounding area to be screened, this tool could be useful in other research areas of neurobiology of disease. For instance, these tools could be used in studies of gene expression changes following ischemic stroke or amyloid plaque deposition and progression of neuroinflammation. Analyzing the gradient of gene expression depending on the distance from damaged area or plaque deposition, could provide insights into biological processes that predict disease progression and/or outcome of treatment strategies. Therefore, these tools have the potential to be used in translational/clinical research. A TBI model was used to demonstrate the utility of SPATA2 in the investigation of fine-grained transcriptome changes at the site of injury and in the surrounding brain areas. Using a single-cell RNA sequencing data set from the same TBI mouse model with multiple cell types from the brain, in combination with cell segmentation of the Visium H&E images, the authors were able to identify single cell transcriptome profiles, that showed cell type density gradients depending on the distance from injury. As a proof of principle, the cells that showed gradient density were identified based on marker genes as macrophages and monocytes. This manuscript has the strength of providing the research community with an open-source tool for in-depth analysis of spatial gene expression data. Although the two disease models that were used for demonstration purposes are brain disorders, I think the tool can be applied to other tissues as well. The authors should expand in the discussion the applicability. If there is a way to test on other tissue, it would make the paper stronger.

We thank the reviewer for the extensive and positive feedback on the presented algorithms. We fully agree that our algorithm can be applied to any kind on samples. Here we focus on

brain disease models but also showed novel features and applications to integrate multiple samples while we still stayed in brain diseased since this is the major focus of our research group. We further added the perspective that our model can be applied to any spatial related pathology in the discussion.

The authors did not mention in the manuscript whether SPATA2 is compatible with Visium only. It would be good to test if it is compatible with other techniques besides Visium, such as GeoMax.

We extend our gratitude to the reviewer for bringing up this important point. Originally, the software SPATA2 was developed for use with spatial transcriptomic platforms, hence the acronym SPATA, signifying **S**patial **T**ranscriptomic **A**alysis. However, we expanded the package to be compatible with a wide range of spatial biology platforms, provided that the data structure adheres to the following criteria:

1. The numeric variables under analysis should correspond to molecule counts, such as RNA read counts, metabolite counts, or protein counts.
2. A clearly defined observational unit must exist to which the numeric variables can be mapped. For example, the Visium platform's observational unit consists of barcoded spots, while the SlideSeq platform utilizes barcoded beads as its observational unit. In the case of the Xenium- or the MERFISH platform, the observational unit is the individual cell.
3. The observations must be equipped with x- and y-coordinates for analysis in two-dimensional space and should be equally distributed over the analyzed tissue.

We provide helping functions to initiate analysis via SPATA2 for the standardized data output of the following platforms:

- MERFISH: `initiateSpataObjectMERFISH()`
- SlideSeq: `initiateSpataObjectSlideSeq()`
- Visium: `initiateSpataObjectVisium()`
- Xenium: `initiateSpataObjectXenium()`

Lastly, the flexible function `initiateSpataObject()`, allows the user to initiate analysis from the output data of any other platform provided that it adheres to the aforementioned requirements.

Information around the platform used is stored inside the created `spata2` object and might decide on which features of the package can be used. (E.g. BayesSpace clustering is only

compatible with the ST or Visium technique.) The Spatial Gradient Screening algorithms are compatible with *all* platforms.

I recommend that in discussion the authors present potential clinical applications, including how this tool might aid drug discovery.

We sincerely appreciate the reviewer's valuable suggestion. As discussed in the revised manuscript, we believe that the framework encompassing spatial annotations, as exemplified by SPATA2, can serve as a catalyst for the development of novel algorithms. These algorithms focus on the integration of image-derived data with expression data, a concept notably explored by Hildebrand et al. in their work from 2021. Additionally, the explicit screening approach offered by SPATA2's spatial annotation screening enables the precise identification of biomarkers with spatial relevance. These biomarkers can play a pivotal role in the advancement of sophisticated histopathological classification methods.

Please also check for editing, small grammatical mistakes and incomplete sentences, such: "SPATA2 implements a variety of external algorithms and presents them in user-friendly wrapper functions that."

We thank the reviewer for his comments and performed editing of the manuscript.

Reviewer #2:

Kueckelhaus and colleagues propose that two classes of gene expression patterns are of interest in relation to histological features in tissues. One class is a set of unidimensional functions along a rectangle defined in an image. The second class is a set of bivariate functions gradually increasing (or decreasing) with the proximity to the outside of a polygon enclosing a region of interest. The authors argue that identifying genes whose expression closely follow these patterns are of greater interest than genes whose means change between histologically defined regions.

It is motivated why the authors find the linear rectangular gradients of the first class interesting. It is not extremely clear why specifically genes increasing in value on the outside of the polygon in the second class is biologically interesting.

We apologize for any prior lack of clarity. The revised introduction of the new manuscript addresses this concern in detail. We hope that this summary clarifies our approach and addresses the reviewer's question: Spatial biology aims to detect recurring spatial pattern. In histological microscopy, for instance, these efforts uncovered histomorphological structures like necrosis in gliomas, serving essential diagnostic purposes. Spatial multi-omic studies have

identified recurrent gene signatures that appear with high spatial segregation marking areas with specific functions or characteristics (Ravi et al. 2022). Whether delineated by prominent histomorphology or spatially segregated expression, we refer to these patterns as spatial niches. Particularly in case of pathologies, we envision these niches as a multifaceted array of factors, each potentially influenced by or influencing the architecture of the tissue in which they are located. We posit that dependencies exist between spatial niches and their immediate surroundings, giving rise to distinct expression patterns when analyzed in relation to their distance from these niches. To capture these patterns, we utilize spatial trajectories and spatial annotations as spatial reference features. When it comes to spatial annotations, our approach involves screening the area outside the polygon, effectively examining the environment surrounding the annotated region. We assume that by using these areas as references, we can uncover emerging patterns and gain valuable insights for our analyses. The idea of using spatial reference features is not entirely novel. The following shows two exemplary works that used similar approaches that can be found in the current literature. In the study conducted by Hildebrandt et al. (2021), a compelling example of the interplay between annotated histological structures and their environments emerges. The researchers investigated gene expression patterns as a function of distance to histologically ambiguous veins within liver lobules in mice. This way, they could classify these histologically ambiguous veins as either central or portal. (For the classification, however, they used genes previously identified by DEA. Their approach does not integrate a specific screening using reference models that capture pattern of continuous gene expression.)

For the sake of identifying genes following these patterns, the authors developed a piece of software that allows a user to interactively define rectangles for the first class, and polygons for the second class, in H&E stain images paired with 10x Visium gene expression measurements. Unlike typical statistical inference, here each gene's observed expression value is compared to a small finite set of manually pre-defined functions (there is no parameter inference to cover a continuous space of functions).

We appreciate the reviewer's concern and would like to clarify our rationale for using predefined models: The primary objective of Spatial Gradient Screening is to provide researchers with a framework that accomplishes two key goals. First, it enables the integration of extensive gene lists identified in preceding steps into spatial hypotheses. Second, it presents results that are intuitive to interpret, shed light on potential relationships, and allow the incorporation of recurring spatial niches into a broader context. Predefined models serve as a means to accurately represent the spatial dependencies of interest, offering a user-friendly approach to integrate these dependencies during the screening process.

Certainly, it's essential to acknowledge a notable limitation of this approach. Depending on the data set and research goals, predefined models may not capture nuanced or unexpected patterns present in the data. The algorithm's reliance on predefined patterns can restrict its ability to uncover unforeseen relationships. Consequently, the default models that SPATA2 provides may not always be suitable for all research inquiries. We suggest the following to deal with this limitation:

- First, we've ensured that users have the flexibility to expand the collection of models individually. For example, Figure 5.1 illustrates the integration of multiple peaking models in cases where a multiple-layer architecture is anticipated. We provide helper functions that allow to form the models according to the expected pattern.
- Second, the results obtained in step 2 can be used to identify genes with non-random expression patterns that may not align with any of the predefined models. Even if the evaluation of each gradient-model fit from the predefined-models do not meet the threshold, collectively they can facilitate subsequent unsupervised clustering approaches to identify patterns that were unexpected but shared among several other genes.

Figure 5.1: Individually created peaking pattern for cases in which the default models used by SPATA2 do not suffice.

Fit between the function and the observed values is evaluated using an average of Pearson correlation and something the authors refer to as 'Residuals Area Over the Curve' (and 'Residuals Area Under the Curve'). These RAOC/RAUC terms appear to be a new definition by the authors, since there is no reference, it is not standard terminology in the field, and Google searching these terms only returns the website for the tool presented in this manuscript. The definition of RAUC in equation (5) is inscrutable. What is the residual of? What

is the function f ? Why is this fitness measure of interest rather than something more standard such as root mean squared error or mean of absolute errors, or some statistical likelihood?

We apologize for any confusion stemming from our initial suggestion of the RAOC evaluation metric. Upon reviewing the concern, it became evident that RAOC bears a striking resemblance to the widely recognized Mean Absolute Error (MAE) metric, as displayed in Figures 5.2 and 5.3. We recognize and apologize for this oversight, attributed to our limited experience in modeling spatial gradients at the time.

After our benchmarking of the evaluation metrics Pearson Correlation (PC), Mean Absolute Error (MAE) and Root Mean Squared Error (RMSE), we decided to use RMSE as the default evaluation metrics given its strikingly high R^2 when used as a predictor of simulated deviations from a ground truth pattern. We'd like to thank the reviewer for suggesting these metrics and hinting at the flaw in our approach.

Due to the resemblance between MAE and RAOC, We decided to remove RAOC from the metrics with which to evaluate gradient-model fits and replace it with MAE. The following figures visually illustrate the concepts and underscore the strong similarity between MAE and RAOC, explaining their consistent outcomes.

To compute the RAOC, we initially plot the residuals (errors in terms of MAE) against the expression estimates and calculate the area under the resulting curve. This area value is then normalized by dividing it by the number of expression estimates to obtain a range of 0-1. We then calculate the residuals area over the curve (RAOC) by subtracting the RAUC value from 1. Due to the striking resemblance to MAE, RAUC (or RAOC) is no longer used.

Figure 5.2: Assessment of a gradient-model fit through MAE: Panel (a) shows relatively low errors per bin due to the good fit of CSTB and the 'Linear descending' model. The opposite is the case for Panel(b) where the fitted model corresponds to a peak.

Figure 5.3: Residuals (errors) are plotted against the distance bins. The residuals area over the curve (RAOC) is displayed as the green area and, conversely, the residuals area under the curve (RAUC) is displayed in red.

Throughout the manuscript, the authors report P-values for the combined RAOC-Pearson fitness measures. It is not described how these P-values are obtained, and it is unclear how that would even be possible with the odd RAOC component.

We extend our apologies for any confusion regarding the inclusion of the RAOC metric. The reported p-values were previously derived using Pearson Correlation analysis between the inferred gradient and the respective model, employing the `stats::cor.test()` function. We acknowledge the statistical flaws of our prior approach. We have completely abandoned the RAOC metric. Furthermore, we have identified MAE and RMSE as the preferred metric for assessing goodness of a gradient-model fits. It is important to note that the p-values we now obtain are no longer associated with step three, the model fitting stage. Instead, they pertain to step two, where gradients are identified that cannot be attributed to randomness. How we obtain these p-values and how they are related to alternative- and null hypotheses is explained above.

In all, the authors wanted to find genes following certain patterns, and wrote an approach to do this, which is presented with some examples in this manuscript. There are no discoveries, no conclusions, no insights, nor confirmations, claimed by the authors. Only a large number of reported observations without connection to any particular hypothesis or question.

It is apparent that our initial presentation has fallen short in conveying our motivation and the overarching goal of our proposed approach to the reviewer. We sincerely hope that the revised manuscript, coupled with our responses addressing the reviewer's concerns, will provide a

clearer and more comprehensive understanding of our research ambitions and the value it brings to the field. We have devoted considerable effort to restructure our manuscript and refine the presentation of our approach.

There are no methods to compare the authors' approach to, since they are likely the first to propose looking at these particular expression patterns. However, there is also not much in terms of investigation that would be typical for a new regression method. No investigation into sensitivity to noise. No investigation into false positive rates using simulated negative data. Etc.

We recognize the reviewer's critique regarding the absence of investigation into false positive rates and sensitivity to noise in our manuscript and fully acknowledge this oversight. We have undertaken an extensive investigation to assess the sensitivity of inferred spatial gradients to noise and their ability to account for randomness. This analysis is presented in Supplementary Figure 11. We observed that, provided an adequate number of expression estimates are included in the gradient, the evaluation metrics we selected exhibit a close linear relationship with introduced noise (randomness) as quantified by the R² between the total variation and the introduced noise. Given that we found variations in this R² value depending on the circumstances of the screening (distance screened, resolution, shape of the outline) the algorithm runs the same simulation conducted for this review with every screening run providing a measure of reliability of the screening set up.

Supplementary Figure 6d-g and Supplementary Figure 13a-f visualize our results regarding false positive and false negative rates as well as the impact of variations introduced by human error. For one thing, we found a constant false positive rate of 2-3%, where inferred gradients of expression variables that consisted only of randomly assigned expression values were identified as non-random. The false-positive rate, in turn, was initially less than 1%, however increased with progressively increasing variation introduced in the reference feature used. Although we found the degree of variation required to cause significant false negative rates to be beyond what is to be expected from human error, we highlight this fact in the revised discussion and emphasize the importance of thorough decision making when placing these reference features.

Minor issues:

How did the authors define the 'tumor', 'transition', and 'infiltrated cortex' classes in the glioblastoma data?

We employed the spatial segmentation tool integrated within SPATA2 to interactively define three distinct regions based on histomorphological features. Tumor regions were characterized by their high cellular density (Figure 5.4 a). The transition zone exhibited a lower cellular density compared to the tumor (Figure 5.4 b), while the infiltrated cortex was identified by its lowest cellular density (Figure 5.4 c). To validate our classification, we performed copy number alteration inference, which revealed significantly increased variations in chromosomes typical of glioblastoma (Main Figure 1f-j). Additionally, we conducted a differential expression analysis (DEA) using the defined groupings, further confirming our classification by detecting upregulation of EGFR in the tumor zone and SNAP25 in the infiltrated cortex zone.

Figure 5.4: Visualization of critical regions within histology for histological classification.

On line 401-401 “Figure 6e-j”, it is unclear if these example genes refer to genes that correlate with Hmox1 / Lcn2 levels, or if they are genes that follow distance based gradients.

We apologize for the lack of clarity. In the screening we used the inferred gradients of these two genes as additional models along with our basic models. Therefore, the screening could identify genes with gradients similar to the gradient of Hmox1 (e.g. Ctsb1, C1qa, Cd8, Main Figure 3e) and similar to the gradient of Lcn2 (e.g. Dhrr1, S100a6, Metrn). We specifically address this in the revised method section describing the analysis steps of the mouse cortex sample.

Figure numbering for supplementary figures seems wrong?

There are several cases where figures in the manuscript appear to be covering the main text, making it impossible to read. For example, line 247 ends with starting a new sentence, but it is not clear where the continuation of this sentence is.

We apologize for any oversights. We have conducted a comprehensive review of figure numbering and placement in the revised manuscript. As a result, there should be no misplaced sentences, figures, or figure numbers remaining.

Reviewer #3:

In this manuscript, the authors present two algorithms, namely Spatial Trajectory Screening (STS) and Image Annotation Screening (IAS). These algorithms aim to capture local gene expression signatures without the prerequisite of prior clusters. They have implemented these algorithms in a high-quality open-source R package. While the introduction of these algorithms is intriguing, there are certain concerns that need to be addressed.

We thank the reviewer for the helpful suggestions and comments.

1. The tissue examined in this manuscript demonstrates notable differentiation among the various histological regions. However, it remains uncertain whether this method can be effectively applied to H&E images lacking evident demarcations. For instance, consider the case of cancerous tissue (HCC.tif) depicted in the following link: <https://doi.org/10.6084/m9.figshare.21061990.v1>, in which the tumor regions are distributed in multiple regions.

We would like to express our gratitude to the reviewer for their insightful comment, which served as a pivotal motivation for us to conduct a more detailed analysis of sample UKF313T. This deeper analysis, in turn, prompted us to integrate multiple necrotic areas distributed across the entire sample into a unified analysis, significantly enhancing the biological insights gained from our study. First, it is essential to highlight that our concept of spatial annotations provides the flexibility to accommodate limitless annotations, each encapsulated within a single S4 object containing metadata and x- and y-coordinates outlining the spatial structure. Unlike data point grouping, annotations are stored independently.

As long as the screening areas of multiple annotations do not overlap, they can be seamlessly integrated and treated as a cohesive unit, as exemplified by the approach taken with the stab wounds in Main Figure 3c-d. In this case, two annotations delineated the spatial extent of the wounds, and during the screening process, both annotations were configured to infer gradients as a function of the distance to stab wounds in general. The screened area(s) did not overlap, however.

As illustrated in Main Figure 4, the presence of multiple annotations of the same type scattered throughout the tissue does not present a challenge, even if the screening areas overlap. If the annotations themselves do not overlap, they can collectively contribute to the screening. Main

Figure 4b demonstrates how the distance to necrotic areas and the inferred orientation of gene expression patterns consider all three annotations incorporated in the analysis. Figure 5.5 illustrates the overlapping screening areas.

Figure 5.5: Screening areas for all three annotations if they would be considered independently. The screening for each would include areas of the same kind, disturbing the inferred gradients.

2. The reliability and validity of the obtained results are subject to the accuracy and quality of the researcher's manual segmentation of the H&E image for spatial domain assignment. In the event of inaccurate or flawed segmentation, there may be concerns regarding the reliability of the findings. Therefore, conducting sensitivity analysis becomes essential to assess the robustness of the results and account for potential variations stemming from segmentation errors.

We sincerely appreciate the valuable feedback provided by the reviewer. In response, we conducted thorough investigations to assess the potential impact of bias introduced by segmentation errors or inherent variations in manual segmentation. Our objective was to understand how deviations in the manually drawn outline of a Spatial Annotation could affect the results of the analysis. To establish a reference point for our analysis, we utilized a simulated dataset based on the original necrotic outline, as described in Section 3. From the "Combined" noise category, we selected a subset of simulated expressions, totaling 600 per pattern, with a False Discovery Rate (FDR) below 0.00001 when screened with the original outline on which the simulations were founded. These 2400 simulated expressions served as our positive ground truth for assessment. For the negative ground truth, we generated 2400 completely random expressions by assigning each spot an expression value drawn from a uniform distribution. These randomly generated simulations constituted the negative ground

truth. The distribution of total variation in these respective simulations is displayed in Supplementary Figure 6g.

Supplementary Figure 6: Spatial Annotations and Investigation of Susceptibility to Human Bias. a) Demonstrates the concept of image annotations, where spatial outlines are manually generated through visual inspection of histomorphological features. b) Illustrates group-based spatial annotations, where outlines are created based on the positioning of data points within the same group. c) Presents numeric spatial annotations, where the expression of a selected feature forms the basis for spatial outline creation. d) Provides representative examples of outlines that progressively diverge from the original necrotic center outline. These examples were chosen from a collection of divergent outlines studied to investigate human bias. e) & f) Depict false positive and false negative rates in spatial annotation screening results with divergent outlines. Each point represents a run, positioned based on the degree of deviation from the original outline and corresponding test performance measures. The false positive rate remains nearly 0, while the false negative rate increases linearly with deviation from the original outline. Simulations with outlines deviating below 15% from the original outline exhibit no false negatives, indicating robustness within this range of deviation. g) Displays the distribution of total variation scores from the 4800 simulations used as ground truth for the screening runs in figures e and f.

To replicate segmentation errors and typical human biases in the form of deviations from manually drawn outlines, we introduced noise to the vertices of the polygon originating from the necrotic center annotation. These variations, a total of 26, each represented by a letter of the alphabet, progressively led to outlines that deviated from the original outline. We quantified the degree of deviation by measuring the overlap between the simulated noisy outline and the

original one. Representative examples of these diverging outlines are provided in Supplementary Figure 6d. Our primary objective was to assess how deviations in manually drawn outlines, influenced by human bias or inherent variability, might impact the analysis in terms of false positives (expression variables incorrectly identified as non-random) and false negatives (expression variables incorrectly identified as random). We subsequently conducted a fresh screening using the ground truth set of expression variables. We then plotted the outline's deviation used for the screening against the ratio of false positives and false negatives from the screening run, as depicted in Supplementary Figure 6e-f. Our observations indicated that the occurrence of false positives consistently remained low, ranging from 2% to 3%, and remained unaffected by the introduced outline deviations. However, when the deviation exceeded 20% from the original outlines, the rate of false negatives exhibited a linear increase with the introduced variations. In summary, there seems to be no risk of incorrectly identifying random variables as non-random due to randomized outlines. Nevertheless, the screening's ability to detect significant non-random patterns diminishes as the accuracy of the outlines declines. Although we believe that the degree of deviation introduced by human manual outlining does not exceed the 25% threshold, we emphasize the importance of precise placement of spatial reference features in our revised discussion. Overall, our findings demonstrate that spatial annotation screening exhibits robustness against this type of deviation.

3. The specification of the spatial trajectory in this study is user-dependent, which introduces three concerns.

Firstly, if users do not possess a priori knowledge or specific interests in the trajectory, is there a method available to generate latent spatial trajectories using algorithms? For instance, in scRNA-seq data analysis, can the cell lineage trajectory inference methods be employed?

We appreciate the reviewer's valuable suggestions. The generation of spatial trajectories relies not only on image and histological architecture but also on versatile considerations. Typically, a spatial trajectory comprises a start and an end point, with the exception of curved trajectories. These points can be defined based on various criteria beyond manual interaction with the image, as offered by the function `createSpatialTrajectories()`. For one thing, during manual interaction the *surface* of the sample can be colored by gene expression or cluster assignment which allows to connect areas based on continuous features and computational results. Additionally, start and end points can be programmatically set using the `addSpatialTrajectory()` function. For instance, researchers can use the centroids of adjacent spatial annotations as inputs, effectively connecting the spatial domains of these annotations and screening the areas in between.

Secondly, is the screening of spatial trajectories sensitive to slight variations, such as changes in the start point, end point, or width? Additionally, the authors established the spatial trajectory of sample #UKF269 with a width of 3mm and length of 5mm. Are there any guidelines or rationale behind these specification?

We sincerely appreciate the valuable feedback provided by the reviewer regarding this aspect. Based on our experience, particularly with sample UKF269T, we have observed that minor variations in the width and length of the drawn trajectory have a negligible impact on the method's output. To highlight the *linear* and *rectangular* nature of the screening process, we deliberately chose a width and length that resulted in a rectangular shape slightly smaller than the sample. This deliberate choice served to underscore that spatial trajectories enable the delineation of specific areas based on the rectangular region formed by the trajectory's length and width. Nonetheless, we acknowledge that we should have provided a more comprehensive explanation of our reasoning behind this approach and the robustness of trajectory variations. In response, we conducted sensitivity tests that encompassed variations due to the following three possible confounders: 1. starting and ending points (length) of the trajectory, 2. deviations in angles as well as 3. width of the screening area. Both, representative examples, as well as resulting test performance measures are displayed in Supplementary Figure 12. Equal to our investigations concerning Spatial Annotation Screening, as outlined in a section above, we established a baseline for our analysis by utilizing a simulated dataset derived from the original trajectory, alongside new simulations generated completely at random. This is visualized in Supplementary Figure 12a. Our primary objective remained consistent: to quantitatively assess the system's performance in accurately categorizing simulations as either random or non-random, even when the trajectory used for screening exhibited deviations from the original. The following sections will explore the sensitivity of Spatial Trajectory Screening (STS) to three potential confounding factors. While the effect on the false negative rate varied among these factors, deviations in each category only influenced the false negative rate when a significant degree of variation was introduced, which surpasses what we would consider realistic in terms of variation introduced by human error. Additionally, it's worth emphasizing that the false positive rate remained consistent. In simpler terms, what was initially simulated as random, consistently retained its random classification irrespective of alterations in the spatial trajectory used for the screening. However, despite providing evidence that spatial trajectory screening is robust against human error, we acknowledge in the revised discussion that careful consideration of the placement of spatial trajectories (and spatial annotations) is advised.

Figure 5.6: Distribution of total variation scores for simulations used for positive and negative ground truth.

Sensitivity to Variations of Start- and Endpoint

To quantify the extent of deviation resulting from alterations in the start and endpoints of the trajectory, as displayed in Supplementary Figure 12b-c, we measured the length of the resulting trajectory and subtracted this value from the original length. In our example, changes in the start and endpoints did impact the ability of STS to detect simulated non-random genes. This, however, was only the case when the changes resulted in a deviation of 1mm or more. The observed variance in the increase of the false negative rate suggested that the positioning of the start and endpoints also can play a significant role.

Sensitivity to Variations in Angle

To quantify the impact of variations in trajectory angles, we conducted a two-step analysis. Initially, we established the angle of the original trajectory as 90° (with 12 o'clock as the reference point). Subsequently, we calculated the differences between the angles of new trajectories and the angle of the original one. This method enabled us to gauge the deviation introduced by alterations in angle. Our analysis unveiled an influence of angle deviation on the false negative rate. As depicted in Supplementary Figure 12f-g, the rate of false negatives displayed a linear increase with angle deviation once it exceeded 10° . While we anticipate that realistic deviations introduced by human judgment are unlikely to surpass 10%, we acknowledge the potential noise factor associated with this kind of deviation during the screening process.

Sensitivity to Variations in the Trajectory's Width

Originally, we set the trajectory width at 4mm to ensure coverage of the entire tissue. While this finding might suggest opting for the widest possible width, caution is advised. Excessively wide trajectories can inadvertently include distortions, especially when spatial niches with high segregation are encompassed within the rectangle. A preliminary examination of spatial clustering results can provide insights into these niches, aiding in determining the optimal width. As displayed in Supplementary Figure 12f-g, the selected width influenced the false negative rate only slightly. Specifically, a narrower width resulted in a higher likelihood of missing non-randomly simulated genes. In general, our recommendation is to encompass as many spots as feasible. The default sets the width of the rectangle equal to the length of the spatial trajectory. This is to include enough points for the LOESS fit to work properly given that its alpha parameter (span, level of detail) depends on the distance to resolution ratio. For instance, a spatial trajectory of 5.7mm with a resolution set to 0.1mm would result in an alpha span of 0.018. For a LOESS fit to work with such a low span, a considerable number of data points is required which might not be provided if the rectangle is too small and excludes too many points. The correction factor adjusts the low span according to the share of missing data points which in turn results in less details in the inferred gradients. Therefore, the function defaults to a screening area of the same width and length.

Supplementary Figure 12: Investigating Susceptibility to Human Bias in Spatial Trajectory Screening. a) Depicts the distribution of total variation scores for positive and negative cases within the ground truth used for investigations, with surface and gradient plots illustrating representative examples from both simulation groups. b-c) Present the results of sensitivity investigations regarding variations in start and endpoint. b) Features representative examples, while c) illustrates the relationship between false positives and false negatives in relation to differences in trajectory length resulting from shifts in the start and endpoint. d-e) Showcase the results of investigations into deviations from the original trajectory by degree. d) Highlights representative examples, while e) illustrates the correlation between false positives and false negatives and the degree of deviation from the original trajectory. f-g) Demonstrate the relationship between false positives and false negatives and changes in the screening area resulting from different width inputs. f) Features representative examples related to the default screening area width (matching the trajectory length). g) Highlights the relationship between false positives and false negatives and deviations from the original trajectory screening, measured by the width parameter input. Note the y-axis scale changes in comparison to the other figures.

Thirdly, an important aspect to address is whether the trajectory can be a curve rather than strictly limited to a straight line. Exploring the possibility of curved trajectories is crucial for capturing more complex spatial relationships and patterns that may not be adequately represented by linear trajectories.

We appreciate the reviewer's suggestion. In response, we have incorporated the concept of curved trajectories into our approach, providing users with the ability to manually draw the trajectory instead of solely specifying start and end points. The revised method section now includes a detailed explanation of how to draw and project surrounding data points, as illustrated in Figure 5.7.

Figure 5.7: Illustration of a curved spatial trajectory.

4. On line 1030, page 31, the statement mentions that "as the trajectory had a length of 5mm, this resulted in 50 bins." Why "that the trajectory had a length of 5mm resulted in 50 bins"? And what is the difference of more or less bins?

We thank the reviewer for pointing out this potential weakness. In response, we have revised our spatial gradient screening approach as outlined above, and it is no longer accurate to speak of bins. In our previous submission we used to infer gradients by grouping data points based on their distance values into distance-bins with a certain binwidth. In the example you mention we used a binwidth of 100um. Given the length of the trajectory of 5mm and the binwidth of 100um (0.1mm) this resulted in 50 groups – 50 distance bins. After binning, we used to average the gene expression in each distance bin and then plotted the averaged expression against the position of each bin, resulting in the gradient. We noted, however, that this approach was susceptible to distance bins with only a few barcoded spots of extremely high or low expression values which in turn was highly susceptible to the chosen binwidth. As detailed above we now use a LOESS fit and estimate gene expression along this fit at certain distance intervals (expression estimates). This turned out to be more robust against these kinds of outliers. Figure 4.3b exemplifies this. After 3mm only few barcoded spots remain to form the pattern at this distance compared to the number of spots at distance 1mm to 2mm, for instance. This is due to the constraints of the tissue edge as displayed in Supplementary

Figure 7a-d. Using distance bins these few points would unproportionally contribute to the overall pattern. Our revised approach is more robust against this. We compared both approaches, estimating the R2 between our total variation metric and the introduced noise and consistently found that the approach using LOESS consistently resulted in better R2 than the approach depending on bins. Therefore, we only use the revised approach and the problem you brought to our attention is no longer an issue.

5. In the analysis of sample #UKF313, for a researcher not a pathologist, it is not an easy thing to identify the necrotic area. But the image annotation constitutes the important first step, it may hinder the applications of SPATA2 for non-pathologist. Have the authors considered employing computational methods for image annotation?

We extend our gratitude to the reviewer for this valuable suggestion, as it prompted us to position our approach within a broader concept and philosophy. In our introduction as well as in our analysis (Main Figure 5), we recognize that spatial niches must not necessarily derive from distinct histomorphological structures. Regarding computational methods for creating spatial annotations in samples lacking distinct histological spatially segregated areas, as exemplified by the necrotic zone in sample #UKF313, we have integrated the concept of image annotations into a larger framework that we term “spatial annotations.” In this context, a spatial annotation corresponds to an area featuring a pattern of interest, whether delineated by histomorphological structures like necrosis (Image Annotations, Supplementary Figure 6a), spatially segregated expression signatures (Numeric Annotations, Supplementary Figure 6b), or the presence of specific groups of data points (Group Annotations, Supplementary Figure 6c). We have already introduced Image Annotations, where the area of interest is manually delineated. For scenarios such as the one you brought to our attention, we refer to two computational approaches for creating spatial annotations, namely Numeric Annotations and Group Annotations. These approaches enable the integration of various computational methods by identifying data points that exhibit computed results, such as membership in clusters or high expression in gene signatures, and outlining their localization. This outlining process is not accomplished through manual delineation, as in the case of Image Annotations, but rather by leveraging the concaveman algorithm.

The concaveman algorithm initiates with the convex hull of the point set and iteratively refines it to attain a more concave shape. By forming triangles and excluding those containing original points, a polygon representing non-convex shapes is generated. This resultant polygon serves as the annotated area. To ensure spatially confined annotations, users can opt for the DBSCAN algorithm to identify spatial outliers that might distort the annotation.

A concise depiction of complete workflow is illustrated on the following pages.

Spatial Annotations from Grouping Algorithms

Here, we use sample #UKF275T, a glioblastoma sample featuring no significant histomorphological areas but a clear separation in clusters of data points as identified by BayesSpace clustering.

Figure 5.8: Panel (a) shows the histology of the sample featuring now prominent areas that could be annotated via ImageAnnotations. Panel (b) shows the data points (spots) as clustered by BayesSpace.

Assuming the researcher would like to conduct Spatial Gradient Screening from the perspective of cluster B2 the corresponding annotation would be a Group Annotation. Using concaveman, the outline of the space covered by cluster B2 can be captured.

Figure 5.9: Panel (a) shows the spots covering the area of interest as well as their computed outline. Panel (b) projects the surrounding on the created annotation.

Spatial Annotations from Expression Data

To generate spatial annotations from continuous data, like gene expression profiles, an initial step involves setting a threshold. While manual thresholding is an option, it can introduce bias. Therefore, we recommend using k-means clustering based solely on the expression levels. This creates two distinct groups: high and low expression. Subsequently, the same filtering procedure is applied. As an illustration, we consider the *Hallmark Hypoxia* signature, which often exhibits spatial segregation in glioblastoma samples.

Figure 5.10: Panel (a) shows the expression values of signature Hallmark Hypoxia. Panel (b) visualizes the division of spots based on their expression of the hypoxia signature in two groups.

Figure 5.11: Panel (a) shows the spots of high hypoxic gene expression. Panel (b) illustrates the identification of spatial outliers and the capturing of the area of the 'hypoxic niche'.

To annotate the spatial niche corresponding to this signature, we first need to identify the data points representing high expression. To circumvent the inclusion of individual data points

dispersed across the sample we integrate the DBSCAN algorithm which identifies spatial outliers. After removing outliers, the remaining spots undergo processing using the concaveman algorithm.

Figure 5.12: Panel (a) shows the outline of the annotated hypoxic niche and projects the surrounding on it. Panel (b) illustrates the gradients of the hypoxic gene expression from the perspective of the 'hypoxic niche'. Note, that in contrast to previous examples the expression from within the niche (it's core) is included in the gradient.

Spatial Annotations Using Individual Approaches

The versatility of SPATA2 extends to seamlessly integrating alternative approaches employed by users outside of the SPATA2 environment. In the context of SPATA2, we have designed three distinct types of annotations: Group Annotations, Image Annotations, and Numeric Annotations. These annotations represent high-level functions integrated into SPATA2, simplifying the implementation of our approach in the researchers downstream analysis called:

- `createGroupAnnotations()`
- `createImageAnnotations()`
- `createNumericAnnotations()`

From a programming perspective, all these annotations are derivatives of the S4 class `SpatialAnnotations`. This class, when utilized, mainly requires a `data.frame` containing the x- and y-coordinates apart from metadata like tags. These coordinates delineate the polygon necessary to outline the area of interest within the spatial data. Thus, if the user has identified specific areas of interest through external processes, he or she can effortlessly incorporate them into our framework using the function `addSpatialAnnotation()`.

6. During the downstream analysis of sample #MCI_LMU, it is stated that "Only genes with a p-value lower than 0.05 were included in the subsequent analysis and visualization steps." However, to effectively control the familywise error in multiple testing scenarios, it is recommended to use adjusted p-values instead of raw p-values. Therefore, the authors should consider incorporating genes with adjusted p-values below 0.05 in their analysis to ensure more rigorous statistical control and accurate interpretation of the results.

We apologize for any confusion in our previous presentation. To clarify, p-values we report in our analysis have always been adjusted, and this remains the case. When we referred to "p-values," we intended to convey "adjusted p-values" since all p-values in our study underwent adjustment via *false discovery rate (FDR)*. In the revised manuscript, we have taken care to consistently refer to these values as "FDR" to avoid any ambiguity. Detailed information on how we obtain these p-values is given in our response above:

7. In Supplementary Figure 4 (f)-(h), left pane, the meaning of the x-axis is not explicitly labeled. This lack of labeling makes it unclear whether it represents the trajectory direction or some other variable. In contrast, Supplementary Figure 3 (iii) clearly labels the x-axis as "Trajectory direction". To ensure consistency and clarity across the figures, it is advisable to label the x-axis consistently across all figures representing the same concept.

We appreciate the reviewer's feedback, and we have made the necessary adjustments to ensure consistent axis labeling in our package functions and manuscript figures. To enhance clarity and intuitiveness, we have added SI units for distance measurements to each x-axis of plots displaying an expression gradient, as we believe this is the most straightforward and easily understandable choice.

8. It appears that SPATA2 is specifically designed for use with the 10X Visium platform. However, it raises the question of whether it can be applied to other platforms commonly used in spatial transcriptomics, such as seqFISH, Slide-seq, Slide-seqV2, or Stereo-seq.

We extend our gratitude to the reviewer for bringing up this important point. Originally, the software SPATA2 was developed for use with spatial transcriptomic platforms, hence the acronym SPATA, signifying **S**patial **T**ranscriptomic **A**nalysis. However, we expanded the package to be compatible with a wide range of spatial biology platforms, provided that the data structure adheres to the following criteria:

4. The numeric variables under analysis should correspond to molecule counts, such as RNA read counts, metabolite counts, or protein counts.

5. A clearly defined observational unit must exist to which the numeric variables can be mapped. For example, the Visium platform's observational unit consists of barcoded spots, while the SlideSeq platform utilizes barcoded beads as its observational unit. In the case of the Xenium- or the MERFISH platform, the observational unit is the individual cell.
6. The observations must be equipped with x- and y-coordinates for analysis in two-dimensional space and should be equally distributed over the analyzed tissue.

We provide helping functions to initiate analysis via `SPATA2` for the standardized data output of the following platforms:

- MERFISH: `initiateSpataObjectMERFISH()`
- SlideSeq: `initiateSpataObjectSlideSeq()`
- Visium: `initiateSpataObjectVisium()`
- Xenium: `initiateSpataObjectXenium()`

Lastly, the flexible function `initiateSpataObject()`, allows the user to initiate the analysis from the output data of any other platform provided that it adheres to the aforementioned requirements.

Information around the platform used is stored inside the created `spata2` object and might decide on which features of the package can be used. (E.g. BayesSpace clustering is only compatible with the ST or Visium technique.) The Spatial Gradient Screening algorithms are compatible with *all* platforms.

Minors:

1. On line 635, page 21, there appears to be a typo in the sentence. The phrase "Additionally, barcode-spots. the local surrounding of image annotations..." seems to be missing proper punctuation and may contain an extra period. Please carefully check the full text.
2. Line 1034, page 34, "in which each row corresponds to ta gene-model fit ...", where "ta" may be "a"?

We apologize for any typographical errors and text inconsistencies that may have arisen due to misplaced figures. We have conducted a thorough review of the revised manuscript to ensure that such issues have been addressed and corrected.

3. To ensure clarity and proper referencing, it is recommended to include the analyzed dataset information in the general title of each main figure. This practice allows readers to immediately identify the specific dataset under investigation.

We appreciate the valuable suggestion from the reviewer. In the revised manuscript, we have added a descriptive title to each figure, including both main figures and supplementary figures. This enhancement aims to enhance overall clarity and comprehension.

Reference:

Hildebrandt, Franziska, Alma Andersson, Sami Saarenpää, Ludvig Larsson, Noémi Van Hul, Sachie Kanatani, Jan Masek, et al. 2021. "Spatial Transcriptomics to Define Transcriptional Patterns of Zonation and Structural Components in the Mouse Liver." *Nature Communications* 12 (1). <https://doi.org/10.1038/s41467-021-27354-w>.

Ravi, Vidhya M., Paulina Will, Jan Kueckelhaus, Na Sun, Kevin Joseph, Henrike Salié, Lea Vollmer, et al. 2022. "Spatially Resolved Multi-Omics Deciphers Bidirectional Tumor-Host Interdependence in Glioblastoma." *Cancer Cell* 40 (6): 639–655.e13. <https://doi.org/10.1016/j.ccell.2022.05.009>.

Reviewers' Comments:

Reviewer #1:

Remarks to the Author:

The authors answered all of my concerns and comments. The manuscript in the revised form is highly improved.

Reviewer #2:

Remarks to the Author:

In this nearly completely rewritten manuscript by Kueckelhaus and colleagues, the authors propose that screening for genes whose expression pattern across a tissue follow user-defined shapes is an advantageous approach to current spatial gene expression analysis strategies.

This workflow proposes users annotate polygons of biological interest based on paired images, then screen genes based on their expression as a function of the distance from the polygon.

The main text is structured such that anecdotes are used to argue for certain pattern choices in numerous settings, where patterns are created, and genes following the patterns are identified. There are some visual illustrations of genes that are identified with this strategy, but not with alternative strategies. The main text mostly serves as a demonstration of using the tool, though there are some comparisons that could warrant some more specifics about the results (listed in 'minor issues' below).

The main novelty is the proposal that genes should follow certain predefined patterns to be of interest. It is hard to argue either for or against this proposal. It would require some database of interesting genes and their patterns, and then see if they tend to follow these patterns. But this does not exist.

Secondarily, the authors have invented a statistical test based on total variation of a LOESS smoother. This ought to be compared in its statistical performance with classically known statistical tests.

Major issues ---

The null and alternative hypotheses described by the authors are the same as the null and alternative hypotheses that define the task of identifying spatially significant genes. The authors have transformed the original coordinate system to a 1-dimensional coordinate system reflecting distance to the annotated polygon, but existing methods such as e.g. SPARKX used by the authors will solve the same problem along the trajectory. Further, since this is now a 1-d problem, all methods that have been proposed to identify genes that vary along pseudotime trajectories also follow the same null and alternative hypotheses.

It would be reasonable to compare the novel statistical test defined here based on total variation with 1-dimensional application of spatial significance methods and pseudotime-dependent gene expression identification methods.

Minor issues ---

Around line 169, it is not clear how many genes do not reflect the corresponding regions, compared to how many that do.

Around line 216, it is not clear how many genes were not identified by clustering DE analysis but were detected by the trajectory analysis. How many known genes from the different categories of immune activity, immune cell types, migration, proliferation, and wound healing, beyond the few genes highlighted?

Around line 243, within what log2 fold change threshold are the changes considered 'nearly identical'?

Around line 274, is there a citation for the prior findings about TCR signaling near necrotic regions?

Reviewer #3:

Remarks to the Author:

The authors have addressed most of the concerns, however, there are still some issues that remain unclear.

1. The revised manuscript demonstrated that the false positives were controlled well while the false negatives not when segmentation error happened. This observation means the increased segmentation error only influence the testing power (increase the Type II error), not the testing size (does not increase Type I error). Figures 6g and 5.6 demonstrate that simulated completely random expressions exhibit a noteworthy distinction in total variations from non-random expressions, discernible to the naked eye. However, this strong signal case is not of interest to researchers, as statistical testing can effortlessly detect non-random effects. The question arises: is the control of Type I error solely due to the strong signal exhibited by the two types of expressions? If there is minimal overlap in total variations, can the Type I error still be effectively controlled?

2. Fig 5.12 panel(b): why are the x-coordinates of the red curve less than 0? meaning < 0 mm in distance?

3. Supplementary Figure 14, the function `SPATA2::spatialAnnotationScreening()` is not found in the package `SPATA2` and <https://themilolab.github.io/SPATA2/reference/index.html>. And what is the meaning of five iterations?

Minors:

1. Typos in the caption of main Figure 3, " Hmox1 and Lcn2.) Ridgeplots visualize the expression gradients of genes". Different font size in the legend of Figure 3b. Please carefully check the possible typos.

2. Line 210, main text, the gene symbol names of mouse tissue should be italicized.

Reviewer #2 (Remarks to the Author):

In this nearly completely rewritten manuscript by Kueckelhaus and colleagues, the authors propose that screening for genes whose expression pattern across a tissue follow user-defined shapes is an advantageous approach to current spatial gene expression analysis strategies.

This workflow proposes users annotate polygons of biological interest based on paired images, then screen genes based on their expression as a function of the distance from the polygon.

The main text is structured such that anecdotes are used to argue for certain pattern choices in numerous settings, where patterns are created, and genes following the patterns are identified. There are some visual illustrations of genes that are identified with this strategy, but not with alternative strategies. The main text mostly serves as a demonstration of using the tool, though there are some comparisons that could warrant some more specifics about the results (listed in 'minor issues' below).

The main novelty is the proposal that genes should follow certain predefined patterns to be of interest. It is hard to argue either for or against this proposal. It would require some database of interesting genes and their patterns, and then see if they tend to follow these patterns. But this does not exist.

We recognize the concern about the current lack of a database for analyzing gene expression in relation to spatial niches. Our manuscript introduces an algorithm aimed at exploring this novel area of research. As outlined in sections two through five, we provide initial evidence of the algorithm's application across both well-established and new biological contexts. These sections are intended to show the preliminary utility of our approach by aligning it with known biological phenomena and suggesting its ability to generate new insights. We hope our work will contribute to the early stages of developing a database for such specialized analyses. Our algorithm's main objective is not solely to identify gene patterns but to offer a more nuanced classification of spatial niches by examining their surroundings. This involves identifying genes whose expression is significantly associated with the spatial characteristics of specific niches, such as those observed in histological microstructures commonly used in medical diagnostics. This exploratory work aims to contribute to a better classification and understanding of heterogeneous pathologies, including CNS tumors, by identifying non-random patterns of genes or other numeric features in the vicinity of certain niches. While we are optimistic about the potential of our algorithm to enrich our understanding of complex biological and pathological landscapes, we are mindful of the preliminary nature of this research and the collaborative efforts required to advance this field further.

Secondarily, the authors have invented a statistical test based on total variation of a LOESS smoother. This ought to be compared in its statistical performance with classically known statistical tests.

Major issues —

The null and alternative hypotheses described by the authors are the same as the null and alternative hypotheses that define the task of identifying spatially significant genes. The authors have transformed the original coordinate system to a 1-dimensional coordinate system reflecting distance to the annotated polygon, but existing methods such as e.g. SPARKX used by the authors will solve the same problem along the trajectory.

We appreciate the opportunity to clarify our position and respectfully highlight a distinction in our approach. The null and alternative hypotheses of our proposed tests, designed to identify genes exhibiting spatially significant or non-random patterns, share conceptual similarities with approaches such as SPARKX. But they are not the same and a crucial difference underpins our methodology: Our tests specifically aim to discern genes (or other numeric features) that manifest non-random expression patterns in relation to defined spatial references, such as delineated areas or spatial trajectories. The following explanation seeks to underscore the different nature of both approaches and further illustrates how they are designed to be compatible rather than competitive.

We propose that the spatial expression patterns of certain genes are closely linked to their distance from certain areas within the tissue. When represented as a function of this distance, these patterns statistically stand out compared to genes that do not adhere to this hypothesis and whose expression patterns are not influenced by proximity to these areas. Identifying these genes and their derived expression patterns is driven by the belief that such patterns, in relation to the area of interest, can offer deeper biological insights into the formation of these areas or the effects their presence has on surrounding tissues. For instance:

A gene whose expression increases with proximity to an area might indicate the tissue's (or cells') response to the presence of that area. Main Figure 3 uses the deliberate infliction of an injury in the mouse cortex as an example to illustrate this line of thought. The transcriptional profile for most of the tissue in this sample aligns with the well-researched and understood structure of the central nervous system, except for the area surrounding the injury. Given that introducing this injury was the only alteration made to the tissue, it's reasonable to assume that any transcriptional deviations in the vicinity of the injury are largely (if not solely) attributable to its presence. This injury impacts its environment, and to better understand the nature and biology of the injury, it's essential to comprehend its effect on the surrounding area by identifying genes that characterize this impact. We hypothesize that the expression patterns of genes affected by the injury will present as "non-random" when depicted as a function of distance to the injury, as opposed to genes unaffected by the injury's presence. Identifying genes whose expression patterns are influenced by the injury, thereby highlighting biological aspects of the injury, constitutes the initial step in our screening and the statistical methodology we propose.

SPARKX also aims to identify genes of spatial interest. However, SPARKX evaluates the sample as a whole and does not accommodate the integration of spatial references indicative of potential biological forces, such as a stab wound in the mouse cortex. Given SPARKX's goal to identify genes showing spatial variability (or spatial non-randomness), it logically detects all genes disrupted by the injury since their expression pattern in 2D space is non-random. Nonetheless, SPARKX will also

capture genes that conform to the highly structured architecture of the mouse cortex, irrespective of their influence by the injury, since their expression patterns are inherently non-random. Thus, using SPARKX alone a researcher focusing solely on genes impacted by the injury will invariably encounter genes deemed spatially significant for reasons other than the injury. Without the capability to incorporate the delineated area of the injury, SPARKX's output cannot distinguish between genes whose spatial variability stems from the injury and those non-randomly expressed in the neocortex for other reasons.

We hope this explanation clearly delineates the distinction between our approach and algorithms like SPARKX, as well as the difference between the null and alternative hypotheses of these tests. We do not intend to replace SPARKX but rather to extend it, by offering a statistical approach to embed the genes identified by algorithms like SPARKX in a more precise hypothesis for further filtering. We recognize, however, that our original phrasing of both, null- and alternative hypothesis, may not have adequately emphasized this critical distinction and have accordingly revised our wording. For each gene tested we formulate the following hypotheses:

Null Hypothesis (H0): The expression pattern of the tested gene does not show spatial significance in relation to specific spatial references, such as delineated areas or spatial trajectories, and is attributable to random chance rather than being influenced by proximity to defined areas within the tissue.

Alternative Hypothesis (H1): The expression pattern of the tested gene exhibits spatial significance when analyzed in relation to specific spatial references, such as delineated areas or spatial trajectories. It forms a recognizable pattern that statistically distinguishes it from genes whose expression is not influenced by proximity to these areas, suggesting a biologically meaningful connection between the gene and the reference feature.

Further, since this is now a 1-d problem, all methods that have been proposed to identify genes that vary along pseudotime trajectories also follow the same null and alternative hypotheses. It would be reasonable to compare the novel statistical test defined here based on total variation with 1-dimensional application of spatial significance methods and pseudotime-dependent gene expression identification methods.

We are grateful to the reviewer for highlighting this aspect. In response, we conducted a comparative analysis of our method's test statistic—the total variation (TV) of a LOESS fit—against two algorithms specifically designed for analysis along pseudotime trajectories, namely tradeSeq and PseudotimeDE. Considering the fundamental purpose of a test statistic is to delineate significance from randomness, we believe that its ability to accurately identify introduced randomness (noise) serves as a crucial benchmark. This criterion allows us to evaluate and contrast the effectiveness of different test statistics relative to our approach, ensuring a comprehensive assessment of their predictive capabilities in distinguishing true biological signals from noise.

First, we conducted a comparison between SPATA2's Spatial Gradient Screening and tradeSeq's associationTest, an established pseudotime-based differential expression test. tradeSeq's associationTest evaluates “gene expression is associated with pseudotime along a given lineage, i.e., whether the smoother is flat or varying along pseudotime”¹. We juxtaposed the ground truth noise of

¹ Van den Berge, K., Roux de Bézieux, H., Street, K. et al. Trajectory-based differential expression analysis for single-cell sequencing data. *Nat Commun* **11**, 1201 (2020). <https://doi.org/10.1038/s41467-020-14766-3>

our simulated dataset against tradeSeq (Wald statistic, referred to as "waldStat" in tradeSeq) using the same simulated expression dataset previously employed to assess total variation's (TV) ability to predict the degree of underlying randomness. This approach, discussed in our response to a major concern raised by Reviewer 3, mirrors the statistical challenge addressed by both SPATA2 SGS and tradeSeq when substituting pseudotime with distance to the ground truth outline.

Our findings, illustrated in Figure 2a-b, reveal that the proposed waldStatistic test significantly correlates with the introduced noise, validating its effectiveness in identifying non-random patterns along a one-dimensional axis. Furthermore, tradeSeq's waldStatistic demonstrated more consistent results across various patterns, except for the "small peak" pattern, compared to our approach (Supplementary Figure 11a). However, the key advantages of our proposed test statistic are underscored by several critical differences:

1. SPATA2 SGS's Total Variation (TV) demonstrates a more robust correlation with ground truth noise levels than tradeSeq's waldStatistic. In the scenario we used (annotation: necrotic_center, distance = 3mm) SAS shows a TV R^2 of 0.75 compared to the Wald Statistic's R^2 of 0.59 when correlating the test statistics and the introduced noise across all patterns.
2. Our approach also benefits from significantly reduced computational time. In direct comparison, processing ten thousand simulated expression variables through the tradeSeq pipeline required 61 minutes on a high-capacity cluster. In contrast, SPATA2's Spatial Annotation Screening completed the same task in a matter of a few minutes on a personal laptop, highlighting our method's efficiency. See Supplementary Figure 14e for our detailed benchmarking results.
3. Unlike tradeSeq, which mandates manual tuning of the number of knots for its smoother - a process requiring visual inspection (as per tradeSeq::evaluateK) - our method does not necessitate such parameter adjustments, simplifying the analysis process and making it more robust to variation in parameter tuning approaches.

All in all, though tradeSeq offers a comparable methodology, it falls short in several areas, emphasizing the benefits of our proposed approach.

In a further analysis, we evaluated another pseudotime-based algorithm, PseudotimeDE², focusing again on its capacity to discern noise levels camouflaging the original pattern. Displayed in Figure 2.1 c-d, PseudotimeDE's test statistic demonstrated commendable performance with an overall R^2 of 0.67. Despite its closer proximity to our method's precision, PseudotimeDE's computational demands significantly detracted from its utility. Processing ten thousand simulated variables on the aforementioned 256GB RAM cluster surpassed a 24-hour timeframe. To feasibly conduct this analysis, we reduced the dataset to a thousand variables, which still necessitated over ten hours of processing time on the same device. Considering the decrease in accuracy and the prohibitive computational duration - especially in the real-life context of datasets often exceeding ten thousand genes - we conclude that PseudotimeDE does not present a viable alternative to our methodology.

Conclusively, established pseudotime-based algorithms exhibit potential in linear noise prediction within expression patterns related to spatial reference features. Still, their less precise noise

² Song, D., Li, J.J. PseudotimeDE: inference of differential gene expression along cell pseudotime with well-calibrated p-values from single-cell RNA sequencing data. *Genome Biol* 22, 124 (2021). <https://doi.org/10.1186/s13059-021-02341-y>

prediction and notably longer computation times compared to our method highlight the advantages of our approach in both prediction accuracy and computational efficiency.

Figure 2.1: Investigations into the ability of different test statistics to predict the degree of noise introduced to hide a ground truth pattern. a) A scatterplot shows the relationship between introduced noise and the Wald Statistic used by tradeSeq across all ground truth pattern. b) Scatterplots show the relationship between noise and the Wald Statistic for each single pattern individually. c) A scatterplot shows the relationship between introduced noise and the test statistic used by PseudotimeDE across all ground truth pattern. b) Scatterplots show the relationship between noise and PseudotimeDE test statistic for each single pattern individually.

Minor issues —

Around line 169, it is not clear how many genes do not reflect the corresponding regions, compared to how many that do.

To address the reviewer's concern, we developed models that delineate the spatial extent of specific areas along the trajectory. Using these models, we screened for genes specific to each area and compared the findings with group-specific genes identified by Differential Expression Analysis (DEA). Our comparison revealed variations in the number of genes identified across different zones, notably between the tumor and transition zones, while differences in the infiltrated cortex zone were minimal. This discrepancy underscores the transcriptional distinctiveness of the infiltrated cortex zone and the shared characteristics between the tumor and transition zones. Furthermore, our analysis demonstrates the Spatial Transcriptomics Screening (STS) method's ability to detect these subtle spatial differences, which DEA does not capture as effectively. Specifically, DEA often misclassifies genes that cross into the adjacent transition zone as exclusive to the tumor zone. This example highlights the potential of our approach to provide details guiding specifically spatial analysis, embedding genes identified by other methods in a supervised spatial context.

Around line 216, it is not clear how many genes were not identified by clustering DE analysis but were detected by the trajectory analysis. How many known genes from the different categories of immune activity, immune cell types, migration, proliferation, and wound healing, beyond the few genes highlighted?

We appreciate the opportunity to provide more insights into our results. To precisely quantify the additional genes identified by Spatial Annotation Screening (SAS) beyond those detected by Differential Expression Analysis (DEA) across categories such as immunity, cellular migration, cellular proliferation, and wound healing, we compiled four comprehensive gene lists. This was accomplished by sourcing a broad array of gene sets from the Gene Ontology Biological Process (GO:BP) mouse database. We specifically sought gene sets tagged with the keywords 'immunity', 'migration', 'proliferation', or 'wound healing'.

Subsequently, we examined these lists, tallying the genes identified by both SAS and DEA within each category. The findings of this analysis are showcased in Figure 2.2e-f. Our results reveal that while DEA successfully identifies a multitude of genes, SAS notably enriches this identification, uncovering a substantial number of additional genes. This underscores the enhanced capacity of SAS to complement traditional analysis methods by integrating more spatial context into the results, thereby enriching hypothesis-driven research with spatially nuanced insights.

Figure 2.2: Visual accompaniments to our responses to the minor concerns raised by Reviewer #2. *a-b)* Comparison of the number of genes identified as marker genes highlights how STS identifies spatial subtleties such as border transgressing genes and those that do not. *c-d)* Example genes identified by STS as actual spatial marker genes of the denoted areas. See Main Figure 2 in the manuscript for examples of genes that do cross the border between tumor and transition zone. *e)* Example genes identified by SAS to be non-random in the light of the presence of the inflicted damage on the mouse cortex. *f)* Barplots quantify the number of genes related to important biological functions such as immunity, cellular migration, cellular proliferation and wound healing not identified by DEA but by SAS. *g)* Barplots highlight the degree of similarity of supposed marker genes identified by DEA.

Around line 243, within what log₂ fold change threshold are the changes considered 'nearly identical'?

We appreciate the chance to elaborate on our methodology regarding the classification of average log₂ fold change (log₂FC) as 'nearly identical'. We did not establish a strict numerical threshold for this designation. Instead, the characterization emerged from an observational analysis conducted during our exploration. Specifically, our investigation aimed to gain a nuanced spatial understanding of gene expression through the application of BayesSpace in conjunction with Differential Expression Analysis (DEA), a goal that proved challenging due to the variability in gene expression patterns, as highlighted in Supplementary Figure 4. In the context of our DEA results, which showcased a range of average log₂FC values from 0.25 to 2.5, we observed instances where average log₂FC values such as 0.72 and 0.83 were remarkably close, prompting us to consider these as 'nearly identical'. This observation is further exemplified in Figure 2.2g, where we demonstrate that the average log₂FC between two distinct groups or areas can be virtually indistinguishable, underscoring the frequent overlaps in spatial gene expression within our sample. This scenario exemplifies why we advocate for a methodological pivot towards screening gene expression patterns using gradients. Unlike traditional approaches, gradient-based screening appreciates the continuous nature of gene expression. Our methodology leverages spatial annotations and trajectories, along with predefined models, to provide a more coherent framework for understanding spatial gene expression patterns. This approach not only addresses the limitations encountered with BayesSpace and DEA but also offers a more sophisticated tool for interpreting complex spatial gene expression data.

Around line 274, is there a citation for the prior findings about TCR signaling near necrotic regions?

We thank the reviewer for pointing out this lack of detail in our citations. In fact, numerous studies have explored the relationship between hypoxia and T-cell activity, yet the exploration of this interplay within a spatial context remains relatively uncharted. A notable contribution to this field is the work by Benotmane, J.K., Kueckelhaus, J., Will, P., et al., who presented "High-sensitive spatially resolved T cell receptor sequencing with SPTCR-seq" in Nature Communications³. This study provides valuable insights into the spatial organization of T-cells in relation to hypoxia and necrosis.

During our examination of the spatial arrangement of T-cells as detailed in Benotmane et al., 2023, we identified a potential necrosis/hypoxia-dependent spatial organization of T-cells. At that time, however, we lacked a suitable tool and method for validating this hypothesis, especially one that could incorporate multiple samples as effectively as demonstrated in Main Figure 5 of our work. The findings related to hypoxia-dependent T-cell formation, as highlighted in our study, serve not only to confirm existing hypotheses but also to illustrate the potential of Spatial Annotation Screening (SAS) in uncovering or confirming new hypotheses from genes identified as spatially variable, employing tools such as SPARKX.

³ Benotmane, J.K., Kueckelhaus, J., Will, P. et al. High-sensitive spatially resolved T cell receptor sequencing with SPTCR-seq. Nat Commun 14, 7432 (2023). <https://doi.org/10.1038/s41467-023-43201-6>

Reviewer #3 (Remarks to the Author):

The authors have addressed most of the concerns, however, there are still some issues that remain unclear.

Majors:

1. The revised manuscript demonstrated that the false positives were controlled well while the false negatives not when segmentation error happened. This observation means the increased segmentation error only influence the testing power (increase the Type II error), not the testing size (does not increase Type I error). Figures 6g and 5.6 demonstrate that simulated completely random expressions exhibit a noteworthy distinction in total variations from non-random expressions, discernible to the naked eye. However, this strong signal case is not of interest to researchers, as statistical testing can effortlessly detect non-random effects. The question arises: is the control of Type I error solely due to the strong signal exhibited by the two types of expressions? If there is minimal overlap in total variations, can the Type I error still be effectively controlled?

We are grateful to the reviewer for highlighting the necessity for further clarification and welcome the chance to provide more insight into our reasoning and the additional simulations we undertook to address this concern. Our response is structured into three sections: The first two offer a detailed overview of our thought process behind the simulations and revisit our previous approach, shedding light on a challenge we faced that is directly related to your concern. The third section presents the outcomes of our supplementary efforts conducted in response to both this challenge and your concern.

Elaboration on our line of thought

Building on our analytical framework, which examines gene expression patterns as a function of distance to spatial niches, we further explain the rationale behind our simulation approach that introduces noise into a perfectly smooth pattern in incremental steps.

In response to the major concerns of reviewer #2, we have already elaborated on our line of thought when speaking of spatial niche dependent expression pattern. We assume that a spatial niche (denoted by a spatial annotation) either influences or is influenced by distinct gene expression patterns. This concept is illustrated in Supplementary Figure 10b through six scenarios, each depicting how the presence of the annotation we labeled "necrotic_center" affects the expression of six simulated genes, represented by six distinct models named after the pattern they exhibit. It goes without saying that it is virtually impossible to obtain such smooth gene expression pattern when conducting real life experiments. Accepting our hypothesis momentarily, that the presence of specific spatial niches correlates with unique gene expression patterns in its environment, it is still save to assume that a spatial niche's influence on gene expression is not isolated. Two kinds of additional factors most likely have an impact on gene expression patterns, too:

- Biological noise: Various additional biological factors, such as neighboring spatial niches, metabolic changes, and the inherent tissue architecture just to name a few.*
- Experimental design technicalities and artefacts.*

We refer to the influence of both kinds on the gene expression pattern as (background) noise. This consideration leads us to define prerequisites for validating our null and alternative hypotheses:

For the null hypothesis (H0): The linkage between a spatial niche and the expression of a gene is either insufficiently strong or non-existent, failing to distinguish itself against the background noise from other biological factors and experimental artifacts.

For the alternative hypothesis (H1): The linkage between a spatial niche and the expression of a gene is robust enough to rise above the background noise from other biological factors and experimental artifacts. This marks it as spatially significant and thus of potential interest.

While technical artifacts are expected to uniformly affect all genes, biological variation likely varies in its impact. (E.g. an adjacent spatial niche might have a different impact on gene X than on gene Y.) Consequently, for a gene's linkage to a spatial niche to be deemed significant, the influence of the niche must predominate in a biological sense. We aimed to integrate this strength of linkage between a niche and a gene in our simulations by integrating incremental steps of noise. In simple terms, a simulation with 30% noise models a gene expression pattern that is 70% influenced by the niche and 30% by other factors. Conversely, a simulation with 88% noise models a gene expression pattern where only 12% is attributable to the niche, with the remaining 88% resulting from other influences. Furthermore, our hypothesis assumes the existence of an optimal outline for the niche. In theory, this optimal outline best represents the spatial extent of the niche and, when utilized to deduce the expression pattern, should result in the smoothest pattern achievable.

Summary of our previously conducted benchmarking

In response to your concern regarding the potential impact of segmentation errors or outline errors introduced by human variability, we had simulated a dataset and had treated our “necrotic_center_outline” based on which we simulated the data as the optimal outline, the ground truth outline. We then proceeded to add incremental steps of segmentation error to the optimal outline simulating increasing degrees of segmentation error and aimed to quantify the effect on the screening results. Using the concept of simulated expression variables with incremental noise percentage we created a ground truth of true positives and true negatives based on the proposed optimal outline and repeated the screening with all deviating outlines. False negatives were identified as genes that, based on the optimal outline, showed non-random expression but were deemed random when assessed against an altered outline with a certain degree of deviation. False positives were the opposite: genes considered random under the optimal outline but identified as non-random with an outline that deviated to a certain extent.

In this response to your concern, we revisit our approach to highlight a challenge encountered during our simulation and benchmarking, which directly relates to your concern: How do we define true positive and true negative gene expression patterns in relation to the ground truth niche outline? Specifically, how pronounced must a niche's influence on a gene's surrounding expression be for the pattern to be clearly recognized as non-random? Conversely, how minimal must this influence be for the pattern to be considered random, indicating no significant association or linkage?

Reviewer 2 rightly points out the absence of a ground-truth database for establishing such positive and negative benchmarks due to the innovative nature of our approach. The challenge we faced was to define a ground truth of true positives and true negatives that, for one thing, undoubtedly matched

the hypothesis it corresponded to (H0 or H1) and, for the other thing, represented gene expression pattern that were realistically to be expected in real life scenarios (unlike the pattern represented in Supplementary Figure 10b).

Using the noise percentage to categorize a simulation as either positive or negative would require a manually set threshold. This in turn, brings up additional problems. For instance, while gene expression at 100% noise is clearly random, arguments could also support the “true randomness” of expressions at noise levels of 99%, 98%, 97% and so on. Conversely, expressions at 0% noise are clearly non-random, but including them as true positives would skew results away from realistic expectations, as such perfect non-randomness is nearly impossible to encounter in real-world scenarios.

Therefore, to establish the positive ground truth, we had opted for the total variation of each simulation when inferred as a gene expression gradient from the optimal outline and the adjusted p-values derived. True positives were declared as simulations deemed significant through their adjusted p-values (FDR) of 0, calculated using the total variation score as detailed in our methods section. To avoid including simulations with unrealistically low total variation, we filtered for those within the range of total variation scores derived from applying real gene expression data to a screening with the same set of spatial annotation, distance and resolution. This provided a benchmark for expected total variation scores with real data. This process established our positive ground truth.

In the last version of the manuscript, the negative ground truth was established by using simulations of 100% noise resulting in a distribution of total variation scores that deviated significantly from the positive ground truth. Before we detail our extended simulation and benchmarking efforts in response to the reviewer’s concern, we wish to clarify a key point.

(Here, we assume that you meant to say “If there is maximal overlap in total variations, can the Type I error still be effectively controlled?” since there is already minimal overlap in total variations in the referenced Supplementary Figure 6g.)

The algorithm’s decision on whether an expression variable is deemed random hinges on the total variation score of an inferred gradient. Consequently, by our methodology’s design, the positive and negative ground truths are inherently non-overlapping. Specifically, an expression variable classified into the negative ground truth but assessed with a total variation score aligning with the positive ground truth must be considered non-random. This is because the total variation score is the metric used to discern if an inferred gradient is random. Therefore, if a total variation score aligns with the distribution typical of the alternative hypothesis, it cannot be considered random, excluding it from the true negative group and vice versa. Thus, the positive and negative ground truth cannot overlap, by definition. However, we recognize the value of additional simulations and benchmarking with a negative ground truth that lies closer to the positive ground truth.

(Figure 3.1 See following page for the figure description.)

Figure 3.1: Additional simulation and benchmarking against segmentation error with a new negative ground truth. a) Barplot of all simulations above fifty percent of introduced noise. (436 simulations per one percentage of noise increase). Colors indicate their status in terms of spatial significance when evaluated using SAS on the ground truth outline. b) A densityplot contrasts the new distribution of total variation in the new negative ground truth with the one used in the previous version of the manuscript. c-f) Examples of simulated gene expression evaluated as random ($FDR > 0.05$) when using the ground truth outline. g-h) Density plots of total variation distribution for each subpopulation of simulations under 70-79% of noise and 80-89% of noise, respectively. i-j) Dotplots contrast the results of the simulations with the previously used ground truth with the new negative ground truths.

Additional simulations and benchmarking

To obtain such a negative ground truth we created another larger set of simulations using the linear descending models as the ground truth pattern with incremental steps of equally distributed noise and evaluated their total variation and their spatial significance in relation to the ground truth outline. Then, we declared the negative ground truth to consist of simulated genes with an adjusted p -value (FDR) of 0.05 or higher. Essentially, the negative ground truth consisted of genes not identified as significant using the ground truth outline.

Figure 3.1a displays a bar plot categorizing simulations across noise percentage (NP) levels in increments of 1%. The plot employs colors to distinguish among " $FDR < 0.05$ ", " $FDR > 0.05$ & $NP < 90\%$ ", " $FDR > 0.05$ & $NP \geq 90\%$ ". It illustrates that, within the 90-99% noise range, none of the 436 simulations per percentage level were deemed spatially significant. Figure 3.1b presents the distribution of total variation, which aligns more closely with the positive ground truth.

This method generated simulations with a wider range of noise percentages than our initial approach, which exclusively used variables with 100% noise. We've categorized the results into two subsets: simulations with noise percentages from 70-79% and those from 80-89%. Figures 3.1c-f feature sample simulations from both categories, while Figures 3.1g-h compare their distributions against those from the previous revision. Additionally, Figures 3.1i-j juxtapose the benchmarking outcomes against segmentation error using the updated negative ground truth with those using the prior version. Although the subgroup of 70-79% noise percentage saw a slight uptick in false positives, this increase remained under 10% and well within control. Notably, the 80-89% noise percentage subgroup did not show a significant rise in false positives. Given a hypothesized true negative ground truth comprising genes within the 70-100% noise range, we anticipate the rate of false positives to fall between 3-7%. It's important to highlight that spatial significance precedes model matching, a crucial step that further filters genes since spatial significance alone does not guarantee biological relevance. The biological significance is determined through models that researchers apply to ascribe biological meaning to identified genes, thereby refining the list of genes for final analysis and interpretation.

2. Fig 5.12 panel(b): why are the x-coordinates of the red curve less than 0? meaning < 0 mm in distance?

We apologize for any confusion. The negative distances shown in the figure arise from including the area inside the spatial annotation, which we term the "Core." As outlined in Supplementary Figure 7, we categorize the areas related to the reference annotation into three zones: Core (inside the annotation), Environment (immediate surrounding up to a certain distance, e.g., 3mm), and Periphery (areas beyond the Environment, not considered in the screening). Throughout the paper, our focus

has primarily been on the Environment and the Core of Spatial Annotations has not been included when inferring the expression gradient of a gene.

However, in Figure 5.12 (last rebuttal letter), we included the Core to demonstrate that the zone we annotated based on high hypoxic gene expression indeed features high hypoxia associated gene expression. Here, a distance of 0 marks the border between the Core and the Environment, showcasing a transition from high to low expression levels at this boundary in the context of hypoxia. Fig. 3.2 below extends the previous version of the referenced Fig. 5.12 highlighting the position of the border with a vertical line.

Spatial Annotation Screening allows for the inclusion or exclusion of the Core in the gene expression pattern analysis, adjustable by the user with the parameter `core = TRUE` or `core = FALSE`. An example for a scenario in which it makes sense to set `core = TRUE` would be, if genes that peak at the border of the spatial annotation are of interest. To identify those one needs to include the gene expression of the annotation core as well as the gene expression of the immediate environment. For this particular visualization (Fig.3.2b), we set `core = TRUE` to affirm the accuracy of our spatial annotation in capturing hypoxia-related gene expression which drops significantly at the border which outlines the area predicted to feature high expression of hypoxic signatures.

Fig. 3.2 a) Shows the surface plot colored by the distance towards the border. Notice the color scale which assigns values below 0 to the spots inside the border (inside the core). b) A vertical line highlights the positioning of the border when inferring the gradient of HM_HYPOXIA (based on which the spatial annotation was created).

3. Supplementary Figure 14, the function `SPATA2::spatialAnnotationScreening()` is not found in the package SPATA2 and <https://themilolab.github.io/SPATA2/reference/index.html>.

We regret any confusion caused. The function `SPATA2::spatialAnnotationScreening()` emerged during recent revisions and was previously known as `SPATA2::imageAnnotationScreening()`. As these revisions are ongoing, the updated functions have not yet been merged into the main branch. Currently, they are available in the development branch labeled "devJK." Within this branch, functions and their documentation are located in the /R directory, with R scripts named after the initial letters of the functions they contain. Hence, you can find the `spatialAnnotationScreening()` function within the R/s.R script. We intend to merge these updates into the main branch and reflect them on the SPATA2 website following the publication of our work.

And what is the meaning of five iterations?

We apologize for the lack of clarification. Five iterations referred to the number of times we ran the algorithm to benchmark the computational time required. In the previously submitted manuscript, we ran the algorithm five times for each condition (number of genes included). We noticed, however, that the number of spots included in the screening (increased with the distance parameter) had a significant effect on the runtime, too. Therefore, we repeated the benchmarking. Furthermore, we increased the number of iterations to 15 as displayed in Supplementary Figure 14e.

Minors:

1. Typos in the caption of main Figure 3, “ Hmox1 and Lcn2.) Ridgeplots visualize the expression gradients of genes”. Different font size in the legend of Figure 3b. Please carefully check the possible typos.

We thank the review for bringing this to our attention. Following the feedback, we have revised the script, ensuring corrections for typos and formatting issues.

2. Line 210, main text, the gene symbol names of mouse tissue should be italicized. *We thank the review for bringing this to our attention. Following the feedback, we have revised the script, ensuring corrections for typos and formatting issues.*

Reviewers' Comments:

Reviewer #2:

Remarks to the Author:

In this minor revision of their previous manuscript, the authors have added a couple of comparisons with alternative methods as a final figure, and conducted an increased number of simulation studies of their proposed method.

The manuscript has been improved.

Reviewer #3:

Remarks to the Author:

The authors addressed all of my concerns.

REVIEWERS' COMMENTS

Reviewer #1 (Remarks to the Author):

No remarks made.

Reviewer #2 (Remarks to the Author):

In this minor revision of their previous manuscript, the authors have added a couple of comparisons with alternative methods as a final figure, and conducted an increased number of simulation studies of their proposed method.

The manuscript has been improved.

We thank the reviewer for the feedback.

Reviewer #3 (Remarks to the Author):

The authors addressed all of my concerns.

We thank the reviewer for the feedback.